# Histone H2B.8 compacts flowering plant sperm through chromatin phase separation

Toby Buttress[1,4], Shengbo He[1,4], Liang Wang[2,3,4], Shaoli Zhou[1,4], Gerhard Saalbach[1], Martin Vickers[1], Guohong Li[3], Pilong Li[2✉] & Xiaoqi Feng[1✉]

Sperm chromatin is typically transformed by protamines into a compact and transcriptionally inactive state[1,2]. Sperm cells of flowering plants lack protamines, yet they have small, transcriptionally active nuclei with chromatin condensed through an unknown mechanism[3,4]. Here we show that a histone variant, H2B.8, mediates sperm chromatin and nuclear condensation in *Arabidopsis thaliana*. Loss of H2B.8 causes enlarged sperm nuclei with dispersed chromatin, whereas ectopic expression in somatic cells produces smaller nuclei with aggregated chromatin. This result demonstrates that H2B.8 is sufficient for chromatin condensation. H2B.8 aggregates transcriptionally inactive AT-rich chromatin into phase-separated condensates, which facilitates nuclear compaction without reducing transcription. Reciprocal crosses show that mutation of *h2b.8* reduces male transmission, which suggests that H2B.8-mediated sperm compaction is important for fertility. Altogether, our results reveal a new mechanism of nuclear compaction through global aggregation of unexpressed chromatin. We propose that H2B.8 is an evolutionary innovation of flowering plants that achieves nuclear condensation compatible with active transcription.

Sperm chromatin undergoes extensive condensation that is essential for male fertility in most animals. During animal sperm maturation, nearly all histones are replaced by small, arginine-rich protamines that facilitate dense packaging of the DNA[1,2]. In mammalian sperm for example, histones are retained at only 5–15% of the genome, whereas the majority of regulatory information carried by histones is lost[1,5]. Unwinding DNA from histones requires DNA strand breaks, which are observed in great quantities during this process[6,7]. Furthermore, protamines preclude transcription[8]. The extreme chromatin compaction enabled by protamines protects genome integrity against genotoxic factors and achieves a small and hydrodynamic sperm head that enhances swimming ability[2,8]. The adoption of such a process illustrates the high evolutionary pressure on sperm fitness[9].

Sperm condensation also occurs in another large group of multicellular eukaryotes: plants. Similar to animals, green algae and non-seed plants (such as liverworts, mosses and ferns) produce motile sperm, which swim through water to reach the egg cell[10]. Consistent with the theory that protamine-mediated sperm condensation evolved to facilitate swimming[2], sperm nuclei in these species are highly condensed by protamines and protamine-like proteins and transcribe very little, if any, RNA[11–14].

Diverged from other land plant species approximately 150 million years ago, flowering plants no longer rely on water for fertilization[10]. Flowering plants produce immotile, transcriptionally active sperm[3,4]. The sperm cells are encapsulated in pollen grains, which are produced by mitotic divisions of the haploid meiotic product called the microspore[10].

The microspore divides once to produce a vegetative cell and a generative cell, the latter of which subsequently divides to generate two sperm cells[10]. During fertilization, the vegetative cell develops into a pollen tube that delivers the sperm to the egg apparatus[15]. Consistent with the high metabolic activity required for powering pollen tube growth, chromatin in vegetative cells is highly decondensed and transcriptionally active[16–18]. By contrast, sperm has highly condensed, histone-based chromatin and small nuclei[19,20]. In the absence of protamines, the mechanism of sperm chromatin condensation in flowering plants is unknown.

To understand the mechanism that underlies sperm condensation in flowering plants, we performed super-resolution imaging and comparative proteomics analyses of sperm cells, vegetative cells and somatic cells from *A. thaliana*. Through these experiments, we identified a specifically expressed histone variant, H2B.8, that colocalizes with chromatin aggregates in the sperm nucleoplasm. Sperm nuclei with *h2b.8* mutations are enlarged and have decondensed chromatin, whereas ectopic expression of H2B.8 in somatic cells causes the opposite phenotype. This result demonstrates that H2B.8 is necessary and sufficient for nuclear and chromatin condensation. H2B.8 aggregates chromatin through a phase-separation mechanism that depends on a conserved intrinsically disordered region (IDR). H2B.8 specifically concentrates unexpressed AT-rich euchromatin, which reduces the nuclear volume while maintaining transcription. H2B.8-induced nuclear compaction is important for fertility, as *h2b.8* mutations reduce male transmission. Collectively, our results explain flowering plant sperm

[1]Cell and Developmental Biology Department, John Innes Centre, Norwich, UK. [2]Beijing Frontier Research Center for Biological Structure, Beijing Advanced Innovation Center for Structural Biology, Tsinghua University–Peking University Joint Center for Life Sciences, School of Life Sciences, Tsinghua University, Beijing, China. [3]Institute of Biophysics, Chinese Academy of Science, Beijing, China. [4]These authors contributed equally: Toby Buttress, Shengbo He, Liang Wang, Shaoli Zhou. ✉e-mail: pilongli@mail.tsinghua.edu.cn; xiaoqi.feng@jic.ac.uk

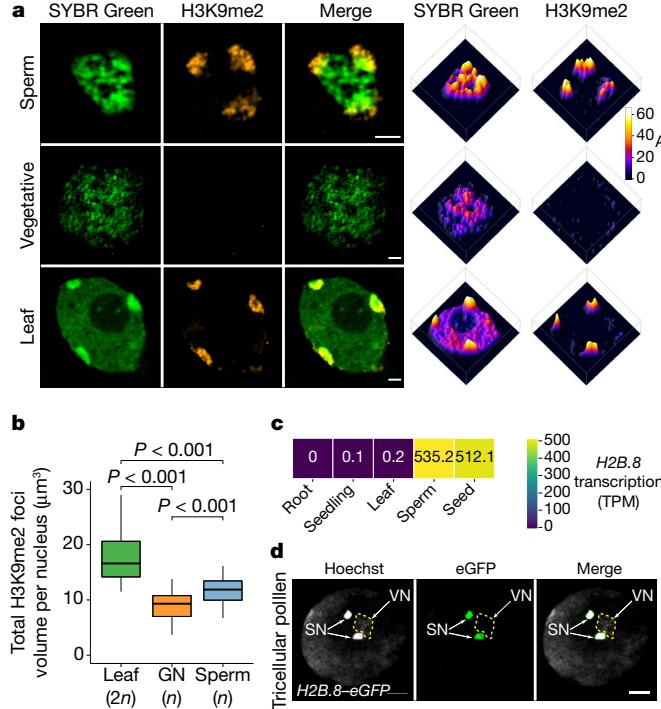

**a** SYBR Green | H3K9me2 | Merge | SYBR Green | H3K9me2

Rows: Sperm, Vegetative, Leaf

(color scale 60, 40, 20, 0 AU)

**b** Total H3K9me2 foci volume per nucleus (μm³)

$P < 0.001$ (Leaf–GN)
$P < 0.001$ (Leaf–Sperm)
$P < 0.001$ (GN–Sperm)

Leaf (2*n*) | GN (*n*) | Sperm (*n*)

**c**

| Root | Seedling | Leaf | Sperm | Seed |
|------|----------|------|-------|------|
| 0 | 0.1 | 0.2 | 535.2 | 512.1 |

*H2B.8* transcription (TPM) — scale 500, 400, 300, 200, 100, 0

**d** Tricellular pollen

Hoechst | eGFP | Merge

*H2B.8–eGFP*

VN, SN labels

**Fig. 1 | Sperm chromatin is aggregated and contains a specific histone variant: H2B.8. a**, Super-resolution 3D-SIM images (left) and associated intensity profiles (right) of wild-type (WT) sperm and vegetative nuclei from pollen and a diploid leaf nucleus. DNA was stained with SYBR Green (green) and H3K9me2 was immunolocalized (orange). Data shown represent three independent experiments. AU, arbitrary units. **b**, Total volume of H3K9me2-enriched heterochromatin foci in a diploid leaf nucleus, generative nucleus (GN) and sperm nucleus. *P* values calculated using one-sided analysis of variance (ANOVA) followed by individual two-sample Tukey tests; *n* = 30 nuclei each examined over two independent experiments. All the boxplots in this work show median (thick black bar) and first and third quartiles, with lower and upper whiskers extending to 1.5-times the interquartile range of the first and third quartiles or the highest and lowest values, respectively. **c**, *H2B.8* transcription levels in indicated tissues and cells. TPM, transcripts per million. **d**, Confocal images of *pH2B.8::H2B.8–eGFP* pollen, in which the eGFP signal is specific to the sperm nuclei (SN). VN, vegetative nucleus (outlined in a dashed line). Data shown represent three independent experiments. Scale bars, 1 μm (**a**) 5 μm (**d**).

condensation and reveal a new mechanism of transcription-compatible chromatin condensation.

## Aggregation of sperm chromatin

Previous studies of *Arabidopsis* using DAPI staining showed that sperm chromatin is highly condensed and had small nuclear size compared to somatic and vegetative cells[19]. To characterize sperm chromatin in more detail, we examined *Arabidopsis* sperm nuclei using super-resolution 3D structured illumination microscopy (3D-SIM). Distinct chromatin aggregates were observed throughout the nucleoplasm in sperm cells (Fig. 1a). By contrast, chromatin in vegetative and leaf cells was more homogenous, although leaf cells had condensed heterochromatin foci at the nuclear periphery (Fig. 1a).

To understand the composition of chromatin within sperm aggregates, we performed immunostaining for histone H3 lysine 9 dimethylation (H3K9me2), a modification associated with silenced heterochromatin[21]. Larger aggregates situated at the sperm nuclear periphery colocalized with H3K9me2 signals (Fig. 1a). This result shows that heterochromatin domains persist in the sperm, as previously reported[16]. However, these heterochromatin foci appeared more enlarged in sperm

cells than in leaf cells (Fig. 1a). To further examine this aspect, we quantified the total volumes of H3K9me2-enriched heterochromatin foci in sperm, in diploid leaf nuclei and in the nuclei of generative cells (which are haploid mother cells that divide into sperm cells). As expected, the volume of heterochromatin foci was reduced by half in haploid generative nuclei compared to diploid leaf nuclei (Fig. 1b). Conversely, sperm heterochromatin foci were 27.5% larger than those of generative cell nuclei (Fig. 1b), which suggested that there was a reduced level of heterochromatin condensation or an increased proportion of the genome incorporated into heterochromatin. The latter scenario is less likely, as less H3K9me2 (compared with total H3) was detected in sperm cells than in leaf cells (Extended Data Fig. 1a). Apart from heterochromatin foci, other chromatin aggregates in sperm were depleted of H3K9me2 (Fig. 1a). This observation indicates that a new mechanism is used to compact the less heterochromatic part of sperm chromatin.

## H2B.8 marks sperm chromatin

To investigate the mechanism of sperm chromatin compaction, we searched for sperm-specific chromatin factors. To that end, we performed mass spectrometry on leaf nuclei and on sperm and vegetative nuclei isolated by fluorescence-activated cell sorting (FACS). We identified a variant of histone H2B, H2B.8 (encoded by *AT1G08170*), which constituted 12.6% of sperm H2B but was absent in nuclei from vegetative and leaf cells (Extended Data Fig. 1b). Consistent with this result, RNA sequencing (RNA-seq) experiments detected abundant *H2B.8* transcript levels in sperm cells, but none from somatic tissues such as leaves, roots and whole seedlings (Fig. 1c). To further examine the protein expression pattern during development, we generated reporter lines by expressing a H2B.8–eGFP fusion protein with the native *H2B.8* promoter in *Arabidopsis* (*pH2B.8::H2B.8–eGFP*). Confocal imaging showed that H2B.8 is incorporated into sperm following the second pollen mitotic division step, when nuclei are compacted and chromatin aggregates (Fig. 1d and Extended Data Fig. 1c). H2B.8 was rapidly lost after fertilization, but reappeared during seed maturation (Extended Data Fig. 1d). No H2B.8–eGFP was observed in any other cell or tissue except sperm and seeds (Extended Data Fig. 1c,d), which is consistent with recently published analyses of H2B expression[22].

## H2B.8 mediates chromatin condensation

Because the presence of H2B.8 correlated with nuclear and chromatin condensation, we proposed that H2B.8-induced chromatin condensation is responsible for sperm compaction. To test this hypothesis, we generated two independent *h2b.8* CRISPR knockout mutants (*h2b.8-1* and *h2b.8-2*) (Extended Data Fig. 2a) and examined their sperm phenotype by confocal and 3D-SIM imaging. Sperm nuclei of the two *h2b.8* mutants were about 40% larger than those of the wild type (Fig. 2a), which suggested that H2B.8 contributes to sperm nuclear compaction. Notably, sperm nuclei of *h2b.8* mutants were smaller than their progenitor, the generative nucleus, which is also haploid (Extended Data Fig. 2b). This result indicated that H2B.8 is not the sole mechanism involved in sperm nuclear compaction. In further support of our hypothesis, chromatin was more homogenous and had reduced aggregation in *h2b.8* mutants (Fig. 2b and Extended Data Fig. 2c–e). To confirm the role of *h2b.8* mutations in these phenotypes, we expressed *pH2B.8::H2B.8–Myc* and *pH2B.8::H2B.8–eGFP* transgenes in the *h2b.8* mutant (*h2b.8-1*; unless otherwise specified, all *h2b.8* mutants hereafter refer to this allele). Both transgenes successfully rescued the sizes of *h2b.8* sperm nuclei to wild-type levels (Fig. 2a), which provided evidence that H2B.8 drives nuclear condensation. Chromatin aggregates were also restored in *pH2B.8::H2B.8–eGFP h2b.8* sperm (Fig. 2c). Furthermore, these restored aggregates colocalized with H2B.8–eGFP (Fig. 2c and Extended Data Fig. 2f), which indicated that H2B.8 is directly involved in forming the aggregates.

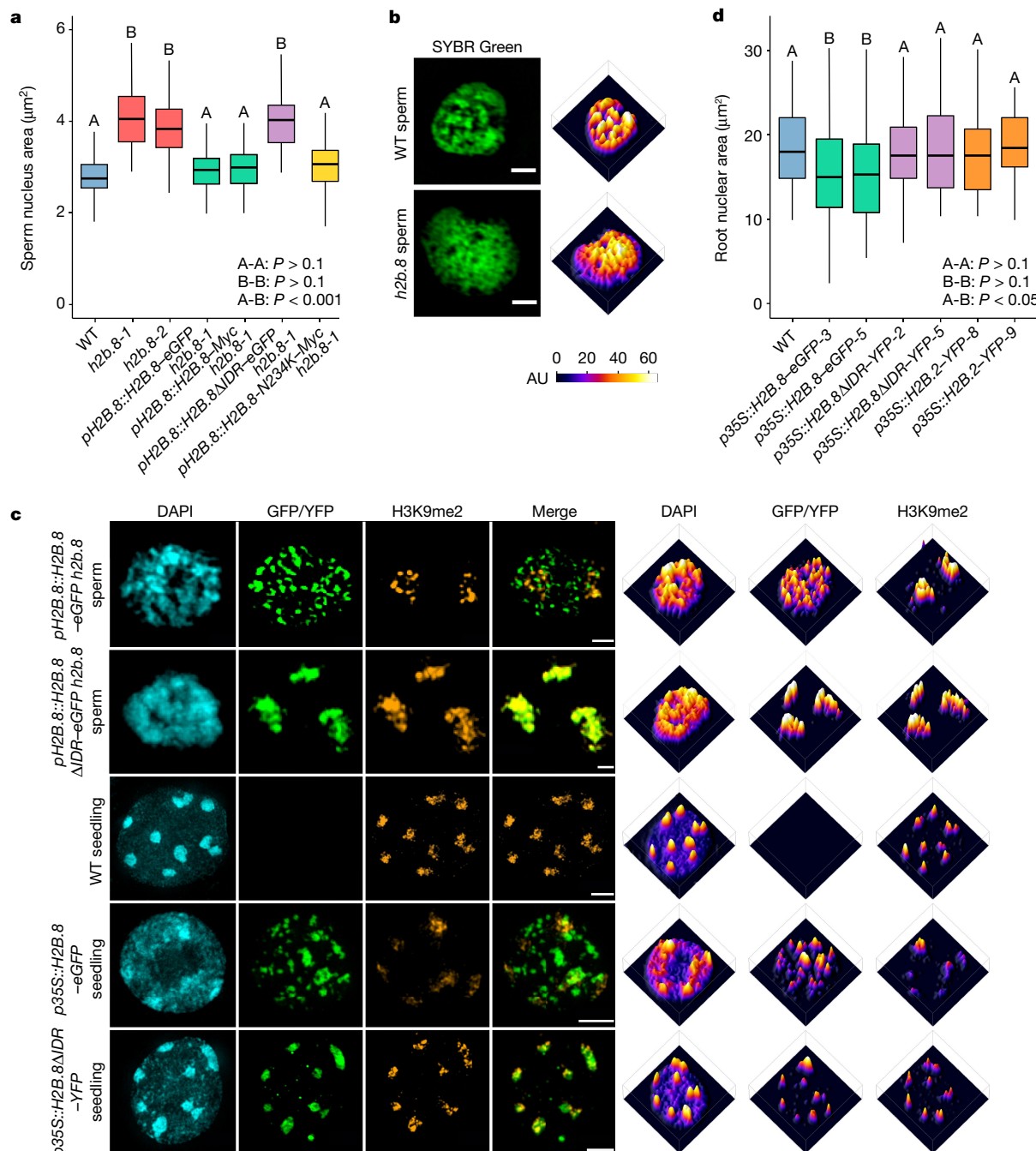

**Fig. 2 | H2B.8 is required and sufficient to drive chromatin and nuclear condensation. a**, Sperm nuclear sizes in WT and other indicated genotypes. *P* values calculated using one-sided ANOVA followed by individual two-sample Tukey tests. Boxplots marked as A and B are significantly different between groups (*P* < 0.001) but not within the group (*P* > 0.1). *n* = 80 (WT, *h2b.8-1*, *pH2B.8::H2B.8–Myc h2b.8-1*, *pH2B.8::H2B.8-N234K–Myc h2b.8-1*), 79 (*pH2B.8::H2B.8–eGFP h2b.8-1, h2b.8-2*) or 39 (*pH2B.8::H2B.8ΔIDR–eGFP h2b.8-1*) nuclei examined over two independent experiments. **b,c**, Super-resolution 3D-SIM images (left) and associated intensity profiles (right) of sperm and seedling nuclei of indicated genotypes (*h2b.8* refers to *h2b.8-1* unless specified

otherwise). Data shown represent three independent experiments. Scale bars, 1 μm (**b**), 1 μm (**c**, upper two panels) and 2 μm (**c**, lower three panels). **d**, Root nuclear sizes in indicated genotypes (numbers after the genotype indicate independent transgenic lines). *P* values calculated using one-sided ANOVA followed by individual two-sample Tukey tests. Boxplots marked as A and B are significantly different between groups (*P* < 0.05) but not within the group (*P* > 0.1). *n* = 100 (WT), 101 (*p35S::H2B.2–YFP-9*), 103 (*p35S::H2B.8–eGFP-3*, *p35S::H2B.8ΔIDR–YFP-5*), 104 (*p35S::H2B.8–eGFP-5*), 107 (*p35S::H2B.8ΔIDR–YFP-2*) or 109 (*p35S::H2B.2–YFP-8*) nuclei examined over two independent experiments.

To test whether H2B.8 is sufficient to drive chromatin aggregation and nuclear compaction, we ectopically expressed H2B.8 using a strong constitutive promoter (*p35S*). Distinctive chromatin aggregates that colocalized with H2B.8–eGFP were induced in *p35S::H2B.8–eGFP* somatic nuclei (Fig. 2c and Extended Data Fig. 2g). Approximately

30% of nuclei contained numerous small H2B.8 aggregates; the remaining 70% had fewer and larger aggregates (Fig. 2c and Extended Data Fig. 2h). The total volume of H2B.8 aggregates was comparable between the two types of nuclei (Extended Data Fig. 2h), which suggested that H2B.8-containing chromatin aggregates were fusing over

time. *p35S::H2B.8–eGFP* expression also reduced the nuclear size in root cells by 22% (Fig. 2d). These results demonstrate that H2B.8 is sufficient for chromatin and nuclear condensation.

## H2B.8 drives chromatin phase separation

H2B.8 is distinguished from other *Arabidopsis* H2B variants by a much longer amino-terminal tail that contains a 93 amino acid IDR (Extended Data Fig. 3a,b). Phylogenetic analysis revealed that H2B.8 is specific to flowering plants and is present in all flowering plant species with published genomes, except the most basal *Amborella trichopoda* (Fig. 3a and Supplementary Table 1). Notably, all identified H2B.8 homologues shared the insertion of an IDR in the histone tail (Fig. 3a and Supplementary Table 1), which indicated its functional importance.

IDRs can drive the formation of biomolecular condensates through phase separation[23,24]. Moreover, phase separation of IDR-containing proteins can mediate the formation and condensation of heterochromatin foci[25,26]. On the basis of this knowledge and the distinctive H2B.8 foci pattern in sperm and somatic cells that ectopically express H2B.8 (Fig. 2c), we proposed that H2B.8 aggregates chromatin through IDR-mediated phase separation.

To test this hypothesis, we assembled nucleosomal arrays using recombinant fluorophore-labelled histone octamers and a DNA template containing 12 repeats of the 601 nucleosome positioning sequence. We then tested the phase-separation properties of these nucleosomal arrays (Fig. 3b and Extended Data Fig. 3c). The addition of cation (K⁺ and/or Mg²⁺) at physiologically relevant concentrations to chromatin reconstituted with either H2B.8 or a canonical H2B (H2B.2) that naturally lacks an IDR induced the formation of phase-separated droplets (Fig. 3b). This result is in line with a previous finding that chromatin undergoes phase separation in vitro at physiological salt conditions[27,28]. Consistent with the reported liquid-like property of chromatin condensates[27], chromatin droplets that contained H2B.8 or H2B.2 showed fluorescence recovery after photobleaching (FRAP; Extended Data Fig. 3d) and droplet fusion following contact (Extended Data Fig. 3e).

FRAP of H2B.8 chromatin droplets was slower than that of H2B.2 droplets (Extended Data Fig. 3d), which reflected the reduced internal droplet dynamics and a more gel-like behaviour. To further test whether H2B.8 confers different phase-separation properties to chromatin, we examined H2B.8-containing and H2B.2-containing nucleosomal arrays under different salt and array concentrations. Under the same physiological salt concentration, H2B.8-containing nucleosome arrays formed phase-separated condensates at lower chromatin concentrations (Extended Data Fig. 3f). Furthermore, unlike typical chromatin phase separation that requires the assistance of physiological cations[27,28], H2B.8-containing nucleosome arrays formed phase-separated droplets under low or no cation conditions (or in the presence of high concentration of the chelating agent EDTA; Fig. 3b and Extended Data Fig. 3g). To test whether this phase-separation property relies on the IDR of H2B.8, we reconstituted nucleosomal arrays using H2B.8 without the IDR (H2B.8ΔIDR) (Extended Data Fig. 3c). Similar to H2B.2-containing chromatin, H2B.8ΔIDR-containing nucleosome arrays did not phase separate at lower array concentrations (<50 nM; Extended Data Fig. 3f) and failed to undergo phase separation in the absence of salt (Fig. 3b and Extended Data Fig. 3g). Consistent with the idea that IDRs promote phase separation through the disordered state instead of specific sequence motifs[23,29], chromatin that contained H2B.8 in which the IDR sequence was randomly scrambled (H2B.8-scrambledIDR) phase separated in a salt-independent manner (Fig. 3b). Taken together, our results demonstrate that the IDR of H2B.8 mediates a new form of chromatin phase separation.

To test whether H2B.8 phase separation is required for chromatin condensation in vivo, we ectopically expressed H2B.8ΔIDR (*p35S::H2B.8 ΔIDR–YFP*) or the canonical H2B.2 in cells. In contrast to the effect of full-length H2B.8, H2B.8ΔIDR or H2B.2 expression did not induce chromatin aggregation and had no effect on nuclear size in root cells (Fig. 2c,d). This result confirmed that chromatin and nuclear condensation are dependent on the IDR of H2B.8. We next expressed H2B.8ΔIDR in *h2b.8* mutant plants (*pH2B.8::H2B.8ΔIDR–eGFP h2b.8*) and examined whether sperm nuclear condensation is also dependent on the IDR. Unlike the full-length H2B.8, H2B.8ΔIDR failed to rescue the sperm nuclear size phenotype of *h2b.8* (Fig. 2a,c). This result demonstrates the importance of the IDR for sperm nuclear condensation.

Beyond the IDR, H2B.8 has several amino acid differences to that of canonical H2Bs in the globular domain, including asparagine 234 (N234), which is canonically a lysine residue that is subject to monoubiquitylation[22] (Extended Data Fig. 3a). The inability of H2B.8ΔIDR to rescue the *h2b.8* sperm nuclear phenotype (Fig. 2a) shows that without the IDR, the H2B.8 globular domain cannot mediate nuclear compaction. Nonetheless, as H2B monoubiquitylation is an important modification[30] that would be precluded by N234, we examined whether N234 is important for H2B.8 function by expressing a mutated H2B.8 with the 234th asparagine replaced by lysine (H2B.8-N234K). The expression of H2B.8-N234K fully complemented the *h2b.8* sperm nuclear size phenotype (Fig. 2a), which shows that N234 is not essential for H2B.8 function.

The dependence of chromatin and nuclear condensation on the IDR of H2B.8 might result from the disordered state of the IDR and associated phase-separation ability or from IDR-mediated recruitment of unknown chromatin-condensing factors. To test these hypotheses, we expressed H2B.8 with a randomly scrambled IDR sequence (H2B.8-scrambledIDR) or with the native IDR replaced by animal IDR sequences of similar negative charge (H2B.8-EWSR-IDR and H2B.8-TAF15-IDR)[31] in tobacco leaves and measured their effects on nuclear size. Similar to the effect of native H2B.8, the expression of H2B.8-scrambledIDR, H2B.8-EWSR-IDR and H2B.8-TAF15-IDR all effectively condensed the nuclei of tobacco epidermis (Extended Data Fig. 3h). This result indicates that the function of H2B.8 relies on phase-separation ability instead of specific sequence motifs within the IDR. In line with this, although the presence of the IDR is conserved among flowering plants, the sequences of H2B.8 IDRs are diverse (Supplementary Table 1).

## H2B.8 is located in silent euchromatin

To further understand H2B.8 activity, we determined the genomic localization of H2B.8 by performing native chromatin immunoprecipitation assay with sequencing (ChIP-seq) on *pH2B.8::H2B.8–eGFP h2b.8* pollen using GFP-specific antibodies. This analysis identified H2B.8 peaks that occupied about 17% of the sperm genome. This value was comparable with our mass spectrometry results, which showed that approximately 13% of canonical H2B is replaced by H2B.8 in sperm (Extended Data Fig. 1b). H2B.8 was most enriched within so-called euchromatic transposable elements (TEs) (Fig. 4a,b), which are AT-rich and depleted of H3K9me2 and other heterochromatic marks[32]. Heterochromatic TEs, which are typically GC-rich and H3K9me2-rich[32], had comparatively little H2B.8 (Fig. 4a,b). H2B.8 was excluded from the bodies of transcribed genes compared with inactive genes and intergenic regions, and H2B.8 enrichment and gene transcription were anticorrelated (Fig. 4a–c). H2B.8 distribution along chromosomes followed that of euchromatic TEs, with H2B.8 most abundant at the edges of pericentromeric regions (Fig. 4d and Extended Data Fig. 4a).

To further understand the chromatin preference of H2B.8 and whether its localization pattern is intrinsically determined by H2B.8 or other sperm-specific components, we performed native ChIP-seq with seedlings of the ectopic H2B.8 expression line (*p35S::H2B.8-eGFP*). This revealed an analogous H2B.8 localization pattern to that in sperm, with enrichment in euchromatic TEs and intergenic regions (Extended Data Fig. 4b–d). Together with the ability of ectopic H2B.8 to condense somatic cell chromatin and nuclei (Fig. 2c,d), these results suggest that H2B.8 deposition does not rely on sperm-specific factors. Next,

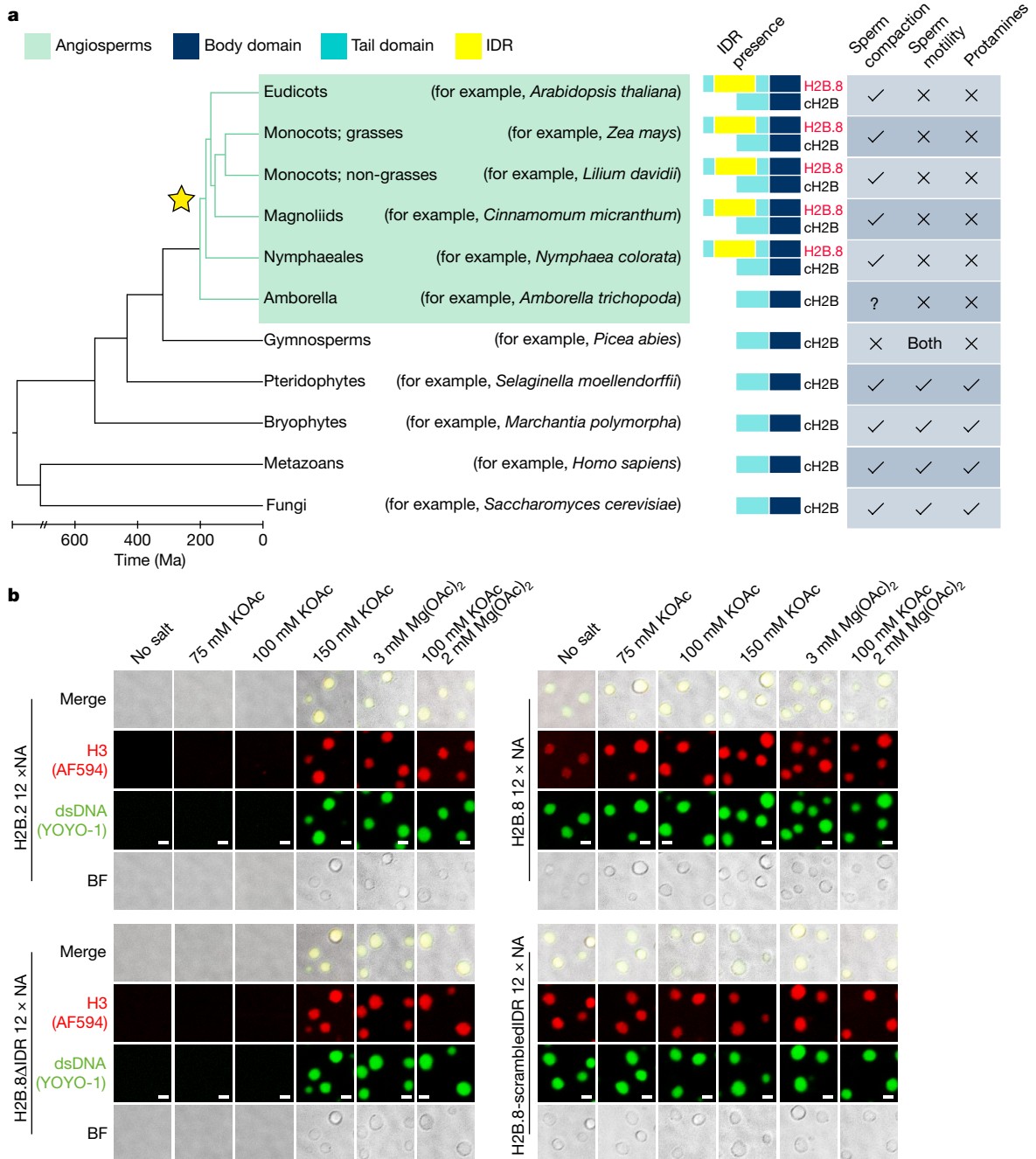

**Fig. 3 | H2B.8 condenses chromatin through IDR-dependent phase separation. a**, Phylogenetic tree illustrating H2B.8 evolution (marked by a star). Sperm chromatin compaction state, sperm motility and the presence of protamines or protamine-like proteins are denoted for represented eukaryote lineages. Ma, million years ago. **b**, In vitro phase-separation assays of nucleosomal arrays (NA) bearing H2B.2, H2B.8, H2B.8ΔIDR and H2B.8 with a scrambled IDR sequence (H2B.8-scrambledIDR) under indicated salt conditions and 100 nM NA concentration. Histone H3 was labelled by Alexa Fluor 594 (AF594, red), and double-stranded DNA (dsDNA) was stained with YOYO-1 (green). BF, bright field. Data shown represent one representative experiment, which has been performed three times with similar results. Scale bars, 2 μm.

utilizing available seedling epigenomic data, we explored H2B.8 associations with other chromatin features. Principal component analysis (PCA) revealed that H2B.8 clustered with neither permissive nor repressive chromatin modifications but associated with GC content (Fig. 4e). Multivariate linear regression modelling of H2B.8 further showed that transcription and GC content were the best predictors of H2B.8 localization, with which strong anticorrelations exist (Fig. 4c,f, Extended Data Fig. 4e,f and Supplementary Table 2). This may explain why H2B.8 is strongly depleted from transcribed genes (Fig. 4b and Extended Data Fig. 4c). In the remainder of the genome, in which there is little transcription, H2B.8 accumulated at GC-poor elements, mostly euchromatic TEs and intergenic regions (Extended Data Fig. 4e,g). In summary, our results suggest that H2B.8 localization is mostly driven by transcription and GC content rather than sperm-specific factors.

It is yet unclear why H2B.8 localization is associated with transcription and GC content; however, our cytological observations provided indications. 3D-SIM showed that unlike the full-length H2B.8, which is largely devoid in heterochromatin, H2B.8ΔIDR preferentially located to heterochromatin when expressed in sperm or seedling cells (Fig. 2c). To validate this observation and to examine the genomic localization

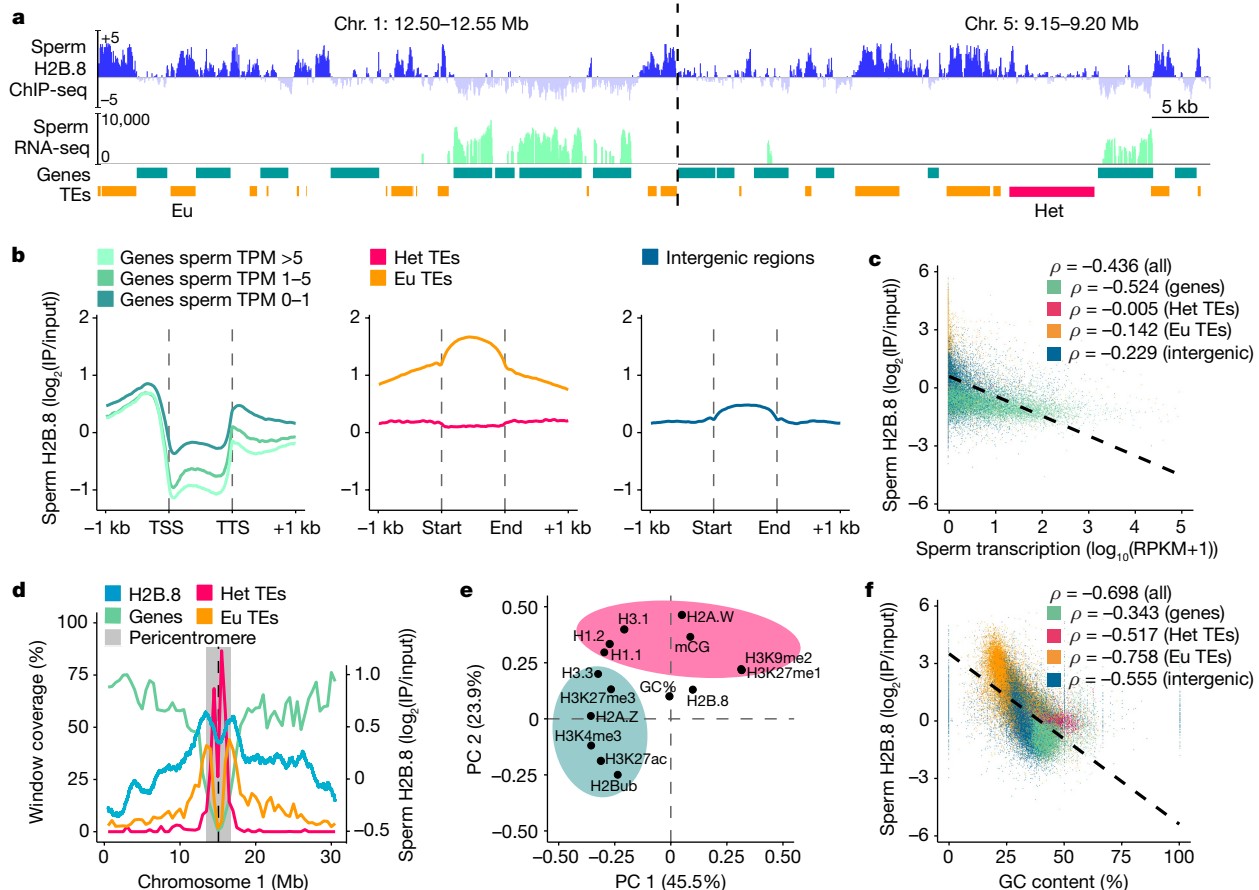

**Fig. 4 | H2B.8 is localized in AT-rich TEs and transcriptionally inactive intergenic regions. a**, Genome snapshots of H2B.8 abundance in sperm (log$_2$(IP/input)), sperm cell transcription (log$_2$(RPKM)) and gene and TE annotations (orange, euchromatic TE; magenta, heterochromatic TE) at representative regions. RPKM, reads per kilobase of transcript per million mapped reads. **b**, Profiles of sperm H2B.8 enrichment over genes (grouped by sperm cell expression), TEs (grouped by chromatin state) and intergenic regions. **c**,**f**, Scatterplots showing anticorrelation of sperm H2B.8 enrichment with sperm transcription (**c**) or GC content (**f**) among indicated genomic features. $\rho$, Spearman's rank. Colours are as **d**. **d**, Coverage of genes, euchromatic TEs and heterochromatic TEs (left $y$ axis, 500-kb windows) and H2B.8 enrichment in sperm (right $y$ axis; 1-kb windows) along chromosome 1. Chromosomes 2–5 are shown in Extended Data Fig. 4a. **e**, PCA of H2B.8 abundance in *p35S::H2B.8–eGFP* seedlings with other chromatin marks. The green and pink shaded areas represent euchromatic and heterochromatic marks, respectively.

of H2B.8ΔIDR, we performed ChIP-seq on H2B.8ΔIDR ectopically expressed (*p35S::H2B.8ΔIDR–eGFP*) seedlings and compared results with *p35S::H2B.8–eGFP* (and *p35S::H2B.2-eGFP* as a control) seedlings. Consistent with the cytology results, H2B.8ΔIDR preferentially located to pericentromeric heterochromatin, and its abundance along chromosomal arms was generally reduced compared with H2B.8 (Extended Data Fig. 5a). Notably, although the deletion of IDR markedly reduced the enrichment of H2B.8 in euchromatic TEs and increased H2B.8 deposition in heterochromatic TEs, it did not disrupt the preferential depletion of H2B.8 from transcribed genes (Extended Data Fig. 5b). This finding indicates that the IDR of H2B.8 is required for the preferential localization of H2B.8 in euchromatin, but not for its exclusion from transcribed regions.

## H2B.8 does not suppress transcription

Chromatin condensation is frequently associated with transcriptional repression[21]. Therefore, the localization of H2B.8 in the non-transcribing parts of the genome could arise through two mechanisms. Either H2B.8 suppresses transcription or it is excluded from transcribed regions. To test these hypotheses, we isolated wild-type and *h2b.8* mutant sperm cells and performed RNA-seq. Among the 12,198 genes expressed in either wild-type or *h2b.8* sperm, none had significantly altered expression in *h2b.8* cells (Fig. 5a). Similarly, we did not find any TEs that were

significantly misregulated in *h2b.8* sperm (Extended Data Fig. 6a). RNA-seq of wild-type and *p35S::H2B.8–eGFP* seedlings provided further support of the negligible effect of H2B.8 on transcription (Extended Data Fig. 6b,c). Therefore, our results indicate that unlike protamines, which condense animal sperm at the expense of transcriptional potential[8], H2B.8 condenses plant sperm without suppressing transcription.

To directly test whether H2B.8 inhibits transcriptional activation, we exposed *p35S::H2B.8–eGFP* and wild-type seedlings to drought stress and examined their transcriptional response. Comparable transcriptomic changes between *p35S::H2B.8–eGFP* and wild-type seedlings following drought treatment were observed (Fig. 5b and Extended Data Fig. 6d). Moreover, ChIP-seq of drought-treated and control *p35S::H2B.8–eGFP* seedlings showed that H2B.8 is evicted from genes induced by drought, such as *Highly ABA-Induced 1 (HAI1)* and *HAI2* (ref. [33]), whereas genes repressed by drought gain H2B.8 (Fig. 5c,d). These results demonstrate that H2B.8 does not inhibit gene transcription but is instead removed by transcription. This process may underlie the localization of H2B.8 in unexpressed euchromatin.

## H2B.8 promotes male fertility

To investigate the biological significance of H2B.8-mediated sperm condensation, we performed reciprocal crosses between the *h2b.8* heterozygous mutant and wild type. When the *h2b.8* heterozygous mutant

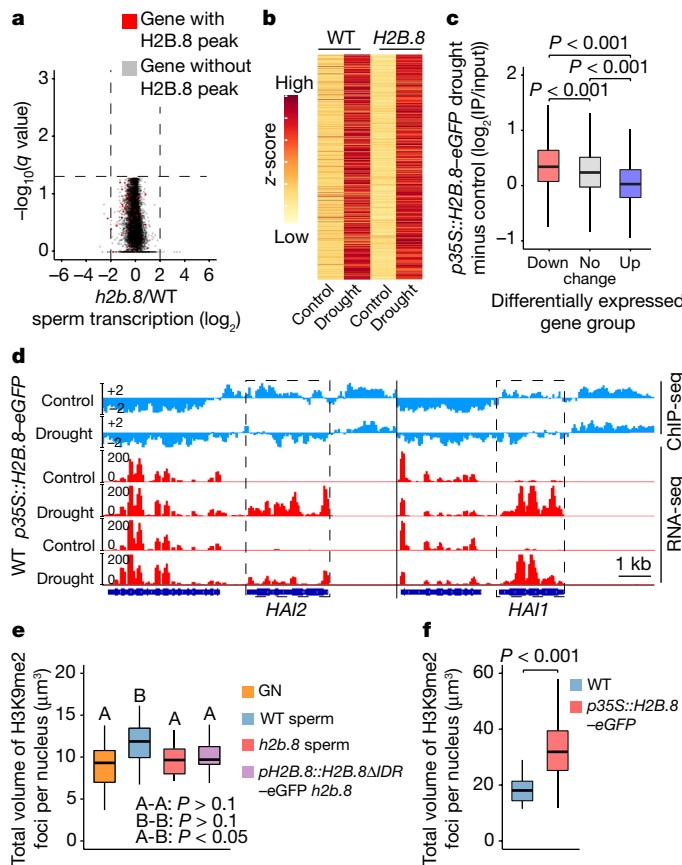

**Fig. 5 | H2B.8-induced chromatin aggregation does not suppress transcription but decondenses heterochromatin foci. a**, Volcano plot showing differential gene expression between *h2b.8* mutant and WT sperm cells. *n* = 12,198 for genes with TPM > 1 in either WT or *h2b.8* mutant. **b**, Heatmap depicting the transcription (*z*-score) of upregulated genes in response to drought stress in WT and *p35S::H2B.8–eGFP* seedlings subject to mock (control) and drought treatment. *n* = 1,605. **c**, Differences in H2B.8 abundance between drought and control conditions over differentially expressed gene groups in *p35S::H2B.8–eGFP* seedlings. *P* values calculated using one-sided ANOVA followed by individual two-sample Tukey tests. *n* = 711 (Down), 26,180 (No change) and 1,605 (Up) genes. **d**, Genome snapshots of example drought-response genes showing H2B.8 abundance (log₂(IP/input)) and transcription (log₂(RPKM)) in seedlings of indicated genotypes under control and drought conditions. **e**, Quantification of H3K9me2-enriched heterochromatin foci in the generative nucleus and sperm nuclei of indicated genotypes. *P* values calculated using one-sided ANOVA followed by individual two-sample Tukey tests. Boxplots marked as A and B are significantly different between groups (*P* < 0.05) but not within the group (*P* > 0.1). *n* = 30 nuclei each from two independent experiments. **f**, Quantification of H3K9me2-enriched heterochromatin foci in WT and *p35S::H2B.8–eGFP* seedling nuclei. *P* value calculated using independent two-sample *t*-test. *n* = 138 and 101 nuclei for WT and *p35S::H2B.8–eGFP*, respectively, examined from two independent experiments.

was used as the male, F₁ progeny were around 30% less likely to carry the *h2b.8* allele than the wild-type allele (*P* < 0.01, Fisher's exact test; Supplementary Table 3). By contrast, the transmission of *h2b.8* was not significantly different from the wild-type allele when passed through the female (*P* = 0.64, Fisher's exact test; Supplementary Table 3), which demonstrated that H2B.8 is important for male fertility. The fertility defect of *h2b.8* is not caused by disrupted pollen germination, as *h2b.8* pollen grains germinated at comparable rates to wild type in vitro (Supplementary Table 4). This result is consistent with the null effect of H2B.8 on transcription (Fig. 5a,b and Extended Data Fig. 6a–d), and suggests that the fertility defect is most probably caused by enlarged sperm nuclei.

Manual crossing is a stressful process, with pistils (the female organs) slightly desiccated and pollinated at an earlier developmental stage than normal. To investigate whether *h2b.8* affects male fertility in a less stressful mating environment, we examined the segregation ratio of progeny generated from self-pollinated *h2b.8* heterozygous plants. We observed the expected Mendelian segregation ratio (*N* = 1,462; Supplementary Table 5), which shows that the *h2b.8* mutation does not affect fertility when plants are allowed to self-fertilize under laboratory conditions. As H2B.8 is transiently expressed in mature seeds (Extended Data Fig. 1d), this result also demonstrated that *h2b.8* does not affect seed development under laboratory conditions. Collectively, our observations suggest that H2B.8 is important for sperm fertility in challenging or stressful situations, such as those created by manual crossing. This finding might be relevant to reproduction under natural environmental conditions, which are usually less favourable than standard laboratory conditions.

### H2B.8 decondenses heterochromatin

In addition to the overall chromatin condensation in sperm observed by 3D-SIM, there was slight decondensation of heterochromatin foci (Fig. 1a,b). To understand whether this is caused by H2B.8, we performed immunostaining using H3K9me2-specific antibodies and measured the volume of H3K9me2 foci in *h2b.8* mutant nuclei. The *h2b.8* mutation significantly reduced the volume of heterochromatin foci in sperm to a level resembling that in the generative nucleus (Fig. 5e). Furthermore, H3K9me2 heterochromatin foci were more enlarged and decondensed in *p35S::H2B.8–eGFP* seedling nuclei than in wild type nuclei (Fig. 5f and Extended Data Fig. 6e). Most wild-type nuclei showed highly condensed heterochromatin foci, but these were rarely found in *p35S::H2B.8–eGFP* seedling nuclei (Extended Data Fig. 6e). The majority of *p35S::H2B.8–eGFP* seedling nuclei exhibited moderately dispersed heterochromatin foci (Fig. 5f and Extended Data Fig. 6e), reminiscent of those in sperm (Fig. 1a,b). This result suggests that H2B.8 causes heterochromatin foci to decondense.

Our data also suggested that decondensation of heterochromatin foci depends on H2B.8 phase separation, as the expression of H2B.8ΔIDR did not affect heterochromatin (Fig. 5e). As heterochromatin foci are also phase-separated condensates[25,26,34], this suggested that there are interactions between the two types of condensates. Indeed, although H2B.8 and heterochromatic condensates were mostly distinct, some physical associations were observed (Fig. 2c and Extended Data Fig. 6e). Collectively, our results suggest that condensation of chromatin through H2B.8 phase separation affects heterochromatin condensation, probably because H2B.8-associated AT-rich euchromatic TEs are interspersed with heterochromatic TEs in pericentromeric regions (Fig. 4d and Extended Data Figs. 4a and 6f,g). Alternative hypotheses that are independent of H2B.8 phase separation are also plausible; for example, H2B.8 might directly recruit factors that interfere with heterochromatin condensation.

### H2B.8 increases chromosomal arm contacts

To understand how H2B.8 mediates chromatin condensation, we performed genome-wide chromosome conformation capture analysis[35] (Hi-C) on seedlings that ectopically express H2B.8 (*p35S::H2B.8–eGFP*) and wild-type controls. The Hi-C libraries were sequenced to single kilobase resolution (Extended Data Fig. 7a), and our wild-type contact matrices were comparable to previously published experiments[36,37] (Extended Data Fig. 7b). As previously shown[37–40], topologically associating domains were absent in *Arabidopsis*, but telomeres were frequently associated, as were centromeres (Extended Data Fig. 7c).

The comparison of *p35S::H2B.8–eGFP* and wild-type Hi-C data revealed alterations in higher-order chromatin architecture. Within chromosomes, ectopic H2B.8 caused increased short-range contacts

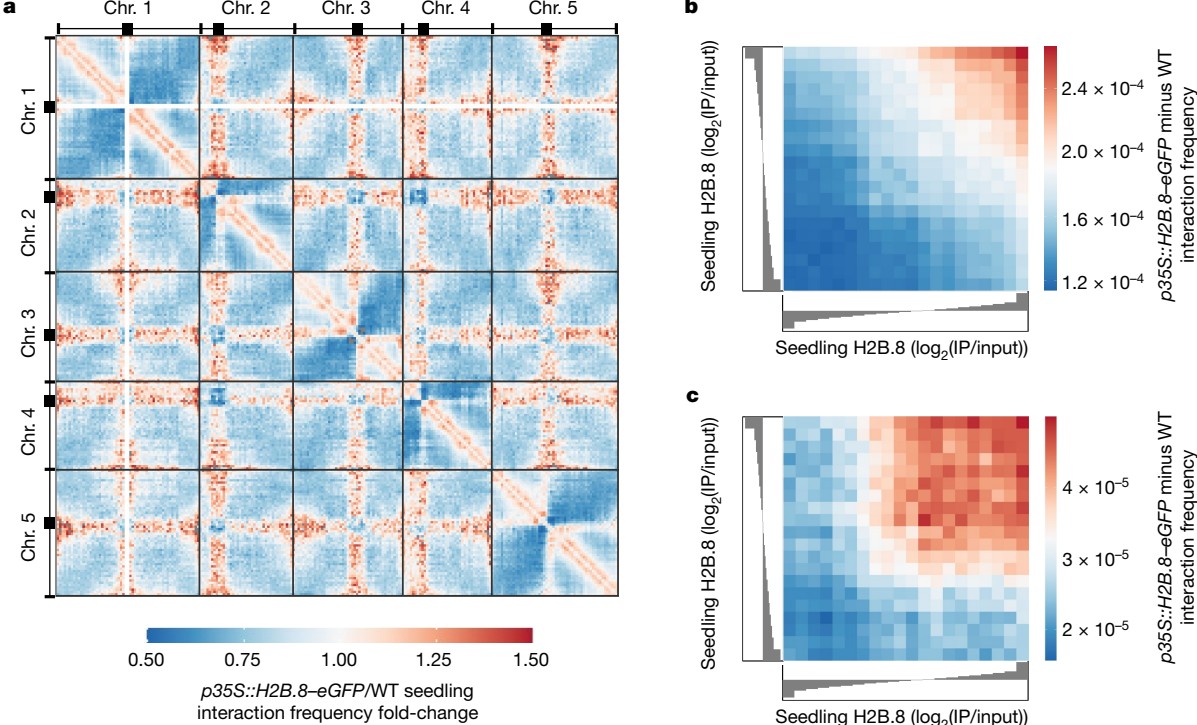

**Fig. 6 | Increased chromosomal interactions between H2B.8-containing loci. a**, Genome-wide interaction frequency fold-change heatmap between WT and *p35S::H2B.8–eGFP* seedlings at 500-kb resolution. **b**, Short-range intrachromosomal interaction frequency difference between *p35S::H2B.8– eGFP* and WT over quantiles of seedling H2B.8 enrichment ($\log_2$(IP/input)).

Spearman's $\rho$ = 0.974. **c**, Long-range interaction frequency difference between *p35S::H2B.8–eGFP* and WT between chromosome arms and pericentromeric regions over quantiles of seedling H2B.8 enrichment ($\log_2$(IP/input)). Spearman's $\rho$ = 0.890.

(200 kb–1.1 Mb) (Fig. 6a and Extended Data Fig. 7d). Short-range intra-chromosomal interactions are indicative of chromatin condensation, which suggested that H2B.8 principally forms aggregates by concentrating linearly proximal regions (Fig. 6a and Extended Data Fig. 7d). In support of this idea, the increase in short-range contacts strongly correlated with local H2B.8 abundance (Spearman's $\rho$ = 0.97) (Fig. 6b and Extended Data Fig. 7e).

Contacts between pericentromeric regions (heterochromatin) and distal chromosomal arms (euchromatin) were also increased in *p35S::H2B.8–eGFP* seedlings (Fig. 6a and Extended Data Fig. 7d). Concomitantly, interchromosomal interactions between pericentromeres were reduced (Fig. 6a and Extended Data Fig. 7d). These alterations are consistent with the cytologically observed heterochromatin decondensation and association of heterochromatin foci and H2B.8 condensates (Fig. 2c and Extended Data Fig. 6e). The effect of H2B.8 was local, as the interactions between pericentromeric regions and chromosomal arms increased at regions with abundant H2B.8 (Fig. 6c and Extended Data Fig. 7f). These observations support the hypothesis that dispersal of heterochromatin foci is caused by H2B.8-mediated aggregation of euchromatic TEs that are abundant in and near pericentromeric regions (Fig. 4d and Extended Data Figs. 4a and 6g). Taken together, our Hi-C and ChIP-seq data demonstrate that H2B.8 achieves a form of global chromatin condensation through the binding and aggregation of transcriptionally inactive AT-rich sequences dispersed throughout the genome.

## Discussion

Our results revealed a mechanism of chromatin condensation driven by H2B.8-induced phase separation (Extended Data Fig. 8). Chromatin aggregates in the sperm nucleus were reduced in *h2b.8* mutants (Fig. 2b and Extended Data Fig. 2c), whereas ectopically expressed

H2B.8 in somatic cells was sufficient to induce chromatin aggregates in an IDR-dependent manner (Fig. 2c). Hi-C and cytological observations revealed that H2B.8 forms chromatin condensates by increasing interactions between H2B.8-enriched chromosomal regions (Figs. 2c and 6b,c and Extended Data Fig. 2h). Owing to H2B.8 deposition within AT-rich sequences in both pericentromeric regions and chromosomal arms (Fig. 4d and Extended Data Figs. 4a and 6g), broad chromosomal regions were concentrated by phase separation. In interphase somatic cells, euchromatin takes up most of the nuclear volume[41]. Because H2B.8 is abundant in euchromatin, H2B.8-induced chromatin condensation is highly effective at condensing nuclei (Fig. 2a,d).

Despite its effectiveness, H2B.8 compacts nuclei without compromising transcription (Fig. 5a,b). For species with swimming sperm, in which the size of the sperm head is crucial[1,2,9], DNA condensation may be paramount and transcription dispensable. Protamines can greatly condense DNA and have been identified in the sperm of most all multicellular eukaryotic lineages except angiosperms (flowering plants) and gymnosperms (Fig. 3a). This includes bryophytes and pteridophytes, both of which have motile sperm and use protamines for sperm condensation[10–14,19] (Fig. 3a). By contrast, H2B.8 is specific to flowering plants. Flowering plants have immotile sperm and may benefit from a less extreme approach that condenses nuclei without limiting transcription. This moderate level of condensation is still important for fertility, as *h2b.8* mutations can significantly reduce male transmission (Supplementary Table 3). As sperm transcription (Fig. 5a and Extended Data Fig. 6a) and in vitro pollen germination (Supplementary Table 4) are not affected by the *h2b.8* mutation, the reduced fertility is probably caused by the enlarged sperm nuclei. We speculate that fertilization, which takes place at ovules deeply embedded in angiosperm maternal tissues, might favour smaller sperm nuclei. Consistent with this idea, gymnosperms, which have exposed ovules, produce sperm with uncondensed nuclei[11] and lack H2B.8 (Fig. 3a). Therefore, H2B.8 is an

evolutionary innovation of angiosperms that achieves a moderate level of chromatin condensation that is compatible with active transcription.

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

# Methods

## Plant growth conditions

*A. thaliana* plants (Col-0 ecotype) used in this study were grown under long day (16 h light, 8 h dark) conditions at 22 °C and 70% humidity. Seedlings were grown on germination medium plates without glucose under the same conditions.

## Generation of CRISPR–Cas9 mutants

Mutant alleles of *H2B.8* were generated using the CRISPR–Cas9 technique[42]. Four single guide RNAs (sgRNAs; Supplementary Table 6) were designed using CHOPCHOP (v.3)[43] and cloned using the Golden Gate system[42]. Constructs were transformed into the *Agrobacterium tumefaciens* strain GV3101 using floral dip to wild-type (WT) Col-0 *A. thaliana*[44]. Transformants were screened by Sanger sequencing. Selected lines were taken to the next generation to produce homozygous mutants without Cas9. Line *h2b.8-1* was genotyped by dCAPS with EcoNI, whereas line *h2b.8-2* was genotyped by PCR (Supplementary Table 6).

## Vector cloning

For the H2B.8 reporter constructs (*pH2B.8::H2B.8–eGFP* and *pH2B.8::H2B.8–Myc*), approximately 2 kb upstream of *H2B.8* was cloned as the promoter and the *H2B.8* gDNA sequence was amplified (Supplementary Table 6). Using MultiSite Gateway Technology (Thermo Fisher Scientific), the PCR products were ligated to P4P1r and pDONR207, respectively. Sequences were assembled to the expression vector pK7m34GW with a carboxy-terminal eGFP or 3× Myc tag in P2rP3. The ectopic H2B.8 expression vector (*p35S::H2B.8–eGFP*) was generated in the same way, but using the *35S* promoter.

The native and ectopic H2B.8ΔIDR expression constructs (*pH2B.8::H2B.8ΔIDR–YFP* and *p35S::H2B.8ΔIDR–YFP*, respectively) were generated by overlapping PCR to remove the IDR sequence, whereas ectopic H2B.2 (*p35S::H2B.2–YFP*) was cloned from gDNA (Supplementary Table 6). Products were ligated to the pCAMBIA1300 vector backbone containing the *35S* promoter and a C-terminal eGFP/YFP using the In-Fusion cloning system (Takara Bio).

The *pH2B.8::H2B.8-N234K–Myc* construct was generated by overlapping PCR (Supplementary Table 6) and ligated to pDONR207 and subsequently conjugated to the *H2B.8* promoter and a C-terminal 3× Myc tag in the pK7m34GW expression vector using MultiSite Gateway Technology (Thermo Fisher Scientific).

For the *p35S::H2B.8-scrambledIDR–eGFP*, *p35S::H2B.8-EWSR1-IDR–eGFP* and *p35S::H2B.8-TAF15-IDR–eGFP* constructs, the scrambledIDR (randomly shuffled amino acid sequence of the IDR by Python 3.9: DEVIQDISANPPVLENEPVTPSEPTVQEDTRECIETPEETPISVPEGEATPET-KVQGDNSDFSSQTRTVDLKEVPSVPPREGTPPTPVVDDVE); EWSR1-IDR[31] (ASTDYSTYSQAAAQQGYSAYTAQPTQGYAQTTQAYGQQSYGTYGQPTDV SYTQAQTTATYGQTAYATSYGQPPTGYTTPTAPQAYSQPVQGYGTGAYDTT TATVTTTQASYAAQSAYGTQPAYPAYGQQPAATAPTRPQDGNKPTETSQ PQSSTGGYNQPSLGYGQSNYSYPQVPGSYPMQPVTAPPSYPPTSYSSTQPT SYDQSSYSQQNTYGQPSSYGQQSSYGQQSSYGQQPPTSYPPQTGSYSQA APSQYSQQSSSYGQQSSFRQDHPSSMGVYGQ), and TAF15-IDR[31] (SDSG SYGQSGGEQQSYSTYGNPGSQGYGQASQSYSGYGQTTDSSYGQNYSGYSS YGQSQSGYSQSYGGYENQKQSSYSQQPYNNQGQQQNMESSGSQGGRAP SYDQPDYGQQDSYDQQSGYDQHQGSYDEQSNYDQQHDSYSQNQQSY HSQ) sequences, respectively, were optimized by *Arabidopsis* codon usage and synthesized. The synthesized fragments were then fused with *H2B.8* by overlapping PCR (Supplementary Table 6). Inserts were cloned to the pCAMBIA1300 vector containing the *35S* promoter and C-terminal eGFP.

The above-mentioned vectors were transformed into *h2b.8-1* mutant plants of $T_3$ generation or WT plants. Single insertion transgenic lines in $T_3$ or $T_4$ generations were used in this study, and at least two independent transgenic lines were included in each assay.

## Sperm, vegetative and leaf nuclei total protein extraction

Sperm and vegetative nuclei were isolated by FACS (two replicates were isolated for each cell type) using a BD FACSMelody cell sorter (BD Biosciences), as previously described[45] (Supplementary Fig. 2a). A total of 2 million sorted nuclei (isolated from roughly 40 ml open flowers) were pooled for each replicate, and 0.45 volume of 3.2× lysis buffer (10% SDS, 100 mM TEAB, pH 7.55) was added. Nuclei were lysed at 95 °C for 5 min, then centrifuged at 13,000*g* for 8 min at room temperature. The lysate was placed into a new tube. One-tenth volume of 12% phosphoric acid was added and mixed by pipetting. Then, six times the volume of S-Trap buffer (90% aqueous MeOH, 100 mM TEAB, pH 7.1) was added and mixed by pipetting. Protein was loaded onto a S-Trap Micro column (Protifi) by centrifugation at 4,000*g* for 30 s. The column was washed three times with 150 µl S-Trap buffer. Protein was digested on the column with 4 µg trypsin in 50 mM TEAB at 47 °C for 1 h. Peptides were eluted sequentially by centrifugation (4,000*g* for 30 s) with 40 µl 50 mM TEAB, 40 µl 0.2% formic acid and 35 µl 50% acetonitrile and 0.2% formic acid.

## Liquid chromatography–mass spectrometry

The eluted peptide solutions were dried down and the peptides dissolved in 0.1% trifluoroacetic acid and 3% acetonitrile for liquid chromatography with tandem mass spectrometry (LC–MS/MS). Sperm and vegetative nuclei samples were analysed by nanoLC–MS/MS on an Orbitrap Fusion Tribrid mass spectrometer coupled to an UltiMate 3000 RSLCnano LC system (Thermo Fisher Scientific). The samples were loaded and trapped using a pre-column with 0.1% trifluoroacetic acid at 20 µl min$^{-1}$ for 3 min. The trap column was then switched in-line with the analytical column (nanoEase M/Z column, HSS C18 T3, 100 Å, 1.8 µm, Waters) for separation using the following long gradient of solvent A (water, 0.05% formic acid) and solvent B (80% acetonitrile, 0.05% formic acid) at a flow rate of 0.3 µl min$^{-1}$: 0–3 min 3% B (trap only); 3–14 min linear increase of B to 13%; 14–113 min increase of B to 39%; 113–123 min increase of B to 55%; followed by a ramp to 99% B and re-equilibration to 3% B.

Data were acquired with the following mass spectrometer settings in positive ion mode. MS1/OT: resolution of 120 K, profile mode, mass range *m/z* of 300–1,800, automatic grain control of 4e$^5$ and fill time of 50 ms. MS2/IT, data-dependent analysis was performed using higher-energy collision dissociation fragmentation with the following parameters: top30 in IT rapid, centroid mode, isolation window of 1.6 Da, charge states of 2–5, threshold of 1.9e$^4$, collision energy of 30, automatic gain control target of 1.9e$^4$, maximum inject time of 35 ms, dynamic exclusion of 1 count, 15 s of exclusion and exclusion mass window of ±5 ppm.

For sperm and vegetative nuclear proteomes, recalibrated peak lists were generated with MaxQuant (v.1.6.1.0)[46] in label-free quantitation mode using the TAIR10_pep_20101214 *Arabidopsis* protein sequence database (TAIR, 35386 entries) plus the MaxQuant contaminants database (245 entries). The label-free quantitation results from MaxQuant with default parameters were used together with search results from an in-house Mascot Server 2.4.1 (Matrix Science) on the same databases. For all searches, a precursor tolerance of 6 ppm and a fragment tolerance of 0.6 Da were used. The enzyme was set to trypsin/P with a maximum of 2 allowed missed cleavages, oxidation (M) and acetylation (protein N-term) were set as variable modifications and carbamido-methylation (C) was set as a fixed modification. The search results were imported into Scaffold 4 (Proteome Software) using identification probabilities of 99% for proteins and 95% for peptides.

## Histone alignments and disorder predictions

Alignments of histone DNA and protein sequences were performed using CLC Main Workbench software (v.8.1; Qiagen). Predictions of

intrinsic disorder were undertaken using PONDR[47] with the VL-XT algorithm. Raw data were plotted using ggplot2 in R (v.3.6.0)[48,49].

## Histone H2B phylogenetic analysis

Plant H2B protein sequences were downloaded from Phytozome[50], Congenie[51], Waterlily Pond[52], Magnoliid genomes[53,54] and UniProt[55]. Human and yeast H2B sequences were obtained from UniProt and used as out-groups for phylogenetics.

Sequences were imported to MEGA-X[56] and aligned using MUSCLE with default parameters. The phylogeny was generated using neighbour-joining testing, applying the Poisson model and allowing for uniform substitution rates. H2B.8 homologues were identified owing to the distinct branch formed, separate from canonical H2B variants. Several representative H2B.8 homologues were searched using BLAST[57] to query whether such homologues are specific to flowering plants. H2B.8 homologues are presented in Supplementary Table 1.

## Confocal microscopy and analysis

Microspores and pollen were isolated as previously described[45], stained with Hoechst 33342 and examined using a Leica SP8X confocal microscope. Young embryos were dissected[58] and stained with propidium iodide for imaging. Mature embryos were isolated from dry seeds using a stereomicroscope. Mature embryos and seedlings (including roots) were stained in PBS with 0.1% Triton X-100 and 0.5 µg ml$^{-1}$ DAPI for 5–10 min before microscopy examination (Zeiss 880, Airyscan mode). Immunofluorescence was performed with 2-week-old seedlings as previously described[18].

Sperm nuclear size was quantified from DAPI-stained whole pollen confocal images using a semi-automated pipeline in ImageJ adapted from ref. [59]. In brief, autothreshold was used to obtain nuclei and then processed using Gaussian blur to smooth edges. The autothreshold was repeated and then nuclei were selected using the wand tool. Measurements were then obtained for nuclear area (µm$^2$). Somatic nuclei selected for analysis were vascular cylinder cells in the elongation zone of the root tip. Such nuclei were selected owing to the ability to accurately identify the cell type within the tissue. Using ImageJ, $Z$-stacks were divided into substacks of different cell layers within the root tip. Maximum intensity projections were then obtained to account for slight differences in the depth of nuclei. Images were analysed in the same semi-automated way as per sperm nuclei. Statistical analysis was undertaken in R; we used analysis of variance followed by Tukey's post hoc test for pairwise comparisons.

Foci were identified in WT and *h2b.8* sperm using the ImageJ plug-in FociPicker3D[60].

For DAPI and H2B.8 colocalization (*pH2B.8::H2B.8–eGFP* and *p35S::H2B.8–eGFP*), nuclei were segmented using Otsu thresholding with 3D ImageJ Suite tools[61]. Fluorescence intensity measurements were extracted for DAPI and GFP channels within each segmented nucleus. Mean values were calculated for 20 × 20 voxel areas. Each volume value was normalized by dividing with the sum of DAPI or GFP intensity for the entire corresponding nucleus. Regions were grouped into eight quantiles by GFP value and plotted against DAPI intensity.

Classification of H3K9me2 foci was undertaken as previously described[18]. H2B.8-mediated chromatin aggregates were classified in the same way.

To quantify the size and number of H2B.8 aggregates and H3K9me2 domains, we used the 3D ImageJ Suite tools[61]. In each fluorescence channel, we undertook 3D Nuclei Segmentation using Otsu thresholding[61]. The number and volumes of segments were plotted in R using ggplot2.

## 3D-SIM and analysis

Sperm and vegetative nuclei were isolated from pollen (collected from about 1 ml of flowers) as previously described[45] and resuspended in 200 µl Galbraith buffer (45 mM MgCl$_2$, 30 mM sodium citrate, 20 mm MOPS and 0.1% Triton X-100, pH 7.0). For generative nuclei, bicellular pollen were manually dissected from flower buds and individually staged by DAPI staining and fluorescence microscopy. Nuclei were extracted from seedlings by finely chopping with a razor blade in lysis buffer (15 mM Tris-HCl pH 7.5, 2 mM EDTA, 0.5 mM spermine, 80 mM KCl, 20 mM NaCl and 0.1% Triton X-100). The suspension was filtered through a 35-µm filter (Corning) into a 1.7 ml tube. Nuclei were pelleted at 500$g$ for 3 min and resuspended in 200 µl lysis buffer.

Nuclei were fixed in solution with 4% MeOH-free formaldehyde (Thermo Scientific) for 5 min. HiQA number 1.5H coverslips (Cell-Path) were washed with 10% HCl for 30 min and then washed three times in H$_2$O for 5 min to remove impurities. Fixed nuclei were spun onto coverslips at 500$g$ for 3 min using a Shandon Cytospin 2. Fixation was repeated by blotting nuclei with 4% MeOH-free formaldehyde for 5 min. Fixative was removed and coverslips were washed three times in PBS, 5 min per wash. Nuclei were blocked with 3% BSA in PBS with 0.1% Tween-20 (PBST) for 30 min in a humidified chamber. If performing immunostaining, antibodies were diluted 200-fold in 3% BSA in PBST and then blotted to nuclei on coverslips. Antibody incubation occurred overnight at 4 °C. Primary antibodies were removed by washing three times with PBST for 5 min. Secondary antibodies were diluted similarly to primary antibodies and then added to nuclei. Incubation occurred for 1 h at room temperature in a humidified chamber. If not performing immunostaining, the protocol resumed at this point. PBST washes were repeated as before. Nuclei were stained in the dark with either DAPI or SYBR Green (Invitrogen) at 2 mg µl$^{-1}$ or 100× dilution, respectively for 5 min. DNA stain was removed by washing in H$_2$O for 5 min. Coverslips were adhered to slides in 13 µl Vectashield H-1000 mounting medium. Nuclei were imaged using a ×63 oil-immersion lens on a Zeiss Elyra PS.1 super-resolution microscope.

3D reconstructions for SIM were undertaken using Zeiss Zen Black software. Intensity profiles associated with images were acquired using the Interactive 3D Surface Plot plugin for ImageJ[62,63].

To acquire voxel intensities from WT and *h2b.8* sperm, nuclei were segmented using Otsu thresholding and individual voxels were extracted. Fluorescence intensity was normalized by dividing by total nuclear intensity. The density of binned voxel intensities was plotted with ggplot2 in R.

## Histone purification from *Escherichia coli*

Sequences for *A. thaliana* histone H2B.8, H2B.8ΔIDR, H2B.8-scrambledIDR, H2B.2, H2A, H3, H3 mutant K9CC110A and H4 were codon optimized for protein expression in *E. coli* and synthesized (Genewiz) into the pET30a+ vector with a non-cleavable C-terminal 8× His-tag then transformed into the *E. coli* strain BL21 and purified as previously described[64]. In brief, cells were grown to an optical density of 0.5–0.6 at 37 °C in LB medium with 30 µg ml$^{-1}$ kanamycin. Histone expression was induced by the addition of 0.5 mM isopropyl-β-d-thiogalactopyranoside and incubated for 4 h at 37 °C. Cells were collected by centrifugation at 4,000 r.p.m. for 30 min and resuspended in 1× PBS. Centrifugation was repeated and cells were resuspended in wash buffer (50 mM Tris-HCl pH 7.5, 100 mM NaCl, 1 mM EDTA and 5 mM 2-mercaptoethanol). Cells were lysed by ultrasonication and debris was pelleted at 18,000 r.p.m. for 20 min. The supernatant was discarded and the pellet was resuspended with wash buffer and 1% Triton X-100. The sample was sonicated and then washed a further two times with wash buffer. The pellet was then resuspended with unfolding buffer (20 mM Tris-HCl pH 7.5, 7 M guanidinium HCl and 5 mM 2-mercaptoethanol) and mixed at room temperature for 1.5 h to fully dissolve the pellet. The sample was centrifuged at 18,000 r.p.m. for 30 min. The supernatant was flash-frozen in liquid nitrogen and then stored at −80 °C.

## Histone octamer and nucleosome reconstitution

DNA templates of 12× 177 bp of the Widom 601 sequence were cloned and purified as previously described[65]. The sequence for the 177 bp DNA sequence is listed as follows:

5′-GAGCATCCGGATCCCCTGGAGAATCCCGGTGCCGAGGCCGCTCA ATTGGTCGTAGACAGCTCTAGCACCGCTTAAACGCACGTACGCGCTGT CCCCCGCGTTTTAACCGCCAAGGGGATTACTCCCTAGTCTCCAGGCAC GTGTCACATATATACATCCTGTTCCAGTGCCGGACCC-3′

Respective histone octamers were reconstituted as previously described[64]. In brief, equimolar amounts of four individual histones were added into unfolding buffer (20 mM Tris-HCl pH 7.5, 7 M guanidinium HCl and 5 mM B-ME) and were then dialysed into refolding buffer (2 M NaCl, 10 mM Tris-HCl pH 7.5, 1 mM EDTA and 5 mM 2-mercaptoethanol) before purification using Superdex 200 columns (Cytiva).

Nucleosomes were assembled using the salt-dialysis method as previously described[64]. Histone octamers and DNA templates were mixed in TEN buffer (10 mM Tris-HCl pH 7.5, 1 mM EDTA and 2 M NaCl) and dialysed for 16–18 h at 4 °C in TEN buffer, which was continuously diluted by slowly pumping in TE buffer (10 mM Tris-HCl pH 7.5, 1 mM EDTA) to lower the concentration of NaCl from 2 to 0.6 M. Samples were collected after final dialysis in HE buffer (10 mM HEPES pH 7.5, 0.1 mM EDTA) for 4 h. Nucleosomal array assemblies were assessed for quality by assembling under low concentration (2.5 μg DNA and the corresponding octamer in 50 μl reconstitution system), and the stoichiometry of histone octamers to DNA template was examined using an FEI Tecnai G2 Spirit 120 kV transmission electron microscope.

### Alexa Fluor 594 H3 K9CC110A labelling and labelled nucleosome reconstitution

Fluorophore labelling of histone H3 K9CC110A was performed through the addition of 1.5 M excess Alexa Fluor 594 (AF594)-C5-maleimide (Invitrogen) to histone in unfolding buffer followed by incubation in the dark for 4 h at room temperature. Conjugation reactions were quenched through the addition of 10 mM DTT, and free fluorophore was removed by flowing AF594-labelled histone H3 through a Superdex 200 Increase 10/300 column (Cytiva) in unfolding buffer. The fluorescently tagged H3 was then combined with corresponding histones in unfolding buffer for histone octamer reconstitution as described above. Histone H3 mutant C110A does not affect nucleosome structure or positioning[66], and H3 mutant K9CC110A is commonly used to install analogues of methyl lysine on cysteine with similar function to their natural counterparts[67,68]. Fluorescently labelled and unlabelled histone octamers were combined at a ratio of 1:100 before reconstitution to reduce the potential impact on samples during fluorescent labelling[27].

### In vitro chromatin condensate formation

In vitro phase-separation experiments were performed as previously described[27]. In brief, experiments were recorded on 384-well glass bottom plates (Cellvis) and sealed with optically clear adhesive film. Before use, 384-well plates were PEGylated with 5K mPEG-silane and passivated with BSA. Nucleosomal array samples were first equilibrated in chromatin dilution buffer (25 mM Tris·OAc, pH 7.5, 5 mM DTT, 0.1 mM EDTA and 0.1 mg ml$^{-1}$ BSA, 5% (w/v) glycerol) and incubated for 5 min at room temperature. Nucleosomal array samples were then added to 1 volume of chromatin dilution buffer in the presence of salt (KOAc and/or Mg(OAc)$_2$) and transferred to the PEGylated and BSA-passivated 384-well plates to incubate for 30 min at room temperature. Double-strand DNA within nucleosomal arrays were stained with YOYO-1 iodide (491/509) (Invitrogen) following the addition of chromatin dilution buffer.

### In vitro FRAP and droplet fusion

In vitro FRAP and droplet fusion experiments were carried out after chromatin condensates were formed using the conditions detailed above, using a NIKON A1 confocal microscope equipped with a ×100 oil-immersion objective. Droplets were bleached with a 561-nm laser pulse (2 repeats, 20% intensity, dwell time 1 s). Post-bleach intensity was normalized to pre-bleach levels to obtain a measure of recovery. NIS-Elements AR Analysis software was used for image analysis.

### Transient expression in tobacco

Vectors (*p35S::H2B.8–eGFP*, *p35S::H2B.8-scrambledIDR–eGFP*, *p35S::H2B.8-EWSR1-IDR–eGFP* and *p35S::H2B.8-TAF15-IDR–eGFP*) were transformed into *Nicotiana benthamiana* by *Agrobacterium* infection for transient expression. Two days after injection, the leaf epidermis was isolated and stained by DAPI solution containing 0.2% (v/v) Triton X-100. Images were obtained using a Leica SP8X confocal microscope.

### Drought stress treatment

Ten-day-old WT and *p35S::H2B.8–eGFP* seedlings (two replicates for each treatment) were collected and incubated for 3 h in 50% (w/v) PEG6000 1× PBS for drought treatment or 1× PBS for controls.

### Histone extraction for western blotting

Leaf tissue (0.5 g) was ground in liquid nitrogen and suspended in 4 ml histone extraction buffer (10 mM Tris-HCl pH 7.5, 2 mM EDTA, 0.25 M HCl, 5 mM DTT and 1× protease inhibitor cocktail (Roche)). The suspension was filtered through one layer of miracloth (Merck Millipore) and centrifuged at 4,000 r.p.m. for 10 min at 4 °C. The supernatant was transferred to a fresh tube and mixed with 1.75 ml trichloroacetic acid solution (T0699, Sigma). The precipitate was collected by centrifugation at 4,000 r.p.m. for 20 min at 4 °C and then washed twice with 1 ml acetone. The pellet was air dried before incubating with 1× LDS sample buffer (NP0007, Invitrogen) overnight. The sample was denatured at 95 °C for 10 min. For sperm, 5 million sperm nuclei were sorted by FACS (Supplementary Fig. 2a). The sperm nuclei were centrifuged at 1,200g for 5 min at 4 °C. The nuclei pellet was resuspended with 1 ml histone extraction buffer and then 0.25 ml trichloroacetic acid solution was added and mixed. Histones were precipitated and eluted in the same way as for leaf. Anti-H3K9me2 antibody (Abcam, ab1220) was diluted 1:5,000 and used for western blotting.

### Pollen and seedling native ChIP-seq library preparation, sequencing and analysis

About 0.5 g of 10-day-old seedlings (two replicates for each genotype) were ground with a pestle and mortar in liquid nitrogen and homogenized in nuclei isolation buffer (0.25 M sucrose, 15 mM PIPES pH 6.8, 5 mM MgCl$_2$, 60 mM KCl, 15 mM NaCl, 1 mM CaCl$_2$, 0.9% Triton X-100, 1 mM PMSF and 1× protease inhibitor cocktail (Roche)) for 15 min. Nuclei were separated from debris by filtering through two layers of miracloth (Merck Millipore). For pollen nuclei, we collected approximately 20 ml of open flowers and isolated pollen in Galbraith buffer (two replicates were isolated for each genotype). Nuclei were released by vertexing pollen with glass beads in 200 μl of nuclei isolation buffer at 2,000 r.p.m. for 3 min. The homogenate was filtered through 40-μm and 10-μm cell strainers successively to obtain nuclei. To maximize nuclei recovery, unbroken pollen grains that remained on the filters were recycled to the glass beads and vortexed with nuclei isolation buffer before proceeding to filter steps again.

Nuclei suspension from seedlings or pollen was centrifuged at 4,000g for 10 min and pellets were resuspended in TM2 (50 mM Tris-HCl, 2 mM MgCl$_2$, 0.25 M sucrose, 1 mM PMSF and 1× protease inhibitor cocktail). After cold centrifugation at 4,000g for 5 min, nuclei were resuspended in MNase digestion buffer (50 mM Tris-HCl pH 7.5, 5 mM CaCl$_2$, 0.25 M sucrose, 1 mM PMSF and 1× protease inhibitor cocktail) with an appropriate amount of MNase (New England Biolabs) and incubated at 37 °C for 10 min. Digestion was stopped by adding EDTA to a final concentration of 25 mM. One-tenth volume of 1% Triton X-100 and 1% sodium deoxycholate was added and the sample was left on ice for 15 min. Then, the reaction was diluted by adding low-salt buffer (50 mM Tris-HCl pH 7.5, 10 mM EDTA, 150 mM NaCl, 0.1% Triton X-100, 1 mM PMSF and 1× protease inhibitor cocktail) and rotated for 1 h at 4 °C. After centrifugation, the supernatant was used for immunoprecipitation with pre-washed GFP-Trap beads (Chromotek) overnight at 4 °C. Beads were

washed twice each with low-salt buffer and high-salt buffer (50 mM Tris-HCl pH 7.5, 10 mM EDTA, 300 mM NaCl, 0.1% Triton X-100 and 1 mM PMSF) and eluted in elution buffer (0.1 M NaHCO$_3$ and 1% SDS) by shaking at 65 °C for 15 min. The eluates were digested with proteinase K and RNase A before phenol–chloroform DNA extraction. Libraries were prepared using Ovation Ultralow System V2 and sequenced on NextSeq 500 (Illumina) with 2× 38 bp paired-end reads.

Sequencing reads were mapped to TAIR10 with Bowtie 2 (v.2.3.4.1)[69], retaining mononucleosomal fragments. Duplicated reads were removed using Samtools-1.7 rmdup. Bigwig files were generated by normalizing IP bam files to respective inputs using deepTools (v.3.1.1)[70]. Two replicates for each experiment were confirmed to be highly correlated; a single replicate was used for downstream analyses. Profiles were visualized using IGV (v.2.6.2)[71].

Data underlying metaplots and heatmaps were generated with deep-Tools and plotted with a custom script in R.

TE classes were defined by seedling H3K9me2 enrichment (log$_2$(Immunoprecipitation/input)). Considering the bimodal distribution of H3K9me2 enrichment at TEs, 0.7 was selected as the cut-off for heterochromatic (>0.7) and euchromatic (<0.7) classes.

To generate peaks, H2B.8 or H2B.8ΔIDR enrichment was calculated over 50-bp windows and those with >1.2 log$_2$(IP/input) were retained. Windows within 150 bp were merged using BEDtools (v.2.28.0)[72]. Regions were filtered by size, with those <200 bp removed from analysis. H2B.8 enrichment was then calculated over the new regions, and those with <1.2 log$_2$(IP/input) were discarded. The remaining regions were defined as H2B.8 peaks.

For genome coverage, peaks were divided into 50-bp windows and partitioned into gene, TE or intergenic groups depending on overlaps. Overlaps with genes and TEs for volcano plots were determined using BEDtools; 25% of the feature was required to be covered by a peak to be defined as an overlap.

Downloaded data[73–80] were mapped and processed in the same way. PCA, and Spearman and Pearson correlations were calculated with deepTools using categorial genomic regions and plotted in R with ggplot2.

### Bisulfite-seq analysis
Downloaded sequencing reads[81] were processed using TrimGalore (v.0.4.1) (https://github.com/FelixKrueger/TrimGalore) with default parameters. Reads were mapped to TAIR10 using Bismark (v.0.22.2)[82], and methylation was called using MethylDackel (v.0.5.2) (https://github.com/dpryan79/MethylDackel), selecting --CHG and --CHH options. CG methylation data were used in PCA.

### Sperm cell and seedling RNA-seq library preparation, sequencing and analysis
Sperm cells were isolated by FACS as previously described[83], with two replicates isolated for each genotype (Supplementary Fig. 2b). RNA was extracted from 1 million sperm cells (isolated from approximately 100 ml of open flowers) per replicate using a Direct-zol RNA Microprep kit. A Plant RNeasy Mini kit was used to extract RNA from 10-day-old seedlings (three or two replicates of seedlings were used for each genotype in normal or drought/mock treatment conditions, respectively). Libraries were prepared using a Universal RNA-Seq library preparation kit and sequenced on NextSeq 500 (Illumina) with single end (76 bp) or paired end (2× 38 bp) reads.

Sequencing reads were mapped to TAIR10 with TopHat (v.2.0.10)[84]. Kallisto (v.0.43.0)[85] and Sleuth-(v.0.30.0)[86] were used to obtain transcript per million (TPM) values and $q$ values, respectively. Differentially expressed genes and TEs were identified by | log$_2$(TPM fold-change) | ≥ 2 (≥1 was used for calling drought-responsive genes) and $q$ < 0.05. Volcano plots were generated with a custom ggplot2 R script. Heatmaps were generated using ggplot2 in R, with the transcriptional changes normalized using the som package.

Downloaded data[87–89] were mapped, and TPM values were obtained in the same way.

### Multivariate linear regression modelling
The genome was divided into 200-bp windows for calculating the abundance of each variable. For ChIP-seq data, log$_2$(IP/input) was calculated for each window using multiBigwigSummary in deeptools. For bisulfite-seq data, CG methylation was calculated using the number of sequenced Cs in the CG context within the window divided by the number of (C+T) in the CG context (windows without CG sites or aligned sequencing reads were removed). for RNA-seq data, log$_{10}$(reads per kilobase of transcript per million mapped reads (RPKM)) was calculated for each window.

Multivariate linear regression models were generated in R using the lm function. For each variable, a linear model was fitted to predict H2B.8 enrichment and the adjusted $R^2$ values are shown in Supplementary Table 2. Variables were gradually added to the model, and the predictive power of the adjusted model was examined by calculating adjusted $R^2$ and Akaike information criterion scores in R (Extended Data Fig. 4f). A separate model was built using categorial variables of genomic regions to assess the association between H2B.8 and genomic features (adjusted $R^2$ of the model is shown in Supplementary Table 2).

### Hi-C library preparation, sequencing and analysis
Two replicates of 10-day-old seedlings (1–3 g) were collected and fixed with 20 ml 2% formaldehyde solution for 15 min in vacuum conditions at room temperature and then quenched by adding 2.162 ml of 2.5 M glycine. Fixed seedling tissue was rinsed with water three times and dried with tissue paper.

The nuclei were released by grinding in liquid nitrogen and then resuspended with 25 ml of extraction buffer I (0.4 M sucrose, 10 mM Tris-HCl pH 8, 10 mM MgCl$_2$, 5 mM β-mercaptoethanol, 0.1 mM PMSF and 13 μl protease inhibitor). Nuclei were filtered through a miracloth (Calbiochem) and then centrifuged at 4,000 r.p.m. for 20 min at 4 °C. The supernatant was discarded and the pellet was resuspended with 1 ml of extraction buffer II (0.25 M sucrose, 10 mM Tris-HCl pH 8, 10 mM MgCl$_2$, 1% Triton X-100, 5 mM β-mercaptoethanol, 0.1 mM PMSF and 13 μl protease inhibitor). Then the mixture was centrifuged at 14,000 r.p.m. for 10 min at 4 °C and the pellet was resuspended with 300 μl of extraction buffer III (1.7 M sucrose, 10 mM Tris-HCl pH 8, 0.15% Triton X-100, 2 mM MgCl$_2$, 5 mM β-mercaptoethanol, 0.1 mM PMSF and 1 μl protease inhibitor). The mixture was loaded onto an equal amount of clean extraction buffer III and centrifuged at 14,000 r.p.m. for 10 min. The pelleted nuclei were washed twice with 1× ice cold CutSmart buffer and finally resuspended in 0.5 ml volume. SDS was applied to permeabilize nuclei at 65 °C for 10 min, Triton X-100 was added to quench SDS. Thereafter, chromatin was digested with 400 units of MboI overnight at 37 °C with gentle rocking. MboI was then denatured to cease activity.

Digested chromatin underwent DNA end-repair with biotin-14-dCTP insertion followed by blunt-end ligation. After decrosslinking with proteinase K at 65 °C, DNA was purified using a phenol–chloroform extraction method. Biotin-14-dCTP was removed from non-ligated DNA fragment ends using T4 DNA polymerase. DNA was sheared to a range of 200–600 bp by sonication. Next, the fragments underwent end-repair and were pulled down by streptavidin C1 magnetic beads to enrich for fragments containing contact information. Fragment ends were then A-tailed, sequencing adapters were ligated and libraries were amplified by PCR for 12–14 cycles. Following purification, libraries were sequenced using an Illumina HiSeq X Ten platform with 2× 150 bp length reads. The Hi-C library construction and sequencing were conducted by Annoroad Gene Technology.

Sequencing reads were mapped to the TAIR10 reference genome using the HiC-Pro (v.2.11.1) pipeline[90]. The bam files (bwt2merged.bam) generated by HiC-Pro containing mapped reads were used as input files for FAN-C (v.0.9.8)[91]. The module 'fanc auto' was applied

to generate 500, 100, 50, 10 and 1 kb contact matrices (hic files). The resultant hic files with 100-kb resolution were directed to the 'fanc expected' module to calculate the expected interaction probability against genomic distance for intrachromosomal interaction. For matrix and score comparisons, the default comparison method of fold-change was used with the 'fanc compare' command. The outputs (hic object) were transferred to text files by 'fanc dump' and were visualized as heatmaps in R using ggplot2. To explore whether higher contacts observed in *p35S::H2B.8–eGFP* depend on H2B.8 incorporation, the genome was binned into 1-kb windows. H2B.8 signals (log$_2$(IP/input)) of each window were generated and sorted into 20 quantiles by strength. The interaction frequency differences (values generated by FAN-C at 1-kb resolution) of each quantile pair for either short-range interactions or interactions between pericentromeric regions and chromosome arms were averaged and plotted as heatmaps. The resolution of our Hi-C data was estimated as previously reported[92]. Our Hi-C data was deemed to achieve 1-kb resolution as 80% of genomic bins (1 kb) had >1,000 contacts. Our WT data were compared to published contact matrices[36,37].

## Male transmission assay

Heterozygous *h2b.8-1* mutant and WT plants were crossed in two separate experiments by two different researchers. The inheritance of mutant versus WT alleles was determined by dCAPS genotyping. In one experiment, WT plants were used as females and crossed with heterozygous *h2b.8-1* mutant plants as males; a total of 743 F$_1$ progeny were genotyped, and the result is listed in Supplementary Table 3. In a second experiment, reciprocal crosses between heterozygous *h2b.8-1* mutant and WT plants were performed; a total of 574 and 575 F$_1$ progeny were genotyped (Supplementary Table 3). Statistical significance was tested using Fisher's exact test in R.

## In vitro pollen germination

In vitro pollen germination of WT and *h2b.8* was undertaken as previously described[18].

## Statistics and reproducibility

Statistical tests performed on experimental data and sample sizes are noted in figure legends. All data points are derived from biological replicates. Boxplots show median (thick black bar) and first and third quartiles, with lower and upper whiskers extending to 1.5-times the interquartile range of the first and third quartiles or the highest and lowest values, respectively. Exact *P* values for each pairwise comparison in all figures are listed in Supplementary Table 7. Micrographs throughout are representative of at least three independent experiments.

## Reporting summary

Further information on research design is available in the Nature Research Reporting Summary linked to this article.

## Data availability

Sequencing data generated in this study (ChIP-seq, RNA-seq and Hi-C) have been deposited in the Gene Expression Omnibus under accession number GSE161366. All remaining data are in the main paper or the supplementary materials.

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

**Acknowledgements** We thank C. Dean, S. Penfield and D. Zilberman for critical reading of the manuscript; S. Lopez and E. Wegel at the John Innes Centre Bioimaging Facility for assistance with microscopy; staff at the Norwich BioScience Institute Partnership Computing infrastructure for Science Group for High Performance Computing resources; and staff at Annoroad Gene Technology (Beijing, China) for Hi-C library preparation and sequencing. This work was funded by a UKRI-BBSRC Doctoral Training Partnerships studentship (BBM0112161 to T.B.), a Marie Skłodowska-Curie Action grant (101033109 to S.Z.), Biotechnology and Biological Sciences Research Council grants (BBS0096201 and BBP0135111 to S.H., M.V. and X.F.), a Centre of Excellence for Plant and Microbial Sciences grant (to S.H. and X.F.), grants from the National Key Research and Development Program (2019YFA0508403 to P.L.; 2017YFA0504202 to G.L.), grants from the National Natural Science Foundation of China (32100417 to L.W.; 31991161 to G.L.; 32150023, 32125010, 31871443 to P.L.), a Beijing Municipal Science and Technology Commission grant (Z201100005320013 to G.L.), a European Research Council Starting Grant ('SexMeth' 804981 to S.Z. and X.F.), and a EMBO Young Investigator Award (to X.F.).

**Author contributions** T.B., S.H. and X.F. conceived the study. T.B., S.H., L.W., S.Z., P.L. and X.F. designed the experiments. L.W. performed in vitro phase separation, FRAP and droplet fusion assays, and G.L. and P.L. supervised these experiments. T.B. (proteomics, 3D-SIM, transgenics, ChIP-seq, RNA-seq and plant phenotyping), S.H. (confocal, transgenics and ChIP-seq) and S.Z. (confocal, stress treatment, ChIP-seq, RNA-seq, western blotting, Hi-C analysis and plant phenotyping) performed all other experiments and analyses (which were supervised by X.F.). G.S. and M.V. assisted T.B. with mass spectrometry and genomic analysis, respectively. T.B., S.H. and X.F. wrote the manuscript, and all authors commented on the manuscript.

**Competing interests** The authors declare no competing interests.

**Additional information**
**Correspondence and requests for materials** should be addressed to Pilong Li or Xiaoqi Feng.

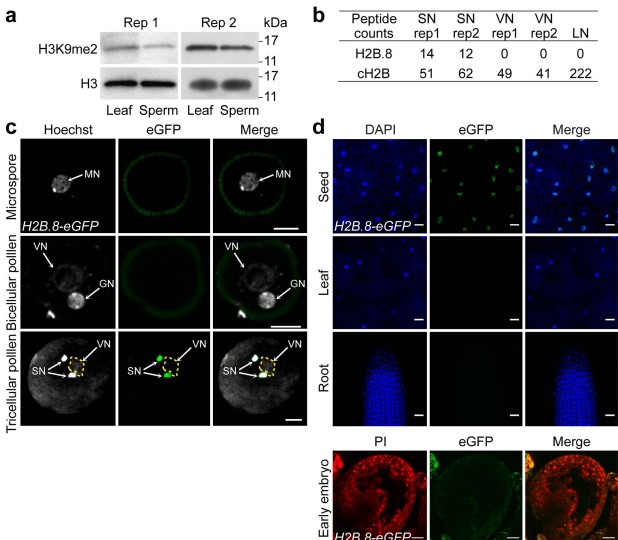

**b**

| Peptide counts | SN rep1 | SN rep2 | VN rep1 | VN rep2 | LN |
|---|---|---|---|---|---|
| H2B.8 | 14 | 12 | 0 | 0 | 0 |
| cH2B | 51 | 62 | 49 | 41 | 222 |

**Extended Data Fig. 1 | H2B.8 is specifically expressed in sperm and mature seeds. a**, Western blot showing H3K9me2 levels in leaf and sperm nuclei, with histone H3 as a control. Data shown represent two independent experiments. Rep, biological replicate. For source data, see Supplementary Fig. 1. **b**, Peptide counts from sperm (SN), vegetative (VN) and leaf nuclei (LN) mass spectrometry. cH2B, canonical H2B. **c**, Confocal images of H2B.8 (*pH2B.8::H2B.8-eGFP*) incorporation during male gametogenesis. Lowest panel is a duplicate of Fig. 1d. MN, VN, GN and SN, respectively, microspore, vegetative, generative and sperm nucleus. Data shown represent three independent experiments. Scale bars, 5 μm. **d**, Confocal images of various tissues from *pH2B.8::H2B.8-eGFP* plants. Data shown represent three independent experiments. Scale bars, 20 μm (early embryo, leaf, and root) and 5 μm (seed).

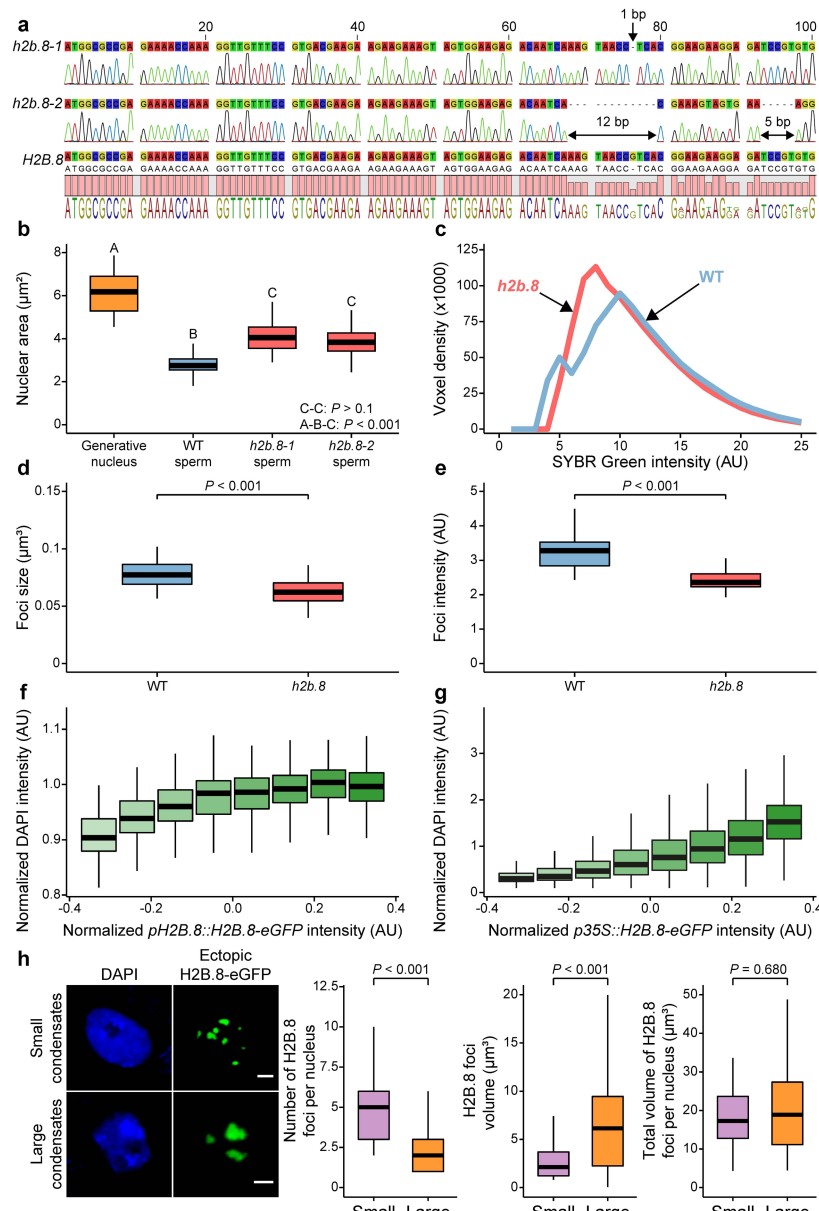

**Extended Data Fig. 2 | H2B.8 is required for sperm chromatin aggregation.** **a**, Alignment of *h2b.8* CRISPR lines. *h2b.8-1* has a single base deletion at 76 bp as indicated by an arrow, leading to a premature stop codon after 33 amino acids. *h2b.8-2* has a 12 bp deletion after 67 bp and another of 5 bp after 92 bp, producing a truncated 54 amino acid protein. **b**, Nuclear sizes of the generative nucleus and sperm nuclei of indicated genotypes. *P*-values, one-sided ANOVA followed by individual two-sample Tukey tests, which adjust for multiple comparisons. Boxplots marked as A, B and C are significantly different between groups (*P* < 0.001) but not within the group (*P* > 0.1). *n* = 30 (generative nucleus), 80 (WT, *h2b.8-1*), and 77 (*h2b.8-2*) nuclei examined over two independent experiments. **c**, Density plot of individual SYBR Green-stained voxel intensities from wild-type (WT, blue) and *h2b.8* (red) sperm nuclei. *n* = 30 nuclei for each. **d**, **e**, Boxplots indicating the volumes (**d**) and intensity (**e**) of SYBR Green foci in WT (blue) and *h2b.8* (red) sperm. *P*-value, independent two-sample t-test. *n* = 30 nuclei for each. AU, arbitrary unit. **f**, **g**, Colocalization of DAPI and H2B.8-eGFP signals in sperm (**f**, *pH2B.8::H2B.8-eGFP*) and in H2B.8 ectopically expressed seedlings (**g**, *p35S::H2B.8-eGFP*). *n* = 37 and 31 nuclei for sperm and seedlings, respectively. **h**, Confocal images and quantification of H2B.8 condensates in *p35S::H2B.8-eGFP* seedlings. *P*-value, independent two-sample t-test. *n* = 30 and 71 for nuclei with small and large H2B.8 condensates, respectively, examined over two independent experiments. Scale bars, 2 μm.

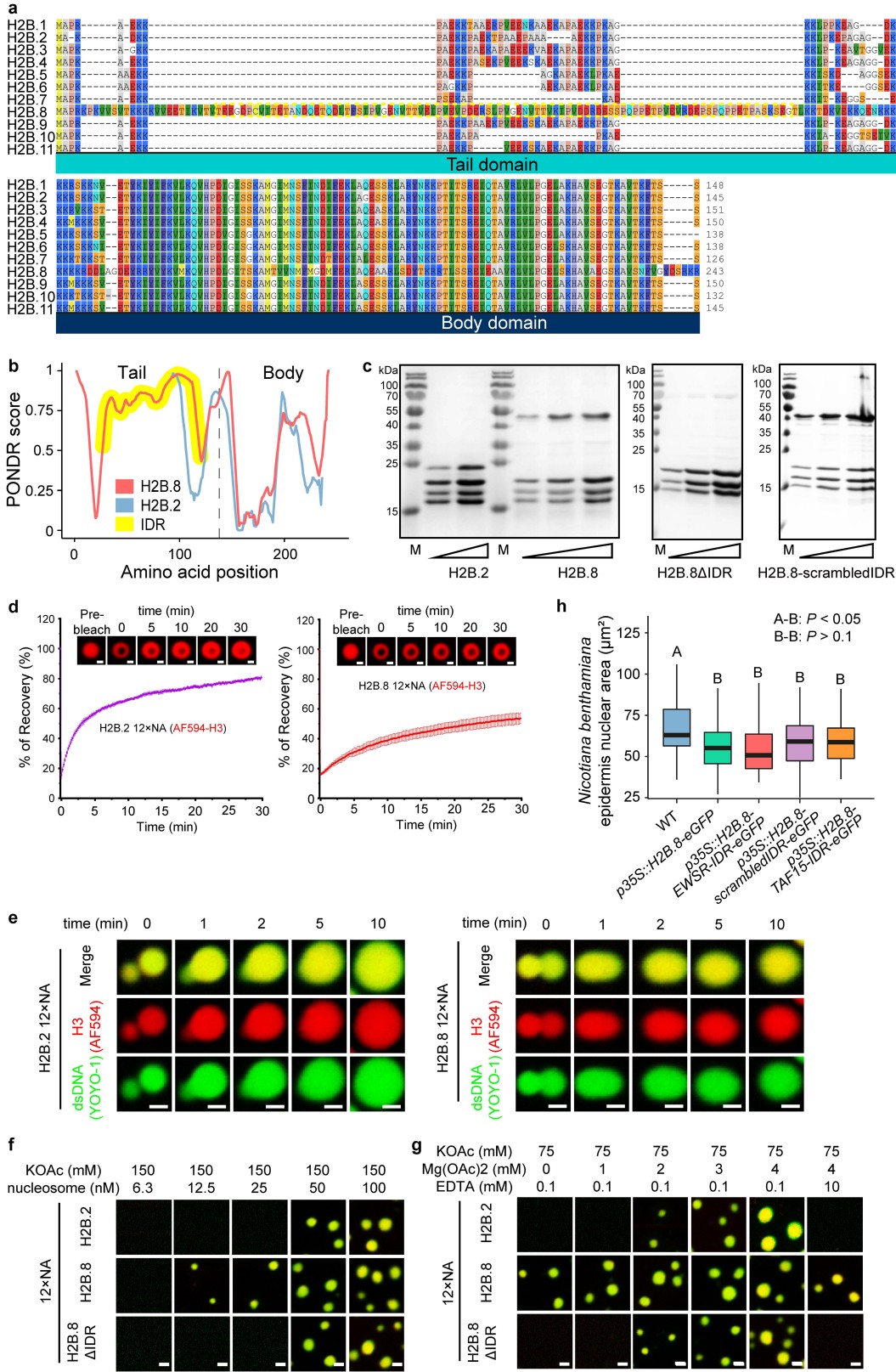

**Extended Data Fig. 3** | See next page for caption.

**Extended Data Fig. 3 | Phase separation property of H2B.8. a**, Alignment of *Arabidopsis* H2B variants (N-terminal tail – upper panel, C-terminal body – lower panel). The N-terminal tail IDR of H2B.8 is highlighted with a yellow dashed box. Amino acids are colored according to the RasMol scheme. **b**, Intrinsic disorder prediction of H2B.8 (red) and H2B.2, a canonical H2B (blue), by PONDR. Histone profiles are aligned at the interchange between tail and body domains (dashed line). H2B.8 N-terminal tail IDR is highlighted in yellow. **c**, SDS-PAGE of reconstituted histone octamers containing H2B.2, H2B.8, H2B.8ΔIDR or H2B.8-scrambledIDR. For gel source data, see Supplementary Fig. 1. **d**, Fluorescence recovery after photobleaching (FRAP) of H2B.2- or H2B.8-containing chromatin droplets. Average relative fluorescence intensity of the photobleached area across 6 individual chromatin droplets was calculated and displayed at each time point. Error bars, ±SD. Scale bars, 1 μm. **e**, Fusion of H2B.2- or H2B.8-containing chromatin droplets (histone H3 and dsDNA were labeled with AF594 and YOYO-1, respectively) over time. Data shown represent one representative experiment, which has been performed twice with similar results. Scale bars, 1 μm. **f, g**, In vitro phase separation of nucleosomal arrays (NA) containing H2B.2, H2B.8 or H2B.8ΔIDR at indicated salt and NA concentrations (in **g**, NA concentration for all panels was 50 nM). Histone H3 and dsDNA were labeled with AF594 and YOYO-1, respectively, and merged images of red and green channels are shown. Data shown represent one representative experiment, which has been performed three times with similar results. Scale bars, 2 μm. **h**, Nuclear size quantification of *Nicotiana benthamiana* epidermis cell nuclei transiently expressing native and chimeric H2B.8 as indicated. Boxplots marked as A and B are significantly different between groups (*P* < 0.001) but not within the group (*P* > 0.1). *n* = 30 (WT, *p35S::H2B.8-EWSR1-IDR-eGFP*, and *p35S::H2B.8-TAF15-IDR-eGFP*), 32 (*p35S::H2B.8-eGFP*), and 33 (*p35S::H2B.8-scrambledIDR-eGFP*) nuclei from two independent experiments.

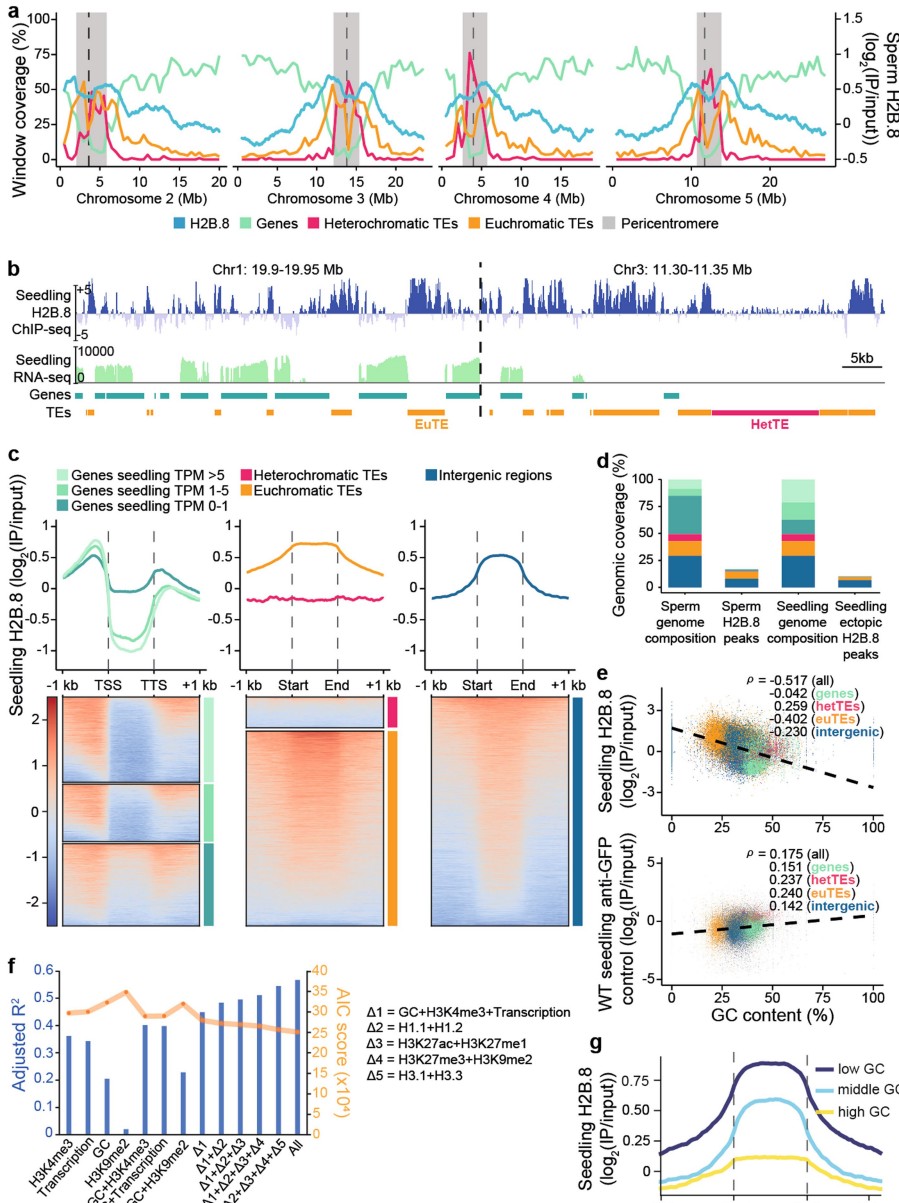

**Extended Data Fig. 4 | Ectopically expressed H2B.8 in seedlings exhibits a similar deposition profile as native H2B.8 in sperm. a**, Coverage of genes, euchromatic TEs and heterochromatic TEs (left Y axis; 500 kb windows) and H2B.8 enrichment in sperm (right Y axis; 1 kb windows) along Chromosomes 2 to 5. Chromosome 1 is shown in Fig. 4d. **b**, Genome snapshots of H2B.8 abundance (log$_2$(IP/input)) and transcription (log$_2$(RPKM)) in *p35S::H2B.8-eGFP* seedlings, and gene and TE annotations (orange, euchromatic TE; magenta, heterochromatic TE) over representative regions. RPKM, Reads Per Kilobase of transcript per Million mapped reads. **c**, Profiles and associated heatmaps of H2B.8 enrichment in *p35S::H2B.8-eGFP* seedlings over genes (grouped by seedling expression), TEs (grouped by chromatin state) and intergenic regions. **d**, Proportions (%) of the genome covered by genes, TEs and intergenic regions in wild-type sperm and

seedling, and by respective H2B.8 peaks in *pH2B.8::H2B.8-eGFP h2b.8* sperm and *p35S::H2B.8-eGFP* seedling. Same color coding is used as in (**c**) and Fig. 4b. **e**, Upper panel, scatterplot showing anticorrelation of H2B.8 enrichment with GC content (%) in *p35S::H2B.8-eGFP* seedlings over denoted genomic features. Lower panel, scatterplot showing the WT seedling anti-GFP enrichment (log$_2$(IP/input)) control against GC content (%). $\rho$, Spearman's Rank. **f**, Prediction of H2B.8 using indicated chromatin features, transcription (log$_{10}$(RPKM+1)) and GC content as predictors. The y-axis shows the adjusted R$^2$ value between the predicted and observed values. **g**, Profiles of H2B.8 enrichment in *p35S::H2B.8-eGFP* seedlings over non-genic (TE and intergenic) regions. The test regions were grouped into three equal parts by GC content levels (low GC < 0.29; 0.29 < middle GC < 0.34; high GC > 0.34).

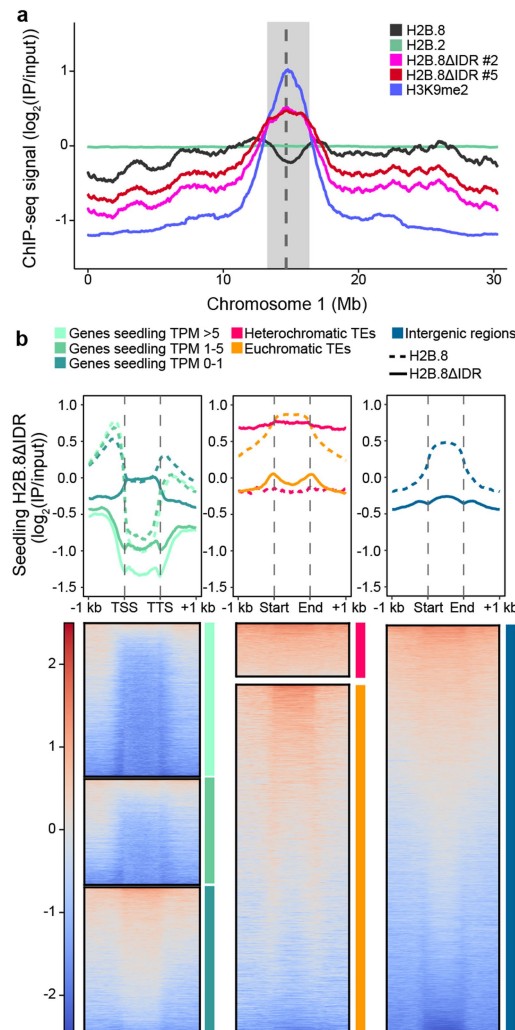

**Extended Data Fig. 5 | The IDR is required for the exclusion of H2B.8 from heterochromatin. a**, Enrichment of H2B.8 (*p35S::H2B.8-eGFP*), H2B.2 (*p35S::H2B.2-eGFP*), H2B.8ΔIDR (*p35S::H2B.8ΔIDR-eGFP* lines #2 and #5) and H3K9me2 in seedlings (log$_2$(IP/input); 1 kb windows). Grey dashed line localizes the centromere and grey shaded area represents pericentromeric heterochromatin. **b**, Profiles and associated heatmaps of H2B.8ΔIDR enrichment in *p35S::H2B.8ΔIDR-eGFP* seedlings over genes (grouped by seedling expression), TEs (grouped by chromatin state) and intergenic regions. For comparison, *p35S::H2B.8-eGFP* ChIP-seq data are shown as dashed lines (as in Extended Data Fig. 4c).

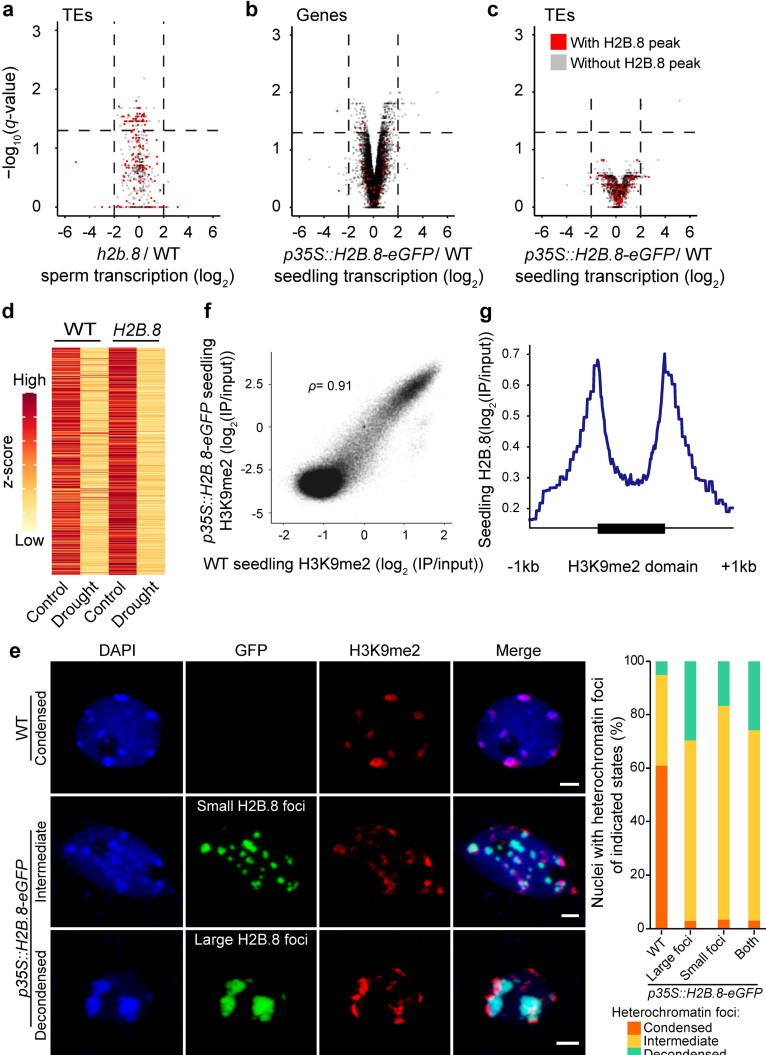

**Extended Data Fig. 6 | Ectopically expressed H2B.8 has a negligible effect on transcription. a-c**, Volcano plots showing differential TE (**a**, **c**) or gene (**b**) expression between *h2b.8* mutant and wild-type (WT) sperm cells (**a**), or *p35S::H2B.8-eGFP* and WT seedlings (**b**, **c**). TPM, Transcripts Per Million. n = 480 for TEs with TPM > 1 in either WT or *h2b.8* mutant sperm, and n = 20285 (**b**) and 1472 (**c**) for genes or TEs with TPM > 1 in either *p35S::H2B.8-eGFP* or WT seedlings. **d**, Heatmap depicting the transcription of genes (z-score) downregulated by drought stress in WT and *p35S::H2B.8-eGFP* seedlings subject to mock (control) and drought treatment. *n* = 771. **e**, Confocal images and quantification of seedling nuclei of indicated genotypes with condensed, intermediately

condensed or decondensed heterochromatin foci (measured by H3K9me2 signal). *n* = 138 (WT), and 71, 30 and 101 (*p35S::H2B.8-eGFP* nuclei with large, small, and both large and small H2B.8 foci, respectively). H2B.8 condensates are largely distinct from heterochromatin foci, overlapping only 19.0% ± 10.7% (standard deviation calculated from 53 nuclei) of the volume of heterochromatin foci. **f**, Scatterplot showing correlation of H3K9me2 levels (log2(IP/input)) over 1 kb windows between WT and *p35S::H2B.8-eGFP* seedlings. *ρ*, Spearman's Rank. **g**, Profile of H2B.8 enrichment in *p35S::H2B.8-eGFP* seedlings over H3K9me2-enriched domains.

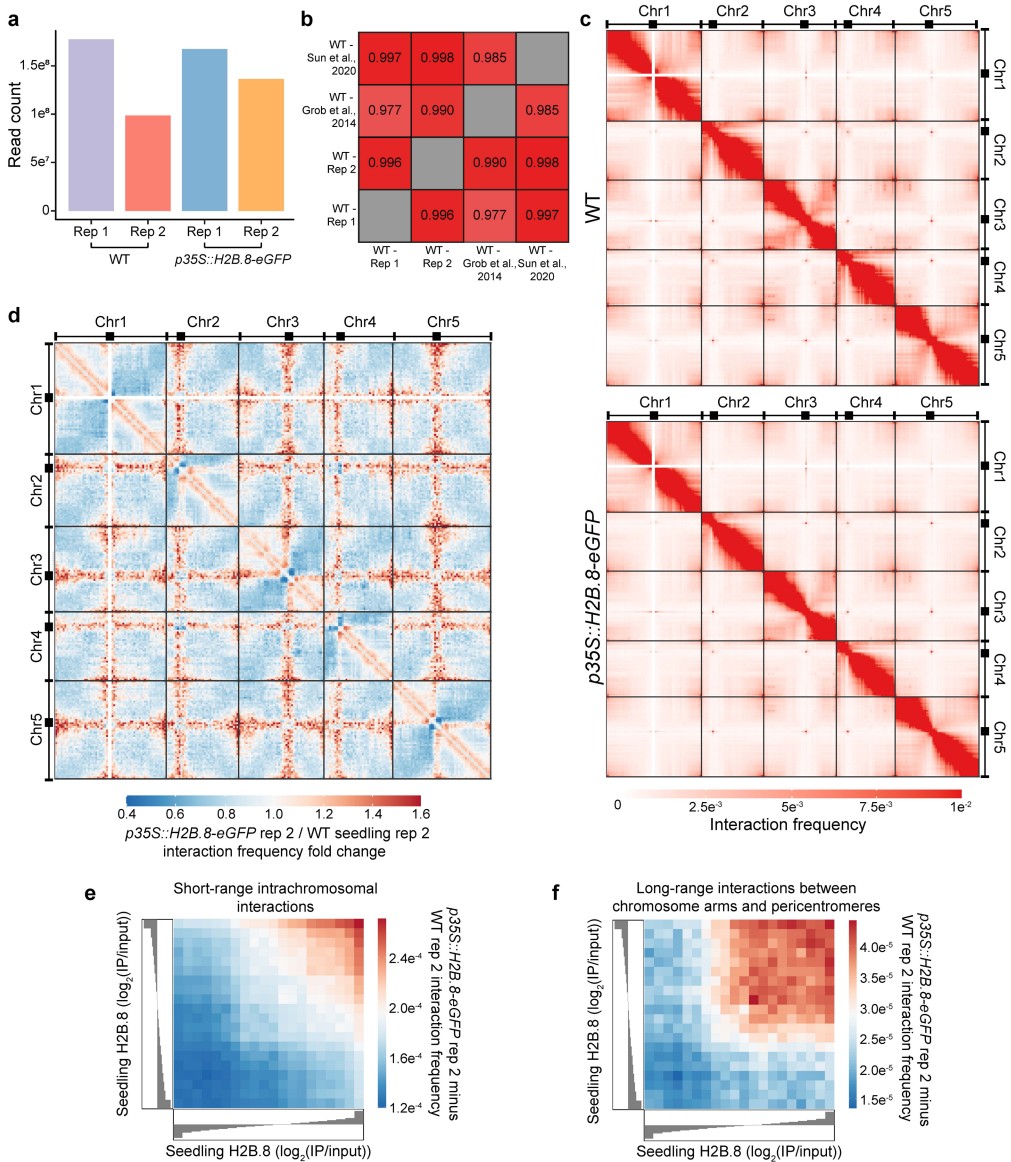

**Extended Data Fig. 7 | H2B.8 affects intra- and inter-chromosomal interactions. a**, Sequencing read counts of wild-type (WT) and *p35S::H2B.8-eGFP* seedling Hi-C libraries. Rep, biological replicate. **b**, Correlations between the Hi-C data generated in this study and previously published[36,37]. *R*, Pearson's correlation coefficient. **c**, Hi-C interaction frequency heatmaps at 500 kb resolution for WT (upper; merged replicates) and *p35S::H2B.8-eGFP* (lower; merged replicates) seedlings. **d**, Genome-wide interaction frequency fold change heatmap between WT (Rep 2) and *p35S::H2B.8-eGFP* (Rep 2) seedlings at

500 kb resolution, as shown in Fig. 6a for Rep 1. **e**, Short-range intrachromosomal interaction frequency difference between *p35S::H2B.8-eGFP* (Rep 2) and WT (Rep 2) over quantiles of seedling H2B.8 enrichment (log$_2$(IP/input)), as shown in Fig. 6b for Rep 1. Spearman's $\rho$ = 0.930. **f**, Long-range interaction frequency difference between *p35S::H2B.8-eGFP* (Rep 2) and WT (Rep 2) between chromosome arms and pericentromeric regions over quantiles of seedling H2B.8 enrichment (log$_2$(IP/input)), as shown in Fig. 6c for Rep 1. Spearman's $\rho$ = 0.870.

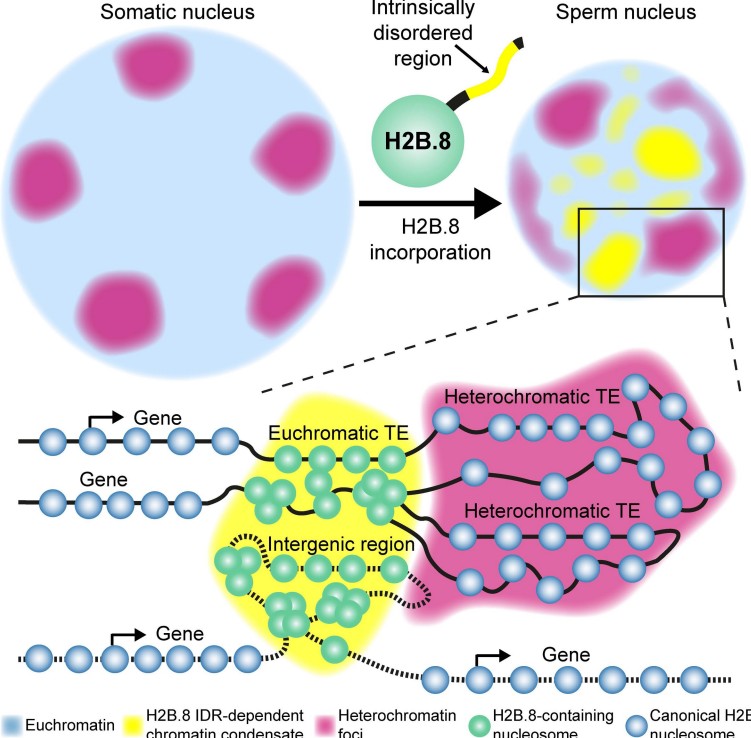

**Extended Data Fig. 8 | Mechanism of H2B.8-mediated sperm condensation.**
H2B.8 drives sperm nuclear condensation via the formation of chromatin condensates (yellow), which is dependent on an intrinsically disordered region (IDR) of H2B.8 conserved among flowering plants. Unlike typical chromatin condensation mechanisms, H2B.8-induced condensation does not inhibit transcription. Condensation is achieved by the specific deposition of H2B.8 into inactive AT-rich chromatin, which alters higher-order chromatin architecture to effectively compact the nucleus without sacrificing transcription. H2B.8-mediated chromatin aggregation disperses heterochromatin foci (pink), suggesting interactions between the euchromatic (yellow) and heterochromatic (pink) chromatin condensates in the nucleus.

# Reporting Summary

## Statistics

For all statistical analyses, confirm that the following items are present in the figure legend, table legend, main text, or Methods section.

| n/a | Confirmed | |
|---|---|---|
| ☐ | ☒ | The exact sample size (*n*) for each experimental group/condition, given as a discrete number and unit of measurement |
| ☐ | ☒ | A statement on whether measurements were taken from distinct samples or whether the same sample was measured repeatedly |
| ☐ | ☒ | The statistical test(s) used AND whether they are one- or two-sided<br>*Only common tests should be described solely by name; describe more complex techniques in the Methods section.* |
| ☐ | ☒ | A description of all covariates tested |
| ☐ | ☒ | A description of any assumptions or corrections, such as tests of normality and adjustment for multiple comparisons |
| ☐ | ☒ | A full description of the statistical parameters including central tendency (e.g. means) or other basic estimates (e.g. regression coefficient) AND variation (e.g. standard deviation) or associated estimates of uncertainty (e.g. confidence intervals) |
| ☐ | ☒ | For null hypothesis testing, the test statistic (e.g. $F$, $t$, $r$) with confidence intervals, effect sizes, degrees of freedom and $P$ value noted<br>*Give P values as exact values whenever suitable.* |
| ☒ | ☐ | For Bayesian analysis, information on the choice of priors and Markov chain Monte Carlo settings |
| ☒ | ☐ | For hierarchical and complex designs, identification of the appropriate level for tests and full reporting of outcomes |
| ☐ | ☒ | Estimates of effect sizes (e.g. Cohen's *d*, Pearson's *r*), indicating how they were calculated |

*Our web collection on statistics for biologists contains articles on many of the points above.*

## Software and code

Policy information about availability of computer code

| Data collection | Nuclei and cell sorting was performed using a BD FACSMelody™ Cell Sorter with BD FACSChorus™ software version 1.3. |
|---|---|
| Data analysis | ChIP-seq reads were mapped using Bowtie2-2.3.4.1 and analyzed using deepTools-3.1.1 and BEDtools-2.28.0. RNA-seq reads were mapped using TopHat-2.0.10. Kallisto-0.43.0 and Sleuth-0.30.0 were used to obtain TPM and q-values, respectively. Hi-C data was mapped using the HiC-Pro-2.11.1 pipeline and analyzed using FAN-C-0.9.8. Downloaded bisulfite-seq reads were processed using TrimGalore-0.4.1, mapped using Bismark-0.22.2 and methylation was called using MethylDackel-0.5.2. |

For manuscripts utilizing custom algorithms or software that are central to the research but not yet described in published literature, software must be made available to editors and reviewers. We strongly encourage code deposition in a community repository (e.g. GitHub). See the Nature Portfolio guidelines for submitting code & software for further information.

## Data

Policy information about availability of data

All manuscripts must include a data availability statement. This statement should provide the following information, where applicable:

- Accession codes, unique identifiers, or web links for publicly available datasets
- A description of any restrictions on data availability
- For clinical datasets or third party data, please ensure that the statement adheres to our policy

Sequencing data generated in this study (ChIP-seq, RNA-seq and Hi-C) have been deposited in the Gene Expression Omnibus (GEO) under accession no. GSE161366.

All remaining data are in the main paper or the supplementary materials. Further information and requests for resources, reagents or code should be directed to Xiaoqi Feng (xiaoqi.feng@jic.ac.uk) and Pilong Li (pilongli@mail.tsinghua.edu.cn).

## Human research participants

Policy information about studies involving human research participants and Sex and Gender in Research.

| | |
|---|---|
| Reporting on sex and gender | *Use the terms sex (biological attribute) and gender (shaped by social and cultural circumstances) carefully in order to avoid confusing both terms. Indicate if findings apply to only one sex or gender; describe whether sex and gender were considered in study design whether sex and/or gender was determined based on self-reporting or assigned and methods used. Provide in the source data disaggregated sex and gender data where this information has been collected, and consent has been obtained for sharing of individual-level data; provide overall numbers in this Reporting Summary. Please state if this information has not been collected. Report sex- and gender-based analyses where performed, justify reasons for lack of sex- and gender-based analysis.* |
| Population characteristics | *Describe the covariate-relevant population characteristics of the human research participants (e.g. age, genotypic information, past and current diagnosis and treatment categories). If you filled out the behavioural & social sciences study design questions and have nothing to add here, write "See above."* |
| Recruitment | *Describe how participants were recruited. Outline any potential self-selection bias or other biases that may be present and how these are likely to impact results.* |
| Ethics oversight | *Identify the organization(s) that approved the study protocol.* |

Note that full information on the approval of the study protocol must also be provided in the manuscript.

# Field-specific reporting

Please select the one below that is the best fit for your research. If you are not sure, read the appropriate sections before making your selection.

☒ Life sciences ☐ Behavioural & social sciences ☐ Ecological, evolutionary & environmental sciences

For a reference copy of the document with all sections, see nature.com/documents/nr-reporting-summary-flat.pdf

# Life sciences study design

All studies must disclose on these points even when the disclosure is negative.

| | |
|---|---|
| Sample size | No sample-size calculations were performed in this study. Sample sizes were chosen to be representative of the data distribution, similar to previous studies and more than sufficient to calculate statistical tests reliably. |
| Data exclusions | No data were excluded from the analysis. |
| Replication | All tests with replicates were successful. Imaging experiments were performed independently 3 times each to ensure reproducibility. ChIP-seq, RNA-seq and Hi-C experiments were undertaken 2-3 times each, as described in the methods. Phase separation assays were performed independently twice. |
| Randomization | Samples were allocated to groups randomly. |
| Blinding | All imaging analyses were double-blinded. Investigators were blinded in fertility experiments. Blinding was not relevant in sequencing experiments as it was not relevant to that type of data collection. |

# Reporting for specific materials, systems and methods

We require information from authors about some types of materials, experimental systems and methods used in many studies. Here, indicate whether each material, system or method listed is relevant to your study. If you are not sure if a list item applies to your research, read the appropriate section before selecting a response.

## Materials & experimental systems

| n/a | Involved in the study |
|---|---|
| ☐ | ☒ Antibodies |
| ☒ | ☐ Eukaryotic cell lines |
| ☒ | ☐ Palaeontology and archaeology |
| ☒ | ☐ Animals and other organisms |
| ☒ | ☐ Clinical data |
| ☒ | ☐ Dual use research of concern |

## Methods

| n/a | Involved in the study |
|---|---|
| ☐ | ☒ ChIP-seq |
| ☐ | ☒ Flow cytometry |
| ☒ | ☐ MRI-based neuroimaging |

# Antibodies

| Antibodies used | anti-H3K9me2; abcam; ab1220<br>GFP-Trap; ChromoTek; gta-100 |
|---|---|
| Validation | ab1220 - The manufacturer has performed peptide ELISA and Western blots to demonstrate ab1220 recognizes H3K9me2 only and not other methylated lysines in histone tails.<br>gta-100 - Manufacturer has validated the use in anti-GFP ChIP-seq along with ~3000 publications. |

# ChIP-seq

## Data deposition

☒ Confirm that both raw and final processed data have been deposited in a public database such as GEO.

☒ Confirm that you have deposited or provided access to graph files (e.g. BED files) for the called peaks.

| Data access links<br>*May remain private before publication.* | https://www.ncbi.nlm.nih.gov/geo/query/acc.cgi?acc=GSE161366 |
|---|---|
| Files in database submission | Native ChIP-seq profiles (IP and inputs) of H2B.8 in sperm (pH2B.8::H2B.8-eGFP) and ectopic H2B.8 (p35S::H2B.8-eGFP), H2B.2 (p35S::H2B.2-eGFP), H2B.8deltaIDR (p35S::H2B.8deltaIDR-eGFP) and H3K9me2 in ectopic H2B.8 (p35S::H2B.8-eGFP) seedlings. Raw data reads (FASTQ format) and normalized tracks (bigwig format) are available along with called peaks (BED format). |
| Genome browser session<br>(e.g. UCSC) | Bigwig files with normalized ChIP-seq profiles are provided in the GEO submission and can be viewed using IGV. |

## Methodology

| Replicates | Two biological replicates of ChIP-seq libraries were produced for each genotype. |
|---|---|
| Sequencing depth | ChIP-seq libraries were sequenced with 2 × 38 bp paired end reads to a depth of 30 million reads. |
| Antibodies | GFP-Trap; ChromoTek; gta-100<br>anti-H3K9me2; abcam; ab1220 |
| Peak calling parameters | Peaks were called using the pipeline as follows. H2B.8 enrichment was calculated over 50 bp windows and those with > 1.2 log2(IP/input) were retained. Windows within 150 bp were merged using BEDtools-2.28.0. Regions were filtered by size, with those < 200 bp removed from analysis. H2B.8 enrichment was then calculated over the new regions, those with < 1.2 log2(IP/input) were discarded. The remaining regions were defined as H2B.8 peaks. |
| Data quality | Peaks were manually validated in the genome browser to ensure they matched the ChIP-seq profiles. |
| Software | BEDtools-2.28.0 |

# Flow Cytometry

## Plots

Confirm that:

☒ The axis labels state the marker and fluorochrome used (e.g. CD4-FITC).

☒ The axis scales are clearly visible. Include numbers along axes only for bottom left plot of group (a 'group' is an analysis of identical markers).

☒ All plots are contour plots with outliers or pseudocolor plots.

☒ A numerical value for number of cells or percentage (with statistics) is provided.

## Methodology

| | |
|---|---|
| Sample preparation | Sperm and vegetative nuclei were isolated by FACS as described by Borges et al. (2012); sperm cells were isolated by FACS as described by Santos et al. (2017). |
| Instrument | BD FACSMelody™ cell sorter. |
| Software | BD FACSChorus™ software version 1.3. |
| Cell population abundance | Sample purity was assessed by microscopy and was >99%. |
| Gating strategy | To isolate sperm and vegetative nuclei, events were gated by SYBR Green intensity (FITC-A) and side scatter area (SSC-A) and then by forward scatter area (FSC-A). Nuclei types form two distinct populations owing to their different fluorescence intensities and nuclear sizes. To purify sperm cells from pollen, all events were gated for SYBR Green intensity (FITC-A) and side-scatter (SSC-A) to remove non-cell events. Sperm cells form a clear population, separate from sperm and vegetative nuclei, when gated for SYTOX Orange staining (PE-Cy7 (YG)-A). |

☒ Tick this box to confirm that a figure exemplifying the gating strategy is provided in the Supplementary Information.

