## [Peer Review File · Nature]

Manuscript Title: Histone H2B.8 compacts flowering plant sperm via chromatin phase separation

Reviewer Comments & Author Rebuttals

Reviewer Reports on the Initial Version:

Referees' comments:

Referee #1 (Remarks to the Author):

In this manuscript, Buttress et al. describe a new mechanism that regulates the size of nuclei in flowering plants. The authors identify a histone variant, H2B.8, which condenses nuclei in sperm cells via formation of gel-like condensates. The proposed model is a very interesting new mechanism that shapes chromatin organization in the nucleus. These results could be of broad interest, also considering the recent debates regarding the nature of chromatin condensates: Buttress et al. show an example of gel-like chromatin condensates that seem to be distinct from heterochromatin condensates, and that modulate the 3D genome organization by bringing together unexpressed euchromatin without affecting expressed genes. If correct, this could be an interesting example of chromatin organization in the nucleus that is uncoupled from gene regulation.

However, some of the claims made by the authors are not well supported by the data, and additional controls or replicates are needed in some experiments. Please find below specific comments.

1. The authors claim that the mechanism of H2B.8's function is via phase separation. While I think that all the evidence gathered points towards phase separation, it must be noted that this hasn't been definitively proven: the IDR of H2B.8 could also be mediating recruitment of a protein which in turn constitutes the mechanism. Therefore, the IDR deletion experiment is not sufficient to prove that phase separation is the mechanism. The authors should do additional experiments like scrambling the order of the amino acids of the IDR, which should maintain the residues important for phase separation but disrupt any possible binding motifs for interaction with other proteins. If the scrambled IDR looks the same as the IDR deletion, then this may not be a phase separation mechanism.

2. The methods section for the phase separation assay is not detailed and it seems the authors simply mixed the H2B.8 with naked DNA. This is not the usual method used to demonstrate phase separation of chromatin in vitro. Usually, nucleosome arrays are made with all histones (H3, H4, H2A, H2B) and then tested for phase separation capacity. Furthermore, there was no H2B.8 alone control in these assays. The authors should provide this data as it may be important to discern what aspects of their assay are creating condensates. The authors should clarify their methods and include the control while also explaining why they did the assay with just DNA, and how could naked DNA affect H2B.8 phase separation capacity in the absence of the other histones.

3. The analyses of the H2B.8 Δ IDR plants are missing for the sperm nuclei. Does H2B.8 Δ IDR rescue

the increased nuclear area in SN (Fig2A). What does H2B.8 Δ IDR look like in sperm nuclei using microscopy? The H2B.8 Δ IDR seems to be particularly enriched at heterochromatic foci which is a bit surprising (Fig. 2C). Is this only in seedling or also in SN? It would be important to know what the ChIP-seq pattern is for H2B.8 Δ IDR to compare it to wt H2B.8. Is the IDR required for localization of H2B.8 to inactive euchromatin (as the microscopy images might suggest)? If this is the case, it would contradict the model as it would suggest that the IDR is required for correct incorporation of H2B.8 on chromatin, and not just for phase separation.

4. The authors talk about a role of H2B.8 in nuclear and chromatin compaction. While I think that the data strongly supports a function of this histone variant in the regulation of nuclear size and in 3D genome organization, an effect on “chromatin compaction” hasn’t been shown. Typically, “chromatin compaction” is used to refer to denser nucleosomes and loss of DNA accessibility. The authors did not perform MNase, ATAC or any in vitro compaction assays. The increase in short range interactions observed by Hi-C could indicate chromatin compaction, but it could also be explained by the phase separation model as an increase in the likelihood of chromatin touching neighboring chromatin because of the condensates. This phase separation model does not require “chromatin compaction” in the sense of increased nucleosome density and loss of accessibility. We think that this should be discussed more clearly to avoid misinterpretation, and perhaps “chromatin compaction” could be replaced with “chromatin condensation”, “chromatin aggregation” or something similar.

5. The statement that “H2B.8 is permissive for transcription” is misleading: this statement implicates that genes can be expressed while being bound by H2B.8. The authors show instead that when genes need to be activated, H2B.8 is evicted, which suggests that these genes might be relocated outside of the H2B.8 condensates in order to be transcribed. While we don’t disagree with the statement that gene activation can occur despite H2B.8, we think it should be clarified and highlighted more that H2B never overlaps with transcribed genes, which in the end suggests that the H2B condensates are not permissive for transcription. Overall, it seems like the important concept is that H2B tends to bind and condense inactive DNA to decrease nuclear size while not affecting gene expression, because it doesn’t bind expressed genes.

6. The authors reference a very general review (ref 24) to support the notion that Arabidopsis sperm nuclei are very small. While this is quite evident from the images, a more careful analysis of the nuclear sizes would benefit the paper. More specifically, given that H2B.8 is incorporated after pollen mitosis 2, the authors should quantify the size of wt sperm nuclei and compare it directly to the size of GN nuclei, which are very similar (haploid as well) but lack H2B.8. How much smaller are SN compared to GN? Does the size of SN nuclei in h2b.8 mutants become the same as GN? If that is not the case, other mechanisms are likely involved in the nuclei compaction as well, and this should be noted.

7. Figure 1: the authors compare the volume of H3K9me2 foci in leaf and sperm nuclei and claim that larger foci in sperm indicates a “reduced level of heterochromatin condensation in sperm”. Is the number of H3K9me2 foci comparable in leaf and sperm nuclei (accounting for the difference in ploidy)? If chromocenters tend to coalesce more in sperm, higher volumes of H3K9me2 foci could be

caused by that, and not by reduced condensation. This should be clarified because it might go against the idea that H2B.8 causes dispersion of heterochromatin foci in SN. As suggested in the previous point, it would be useful to compare the H3K9me2 foci volumes in GN as well, given that GN has the same ploidy and is therefore a much better control for figure 1b. If the H2B.8 causes some dispersion of heterochromatin, the H3K9me2 foci volumes should be lower in GN than in SN.

8. The following sentence from the abstract is not very clear before reading the paper and therefore should be rephrased: “H2B.8 also intermixes inactive AT-rich chromatin and GC-rich pericentromeric heterochromatin, altering higher-order chromatin architecture”.

9. The authors show that H2B.8 ChIP-seq localizes to AT-rich inactive chromatin. Given the possible influence on this result of PCR amplification biases or non-specific binding of antibodies, it would be appropriate to validate the result by performing a side-by-side anti-GFP ChIP-seq on Col0 or h2b.8 tissue, and using this as negative control in addition to the input control.

10. The authors claim that “The best predictor of H2B.8 localization is GC content” and reference a PCA analysis and some scatterplots. The PCA analysis performed in Fig4F nicely shows that H2B.8 doesn't cluster closely to most eu- or heterochromatin marks, however, it is not the most appropriate statistical method to show that the best predictor of H2B.8 is GC content. To really compare the different predictors, the authors could make a multivariate linear regression model (or other more complicated models) and show the coefficients for each variable. The model could also include a categorical variable of region labels (intergenic, euTE, hetTE, expressed gene, unexpressed gene...).

11. The Hi-C result is an important part of the paper, as it nicely complements the imaging data to support a role of H2B.8 in chromatin organization. However, the authors only have a single replicate of Hi-C for each genotype. Given that the observed effect is very mild it would be critical to have several replicates to confirm the result.

12. The methods section should be more detailed in some paragraphs, to ensure repeatability of the experiments. For example: how many nuclei were sorted for total protein extraction or for RNA extraction and how much starting material was used? What amount of open flowers was used for pollen ChIP-seq and how was the vortexing performed? What does “twice” mean in “Nuclei were released by vortexing pollen with glass beads twice in nuclei isolation buffer.”? How did you do your phase separation assays? More detail is needed.

13. In the “reporting summary” you confirmed that the flow cytometry plots are available and conform the standards, but we could not find the sorting plots in any of the supplemental figures. Please provide all the information and figures that need to be provided for reproducibility purposes.

14. The authors state “we searched for sperm-specific chromatin factors by performing mass spectrometry on leaf nuclei and FACS-isolated sperm and vegetative nuclei”, but you only report the data for the sperm sample (Extended figure 1A). Moreover, the raw data for the mass spectrometry is missing. The authors should at least include the vegetative and leaf cells H2B data in the table

(Extended figure 1A) and should provide a justification for not including the raw data. This might need to be included in the “Reporting Summary”.

15. In the scatterplots of the RNAseq data, the red dots are barely visible. I recommend making the grey dots more transparent or empty. Moreover, the red vs grey labels are quite confusing because you showed that H2B.8 doesn't bind expressed genes. So, why do you have expressed genes overlapping with H2B.8 peaks?

16. Heatmaps in Fig5b and extended data Fig 5d: what does “high” and “low” mean? Is that a z-score? Or other type of normalization? Should be written in the legend.

17. Very minor point for Fig1c: given how small and simple the heatmap is, the authors could easily add the actual number of TPM for each tissue, instead of just the color. From the colors alone it is difficult to understand if the values are ~ 0 or ~ 50 .

Referee #2 (Remarks to the Author):

The regulation of sperm chromatin in flowering plants is distinct from that in animals and some other plants, in that they do not utilize protamines. Therefore, understanding the mechanisms of genome compaction in sperm of flowering plants is likely to uncover new principles of chromatin compaction. Here the authors investigate the roles of a plant sperm specific histone H2B.8. They propose this histone confers unique properties to the genomes of flowering plant sperm cells. Using the combination of high-resolution light microscopy, ectopic histone expression, biochemistry, transcription analysis and assessment of inter-chromosomal interactions they make the following findings. (a) H2B.8 contributes to compaction of sperm chromatin and nuclei and can drive some chromatin and nuclear compaction, and an increase in inter-chromosomal contacts when ectopically expressed in somatic cells (eg seedling and root cells); (b) H2B.8 added to DNA can form condensate-like aggregates in vitro and in cells. The condensate-like aggregates do not show rapid FRAP in either context; (c) H2B.8 does not contribute to transcriptional regulation in sperm cells but its presence is anti-correlated with heterochromatin puncta size/number; (d) H2B.8 is found at AT rich regions enriched in transposable elements.

Based on these findings the authors propose that H2B.8 condenses transcriptionally inactive AT-rich regions into condensates that also decondense heterochromatin resulting in a new type of condensed nuclear genome state that still allows for transcription. While the data do present some initial interesting correlations, they do not support the model. As such the weakest part of the study is the connection with phase-separation. In addition, alternative models are not mentioned or tested. Overall substantial additional work is needed for this study to reach the current field standard for work implying a role for phase-separation in genome regulation. Without such additional studies suggested below, the findings showing the impact of a previously identified sperm specific histone on global chromatin structure are not novel enough to be of broad interest.

Below are my major concerns and suggestions for additional experiments.

1. A central premise of this work is that H2B.8 promotes phase-separation. In cells H2B.8 will be part of nucleosomes. The studies shown here with only DNA and H2B.8 are not physiologically relevant. Prior work has shown how chromatin can intrinsically phase-separate. Hence at a minimum, the authors need to test if chromatin reconstituted with canonical plant H2B vs. H2B.8 shows different phase-separation properties using the types of assays used in PMID: 31543265. For example, compare the concentration of salt needed for phase-separation and the FRAP properties when H2B.8 is present in nucleosome arrays vs. canonical H2B.

2. If the authors are able to carry out the studies in 1 and find that H2B.8 directly promotes phase-separation of chromatin, then they can mutate the tail of H2B.8 and quantify its effects on the phase-separation properties of chromatin.

3. The in vivo FRAP studies are confusing. If the authors are carrying out FRAP on H2B.8 incorporated into chromatin, then the off rates of the histone are likely slower than the timescale of the FRAP experiments. In this context interpreting the cellular H2B.8 FRAP data as indicating a gel-like state is misleading.

4. Alternative models need to be considered and tested.

(i) In prior work referenced here-reference #27- studies were carried out on H2B.8 in Arabidopsis. Reference #27 mentions that H2B.8 lacks a mono-ubiquitylation site associated with transcription regulation and also proposes structure-based models for how H2B.8 may affect chromatin. The authors should comment on how their studies integrate these prior findings and models.

(ii) The authors suggest that H2B.8 condensates exclude heterochromatin condensates. However, a simple alternative model is one where H2B.8 prevents recruitment of heterochromatin factors through either specific PTMs or by recruiting factors that inhibit heterochromatin formation. Thus, indirect effects of H2B.8 also need to be considered rather than just direct effects. In this context, have the authors carried out IP-MS on H2B.8 to assess what factors it interacts with and/or whether it has PTMs differences from canonical H2B that could inhibit heterochromatin formation?

5. On line 9-10, the authors say; "However, these heterochromatin foci are enlarged in the sperm in comparison to leaf cells (Fig. 1a, b), despite sperm, being haploid, having only half as much DNA as leaf cells. Such enlargement indicates a reduced level of heterochromatin condensation in sperm." It is quite possible that the puncta are larger because a larger proportion of the genome is in heterochromatin. The authors need to test or provide evidence against this alternative possibility before making conclusions that there is a reduced level of heterochromatin condensation.

Referee #3 (Remarks to the Author):

The authors of this manuscript entitled "A histone variant condenses flowering plant sperm via chromatin phase separation" report that the histone H2B variant H2B.8 mediates chromatin

compaction in sperm by phase separation. H2B.8 was previously shown to be an angiosperm-specific histone variant, but the functional role remained elusive. The authors convincingly show that the presence of the N-terminally-located intrinsically disordered region confers the ability to H2B.8 to undergo phase separation and to form condensates. This special property most likely allows angiosperms to compact their sperm genome in the absence of protamines while maintaining transcriptional competency. These are very interesting findings worth to be reported. One aspect that would strongly add to this manuscript are data demonstrating the functional relevance of H2B.8. If chromatin compaction mediated by H2B.8 is relevant, decrease of sperm fitness would be expected. This could be tested by e.g. sperm competition experiments of h2b.8 mutant and wild-type sperm. But also without this data, this is a very nice, carefully done and important study, adding to our knowledge the mechanistic basis of reproductive traits.

Other comments

1. The authors state: „H2B.8 condensates are largely devoid of H3K9me2 and distinct from heterochromatin foci (Fig. 5e).” This requires quantitative validation. Based on the provided pictures I am not entirely convinced that this statement is correct. In fact, the conclusion of the authors that H2B.8 causes heterochromatin foci to decondense, would be in agreement with a co-localization of H3K9me2 and H2B.8 condensates. The interaction between H2B.8-induced heterochromatin and H3K9me2 would be most convincingly demonstrated by H3K9me2 ChIP in H2B.8 overexpressing lines.
2. The claim of increased short range and depleted long-range contacts is not obvious based on extended Data Fig. 6d. As shown by the authors, also long-range interactions correlate with abundance of H2B.8, similar to short range interactions (Fig. 6b,c), making the proposed scenario unlikely and inconsistent with the statement “The effect of H2B.8 is local, as the interactions between pericentromeric regions and chromosomal arms increase at regions with abundant H2B.8 (Fig. 6c).”
3. I found this statement rather counterintuitive: “However, the localization of H2B.8 in the non-transcribing parts of the genome suggests that it may not adversely affect gene expression.” If H2B.8 localizes in the non-transcribed part of the genome, one would expect that it negatively affects transcription. It has been nicely shown by the authors that this is not the case, nevertheless, it cannot be predicted based on the localization of H2B.8.

Author Rebuttals to Initial Comments:

We appreciate the constructive suggestions of the reviewers and we edited the manuscript in accordance with reviewer suggestions to include additional experiments and analyses. Below we respond to the specific comments of each reviewer, with the comments reproduced in italics.

Response to Referee #1

In this manuscript, Buttress et al. describe a new mechanism that regulates the size of nuclei in flowering plants. The authors identify a histone variant, H2B.8, which condenses nuclei in sperm cells via formation of gel-like condensates. The proposed model is a very interesting new mechanism that shapes chromatin organization in the nucleus. These results could be of broad interest, also considering the recent debates regarding the nature of chromatin condensates: Buttress et al. show an example of gel-like chromatin condensates that seem to be distinct from heterochromatin condensates, and that modulate the 3D genome organization by bringing together unexpressed euchromatin without affecting expressed genes. If correct, this could be an interesting example of chromatin organization in the nucleus that is uncoupled from gene regulation.

However, some of the claims made by the authors are not well supported by the data, and additional controls or replicates are needed in some experiments. Please find below specific comments

We thank the reviewer for the positive comments, and helpful questions and suggestions that strengthened our paper.

1. The authors claim that the mechanism of H2B.8's function is via phase separation. While I think that all the evidence gathered points towards phase separation, it must be noted that this hasn't been definitively proven: the IDR of H2B.8 could also be mediating recruitment of a protein which in turn constitutes the mechanism. Therefore, the IDR deletion experiment is not sufficient to prove that phase separation is the mechanism. The authors should do additional experiments like scrambling the order of the amino acids of the IDR, which should maintain the residues important for phase separation but disrupt any possible binding motifs for interaction with other proteins. If the scrambled IDR looks the same as the IDR deletion, then this may not be a phase separation mechanism.

We thank the reviewer for raising this important point. As suggested by the reviewer, we performed experiments using a mutated H2B.8 with the IDR amino acid sequence randomly scrambled. We found that H2B.8 with scrambled IDR (H2B.8-scrambledIDR) undergoes phase separation (Fig. 3b) and effectively reduces somatic nuclear size (Extended Data Fig. 3d), similar to the native H2B.8 and in contrast to H2B.8 Δ IDR (Fig. 2d, 3b, Extended Data Fig. 3d).

In addition, we replaced the H2B.8 IDR with animal IDRs of similar negative charge (EWSR1- and TAF15-IDR). As with the result from H2B.8 with scrambled IDR, expression of H2B.8-EWSR1-IDR and H2B.8-TAF15-IDR significantly reduces somatic nuclear size (Extended Data Fig. 3d). Taken together, these results unambiguously demonstrate that it is the phase separation ability of H2B.8, instead of the IDR sequence and potentially associated factors, that is essential for H2B.8 function. Consistent with this finding, although the IDR in the histone tail exists in all H2B.8 homologs in flowering plants, their sequences are highly variable and lack homology.

We have replaced the old Fig. 3b and Extended Data Fig. 3c, d with these new results, and added the following paragraph:

“The dependence of chromatin and nuclear condensation on the IDR of H2B.8 might result from the disordered state of the IDR and associated phase separation ability, or IDR-mediated recruitment of unknown chromatin condensing factors. To test these hypotheses, we expressed H2B.8 with a randomly scrambled IDR sequence (H2B.8-scrambledIDR) or with the native IDR replaced by animal IDR sequences of similar negative charge (H2B.8-EWSR-IDR and H2B.8-TAF15-IDR)⁴² in tobacco leaves and measured their effects on nuclear size. Resembling the effect of native H2B.8, the expression of H2B.8-scrambledIDR, H2B.8-EWSR-IDR and H2B.8-TAF15-IDR all effectively condensed the tobacco epidermis nuclei (Extended Data Fig. 3d). This indicates that H2B.8 function relies on the phase separation ability, instead of specific sequence motifs within the IDR. Consistently, while the presence of the IDR is conserved among flowering plants, the sequences of H2B.8 IDRs are diverse (Supplementary Table 1).” (Lines 193-204).

2. The methods section for the phase separation assay is not detailed and it seems the authors simply mixed the H2B.8 with naked DNA. This is not the usual method used to demonstrate phase separation of chromatin in vitro. Usually, nucleosome arrays are made with all histones (H3, H4, H2A, H2B) and then tested for phase separation capacity. Furthermore, there was no H2B.8 alone control in these assays. The authors should provide this data as it may be important to discern what aspects of their assay are creating condensates. The authors should clarify their methods and include the control while also explaining why they did the assay with just DNA, and how could naked DNA affect H2B.8 phase separation capacity in the absence of the other histones.

We agree with the reviewer that phase separation assays using H2B.8 with naked DNA do not make physiological sense. Therefore, in accordance, we assembled nucleosome arrays using recombinant *Arabidopsis* histone H3, H4, H2A, and H2B.2 (a canonical H2B) or H2B.8 (native, Δ IDR, or with scrambled IDR) with DNA and tested their phase separation capacities (Fig. 3b). As previously reported (Gibson et al., 2019 Cell; Strickfaden et al., 2020 Cell), we found that nucleosome arrays containing canonical H2B.2 or H2B.8 (native, Δ IDR, or with scrambled IDR) undergo phase separation at the presence of cation (Mg^{2+}). However, without salt, only nucleosome arrays containing native H2B.8 and H2B.8-scrambledIDR can phase separate, suggesting a distinct mechanism of chromatin phase separation endowed by the IDR of H2B.8, which is absent from canonical H2B.

We have replaced Fig. 3b and Extended Data Fig. 3c with these new results and revised the corresponding result and method sections.

Now in the results section: “To test this hypothesis, we expressed *A. thaliana* core histones (H2A, H2B, H3 and H4) in *E. coli*, assembled them into nucleosome arrays using a DNA template containing 12 repeats of the 601 nucleosome positioning sequence, and tested the phase separation properties of these nucleosome arrays (Fig. 3b, Extended Data Fig. 3c). We found that the addition of magnesium (Mg^{2+}) at physiologically relevant concentrations to chromatin reconstituted with either H2B.8 or a canonical H2B (H2B.2) that naturally lacks an IDR induces the formation of phase-separated condensates (Fig. 3b), consistent with the previous finding that chromatin undergoes phase separation *in vitro* at physiological salt conditions^{38,39}. However, without Mg^{2+} , H2B.8-containing nucleosome arrays can still form phase-separated condensates, whereas H2B.2-containing nucleosome arrays remain homogenous (Fig. 3b). H2B.8-containing chromatin condensates are insensitive to salt, as they

do not grow when salt is introduced into the solution (Fig. 3b). To test if the cation-independent phase separation ability of H2B.8-containing chromatin relies on the IDR, we reconstituted nucleosome arrays using H2B.8 without the IDR (H2B.8 Δ IDR) (Extended Data Fig. 3c). H2B.8 Δ IDR-containing chromatin fails to undergo phase separation in the absence of salt (Fig. 3b), showing that without the assistance of cations the IDR is critical for condensate formation. Consistent with the idea that IDRs promote phase separation via the disordered state instead of specific sequence motifs^{29,40}, we found chromatin containing H2B.8 with the IDR sequence randomly scrambled (H2B.8-scrambledIDR) is able to phase separate in a salt-independent manner (Fig. 3b). Taken together, our results demonstrate that the IDR of H2B.8 mediates a novel form of chromatin phase separation that is independent of physiological cations.” (Lines 147-167).

3. The analyses of the H2B.8 Δ IDR plants are missing for the sperm nuclei. Does H2B.8 Δ IDR rescue the increased nuclear area in SN (Fig2A). What does H2B.8 Δ IDR look like in sperm nuclei using microscopy? The H2B.8 Δ IDR seems to be particularly enriched at heterochromatic foci which is a bit surprising (Fig. 2C). Is this only in seedling or also in SN? It would be important to know what the ChIP-seq pattern is for H2B.8 Δ IDR to compare it to wt H2B.8. Is the IDR required for localization of H2B.8 to inactive euchromatin (as the microscopy images might suggest)? If this is the case, it would contradict the model as it would suggest that the IDR is required for correct incorporation of H2B.8 on chromatin, and not just for phase separation.

We appreciate the great suggestions by the reviewer. Accordingly, we expressed H2B.8 Δ IDR in *h2b.8* mutant and observed that unlike H2B.8, H2B.8 Δ IDR fails to rescue the enlarged sperm nuclear size in *h2b.8* mutant (this new result has been added to Fig. 2a). This demonstrates that in sperm, the IDR is required for H2B.8’s function in nuclear compaction, similar to the situation in somatic cells ectopically expressing H2B.8 or H2B.8 Δ IDR (Fig. 2d).

As suggested by the reviewer, we also examined the subcellular localization H2B.8 Δ IDR in sperm nuclei. Similar to ectopically expressed H2B.8 Δ IDR in seedling cells (Fig. 2c), H2B.8 Δ IDR preferentially localizes to heterochromatic foci (this new result has been added to Fig. 2c). This suggests that the IDR is important for the exclusion of H2B.8 from heterochromatin regions. Furthermore, we performed ChIP-seq on H2B.8 Δ IDR ectopically expressed seedlings, as suggested by the reviewer. We found that compared to full-length H2B.8, H2B.8 Δ IDR preferentially locates to pericentromeric heterochromatin (new Extended Data Fig. 5a), consistent with our cytological observation (Fig. 2c). The localization of H2B.8 Δ IDR along chromosome arms is generally reduced (Extended Data Fig. 5a). However, within chromosome arms, the general depletion of H2B.8 from transcriptionally active regions is unaffected by the deletion of IDR (new Extended Data Fig. 5a, b). In all, these experiments indicate that the IDR is required for the preferential localization of H2B.8 in euchromatin, but not for its exclusion from transcribed regions. This result does not contradict our model that the phase separation capability of H2B.8 is essential for its nuclear compaction function, which is further strengthened by new experiments described in our reply to Comment 1. It only adds to the model that the IDR of H2B.8 is not only important for phase separation, but also required for the localization of H2B.8 within chromatin. We have added the following paragraph to the corresponding results section to include these new results:

“It is yet unclear how H2B.8 localization is determined by GC content and transcription; however, our cytological observations have provided indications. 3D-SIM shows that unlike the full-length H2B.8 that largely avoids heterochromatin, H2B.8 Δ IDR preferentially locates to heterochromatin when expressed in sperm or seedling cells (Fig. 2c). To validate this

observation and examine the genomic localization of H2B.8 Δ IDR, we performed ChIP-seq on H2B.8 Δ IDR ectopically expressed (*p35S::H2B.8 Δ IDR-eGFP*) seedlings and compared with that of *p35S::H2B.8-eGFP* (and *p35S::H2B.2-eGFP* as a control). Consistent with the cytology, we found that H2B.8 Δ IDR preferentially locates to pericentromeric heterochromatin, and its abundance along chromosomal arms is generally reduced compared to H2B.8 (Extended Data Fig. 5a). Interestingly, although the IDR deletion drastically reduces the enrichment of H2B.8 in euchromatic TEs and increases H2B.8 deposition in heterochromatic TEs, it does not disrupt the preferential depletion of H2B.8 from transcribed genes (Extended Data Fig. 5b). This indicates that the IDR of H2B.8 is required for the preferential localization of H2B.8 in euchromatin, but not for its exclusion from transcribed regions.” (Lines 236-249).

4. The authors talk about a role of H2B.8 in nuclear and chromatin compaction. While I think that the data strongly supports a function of this histone variant in the regulation of nuclear size and in 3D genome organization, an effect on “chromatin compaction” hasn’t been shown. Typically, “chromatin compaction” is used to refer to denser nucleosomes and loss of DNA accessibility. The authors did not perform MNase, ATAC or any in vitro compaction assays. The increase in short range interactions observed by Hi-C could indicate chromatin compaction, but it could also be explained by the phase separation model as an increase in the likelihood of chromatin touching neighboring chromatin because of the condensates. This phase separation model does not require “chromatin compaction” in the sense of increased nucleosome density and loss of accessibility. We think that this should be discussed more clearly to avoid misinterpretation, and perhaps “chromatin compaction” could be replaced with “chromatin condensation”, “chromatin aggregation” or something similar.

We agree and have replaced all mentioning of “chromatin compaction” with “chromatin condensation” or “chromatin aggregation”.

5. The statement that “H2B.8 is permissive for transcription” is misleading: this statement implicates that genes can be expressed while being bound by H2B.8. The authors show instead that when genes need to be activated, H2B.8 is evicted, which suggests that these genes might be relocated outside of the H2B.8 condensates in order to be transcribed. While we don’t disagree with the statement that gene activation can occur despite H2B.8, we think it should be clarified and highlighted more that H2B never overlaps with transcribed genes, which in the end suggests that the H2B condensates are not permissive for transcription. Overall, it seems like the important concept is that H2B tends to bind and condense inactive DNA to decrease nuclear size while not affecting gene expression, because it doesn’t bind expressed genes.

We could not find the statement “H2B.8 is permissive for transcription” in our manuscript, but we agree with the need to clarify and highlight the fact that H2B.8 does not affect gene expression as it is excluded from transcribed regions. Therefore, we have revised the corresponding results section, which now reads “the localization of H2B.8 in the non-transcribing parts of the genome could arise via two mechanisms. Either H2B.8 suppresses transcription, or it is excluded from transcribed regions. To test these hypotheses, ... These results demonstrate that H2B.8 does not inhibit gene transcription, but is instead evicted by transcription, which likely underlies the localization of H2B.8 in unexpressed euchromatin.” (Lines 252-270).

6. The authors reference a very general review (ref 24) to support the notion that Arabidopsis sperm nuclei are very small. While this is quite evident from the images, a more careful analysis

of the nuclear sizes would benefit the paper. More specifically, given that H2B.8 is incorporated after pollen mitosis 2, the authors should quantify the size of wt sperm nuclei and compare it directly to the size of GN nuclei, which are very similar (haploid as well) but lack H2B.8. How much smaller are SN compared to GN? Does the size of SN nuclei in h2b.8 mutants become the same as GN? If that is not the case, other mechanisms are likely involved in the nuclei compaction as well, and this should be noted.

We thank the reviewer for the helpful suggestion. Accordingly, we quantified the size of generative cell nuclei (GN) and compared it with that of WT and *h2b.8* mutant sperm nuclei. This result is displayed in Extended Data Fig. 2b, and shows that sperm nuclei are indeed highly condensed, being significantly smaller than GN even though sperm and GN are both haploids. The size of *h2b.8* mutant sperm nuclei is significantly bigger than that of WT sperm nuclei, but still smaller than GN (Extended Data Fig. 2b), suggesting other mechanisms are involved in sperm nuclei compaction. We agree with the reviewer that this is an interesting point worthy of discussion. Therefore, we have added the following sentence accordingly:

“Notably, sperm nuclei of *h2b.8* mutants are still smaller than their progenitor, the generative nucleus, which is also haploid (Extended Data Fig. 2b), indicating that H2B.8 is not the sole mechanism involved in sperm nuclear compaction.” (Lines 116-118).

7. Figure 1: the authors compare the volume of H3K9me2 foci in leaf and sperm nuclei and claim that larger foci in sperm indicates a “reduced level of heterochromatin condensation in sperm”. Is the number of H3K9me2 foci comparable in leaf and sperm nuclei (accounting for the difference in ploidy)? If chromocenters tend to coalesce more in sperm, higher volumes of H3K9me2 foci could be caused by that, and not by reduced condensation. This should be clarified because it might go against the idea that H2B.8 causes dispersion of heterochromatin foci in SN. As suggested in the previous point, it would be useful to compare the H3K9me2 foci volumes in GN as well, given that GN has the same ploidy and is therefore a much better control for figure 1b. If the H2B.8 causes some dispersion of heterochromatin, the H3K9me2 foci volumes should be lower in GN than in SN.

We thank the reviewer for the constructive suggestions. Indeed, we found chromocenters tend to coalesce more in sperm. Therefore, we quantified the total volumes of H3K9me2 foci in each leaf, generative (GN) and sperm nucleus (SN), which replaced the original Fig. 1b that displays the volumes of individual H3K9me2 foci. We found that as expected, H3K9me2 foci in the haploid GN are of half of the volume of those in the diploid leaf nucleus. However, total volume of H3K9me2 foci in SN is ~30% larger than that in GN, despite that SN and GN are of the same ploidy (Fig. 1b). This strengthened our conclusion that heterochromatin condensation is reduced in sperm.

We have accordingly revised the relevant results section, which now reads “However, these heterochromatin foci appear enlarged in the sperm in comparison to leaf cells (Fig. 1a). To examine this further, we quantified the total volumes of H3K9me2-enriched heterochromatin foci in sperm and diploid leaf nuclei, as well as the nucleus of the generative cell, the haploid mother cell that divides into sperm cells. We found that the volume of the heterochromatin foci is reduced by half in the haploid generative nucleus compared to the diploid leaf nucleus, as expected (Fig. 1b). However, sperm heterochromatin foci are 27.5% larger than those of the generative nucleus (Fig. 1b)” (Lines 81-88).

We have also taken our experiments further and quantified the total volume of H3K9me2 foci in SN of the *h2b.8* mutant and *h2b.8* mutant carrying a *pH2B.8::H2B.8ΔIDR-eGFP* transgene.

h2b.8 mutation reduces H3K9me2 foci volume to that of the GN, indicating that H2B.8 causes the dispersion of heterochromatin in the SN (Fig. 5e). Furthermore, the expression of *H2B.8ΔIDR* fails to rescue the heterochromatin condensation phenotype of *h2b.8* (Fig. 5e), showing that the IDR is important for H2B.8's effect on heterochromatin decondensation. We have displayed this result as a new figure panel, Fig. 5e, and described it in the results section as follows:

“Besides the overall chromatin condensation in sperm, we observed slight decondensation of heterochromatin foci via 3D-SIM (Fig. 1a, b). To understand if this is caused by H2B.8, we performed immunostaining using H3K9me2 antibodies and measured the volume of H3K9me2 foci in *h2b.8* mutant nuclei. We found the *h2b.8* mutation significantly reduces the volume of heterochromatin foci in sperm, to a level resembling that in the generative nucleus (Fig. 5e).” (Lines 296-300).

“Our data also suggests that decondensation of heterochromatin foci is dependent on H2B.8 phase separation, as the expression of *H2B.8ΔIDR* does not affect heterochromatin (Fig. 5e).” (Lines 307-308).

8. *The following sentence from the abstract is not very clear before reading the paper and therefore should be rephrased: “H2B.8 also intermixes inactive AT-rich chromatin and GC-rich pericentromeric heterochromatin, altering higher-order chromatin architecture”.*

We agree that this sentence is confusing without the context. Since we added an extra results section on the fertility effect of H2B.8 according to Reviewer 3's comment, we deleted this sentence from the abstract and replaced it with a sentence describing the fertility result. Please refer to our reply to Reviewer 3's general comment for detailed explanation.

9. *The authors show that H2B.8 ChIP-seq localizes to AT-rich inactive chromatin. Given the possible influence on this result of PCR amplification biases or non-specific binding of antibodies, it would be appropriate to validate the result by performing a side-by-side anti-GFP ChIP-seq on Col0 or h2b.8 tissue, and using this as negative control in addition to the input control.*

We agree and have performed the experiments as requested. The anti-GFP ChIP-seq on Col-0 WT seedlings, as a negative control, shows no negative correlation with GC content (displayed in the revised Extended Data Fig. 4d), in contrast to the strong negative correlation between H2B.8 and GC content (Extended Data Fig. 4d). This serves as an additional negative control that validates the preference of H2B.8 towards AT-rich chromatin observed from our H2B.8 ChIP-seq experiment. This result is further supported by our additional anti-GFP ChIP-seq experiments on H2B.2 and H2B.8ΔIDR (Extended Data Fig. 5a, b), which also show no preference towards AT-rich sequences.

10. *The authors claim that “The best predictor of H2B.8 localization is GC content” and reference a PCA analysis and some scatterplots. The PCA analysis performed in Fig4F nicely shows that H2B.8 doesn't cluster closely to most eu- or heterochromatin marks, however, it is not the most appropriate statistical method to show that the best predictor of H2B.8 is GC content. To really compare the different predictors, the authors could make a multivariate linear regression model (or other more complicate models) and show the coefficients for each variable. The model could also include a categorical variable of region labels (intergenic, euTE, hetTE, expressed gene, unexpressed gene...).*

We thank the reviewer for the helpful suggestion. Accordingly, we generated multivariate linear regression models for H2B.8 using all variables in our PCA analysis (Fig. 4f) and categorical variables of genomic regions. The result has been added to the manuscript as Supplementary Table 2, which shows that GC content and histone modifications associated with transcription (H3K4me3 and H2Bub) have the largest absolute beta coefficients among all variables and strongly anti-correlate with H2B.8. For the categorical variables, expressed genes have the strongest negative coefficients, while euchromatic TEs (euTE) and intergenic regions have the strongest positive coefficients.

We have accordingly revised the text, which now reads “Principal component analysis (PCA) revealed that H2B.8 clusters with neither permissive nor repressive chromatin modifications, but associates with GC content (Fig. 4f). Multivariate linear regression modelling of H2B.8 further showed that GC content and transcription are the best predictors of H2B.8 localization, with which strong anti-correlations exist (Fig. 4g, Extended Data Fig. 4d, Supplementary Table 2).” (Lines 226-230).

11. The Hi-C result is an important part of the paper, as it nicely complements the imaging data to support a role of H2B.8 in chromatin organization. However, the authors only have a single replicate of Hi-C for each genotype. Given that the observed effect is very mild it would be critical to have several replicates to confirm the result.

We agree and have accordingly performed Hi-C experiments with an additional biological replicate for each genotype (added to Extended Data Fig. 7a-c). Analyses with data obtained from these new replicates (Extended Data Fig. 7d-f) show the same results as the original set of replicates (Fig. 6a-c), supporting our conclusions. We have accordingly revised Extended Data Fig. 7a-c and added new panels (Extended Data Fig. 7d-f) to incorporate data from these new replicates.

12. The methods section should be more detailed in some paragraphs, to ensure repeatability of the experiments. For example: how many nuclei were sorted for total protein extraction or for RNA extraction and how much starting material was used? What amount of open flowers was used for pollen ChIP-seq and how was the vortexing performed? What does “twice” mean in “Nuclei were released by vortexing pollen with glass beads twice in nuclei isolation buffer.”? How did you do your phase separation assays? More detail is needed.

We thank the reviewer for pointing this out, and have extensively revised the methods section accordingly.

The specific questions raised by the reviewer are addressed at Lines 672-673, 792-843, 872-878, and 926-927. Other revisions can be found at Lines 855, 863, 868, and 941.

13. In the “reporting summary” you confirmed that the flow cytometry plots are available and conform the standards, but we could not find the sorting plots in any of the supplemental figures. Please provide all the information and figures that need to be provided for reproducibility purposes.

We apologize for our oversight to include the flow cytometry plots, which are now added as Extended Data Fig. 9 and the information is provided in the legend.

14. The authors state “we searched for sperm-specific chromatin factors by performing mass spectrometry on leaf nuclei and FACS-isolated sperm and vegetative nuclei”, but you only

report the data for the sperm sample (Extended figure 1A). Moreover, the row data for the mass spectrometry is missing. The authors should at least include the vegetative and leaf cells H2B data in the table (Extended figure 1A) and should provide a justification for not including the raw data. This might need to be included in the “Reporting Summary”.

We agree and have accordingly included the vegetative and leaf nucleus H2B data in the table (Extended Data Fig. 1b). The justification for not including the raw data is included in the Reporting Summary: “All relevant information on H2B from our mass spectrometry experiment is provided in Extended Data Fig. 1b; the raw data, which are largely irrelevant to the present study, are not provided”.

15. In the scatterplots of the RNAseq data, the red dots are barely visible. I recommend making the grey dots more transparent or empty. Moreover, the red vs grey labels are quite confusing because you showed that H2B.8 doesn't bind expressed genes. So, why do you have expressed genes overlapping with H2B.8 peaks?

We have redrawn the scatterplots by making the red dots solid and the grey dots more transparent (and making all dots smaller). The red dots are now much more visible (revised Fig. 5a and Extended Data Fig. 6a-c).

We apologize for the confusion regarding the “expressed genes” in the scatter plots as we did not make it clear in the figure legend. Instead of showing all genes, we only plotted for genes with detectable transcription in either genotype to avoid the situation that differentially expressed genes are masked in the plots by the vast majority of genes that are not expressed. To ensure the inclusion of all meaningful data points, we set a low expression cutoff, TPM > 1, for calling “expressed genes”. Indeed, the median TPM of the “expressed genes” that overlap with H2B.8 peaks (ie. the red dots in Fig. 5a) is only 2.3. We have now made this clear in corresponding figure legends (Fig. 5a, Extended Data Fig. 6a-c).

16. Heatmaps in Fig5b and extended data Fig 5d: what does “high” and “low” mean? Is that a z-score? Or other type of normalization? Should be written in the legend.

Yes, it is z-score. We now label this clearly in the figure panels and legends (Fig. 5b, Extended Data Fig. 6d).

17. Very minor point for Fig1c: given how small and simple the heatmap is, the authors could easily add the actual number of TPM for each tissue, instead of just the color. From the colors alone it is difficult to understand if the values are ~0 or ~50.

The values are now added to Fig. 1c.

Response to Referee #2

The regulation of sperm chromatin in flowering plants is distinct from that in animals and some other plants, in that they do not utilize protamines. Therefore, understanding the mechanisms of genome compaction in sperm of flowering plants is likely to uncover new principles of chromatin compaction. Here the authors investigate the roles of a plant sperm specific histone H2B.8. They propose this histone confers unique properties to the genomes of flowering plant sperm cells. Using the combination of high-resolution light microscopy, ectopic histone expression, biochemistry, transcription analysis and assessment of inter-chromosomal interactions they make the following findings. (a) H2B.8 contributes to compaction of sperm chromatin and nuclei and can drive some chromatin and nuclear compaction, and an increase in inter-chromosomal contacts when ectopically expressed in somatic cells (eg seedling and root cells); (b) H2B.8 added to DNA can form condensate-like aggregates in vitro and in cells. The condensate-like aggregates do not show rapid FRAP in either context; (c) H2B.8 does not contribute to transcriptional regulation in sperm cells but its presence is anti-correlated with heterochromatin puncta size/number; (d) H2B.8 is found at AT rich regions enriched in transposable elements.

Based on these findings the authors propose that H2B.8 condenses transcriptionally inactive AT-rich regions into condensates that also decondense heterochromatin resulting in a new type of condensed nuclear genome state that still allows for transcription. While the data do present some initial interesting correlations, they do not support the model. As such the weakest part of the study is the connection with phase-separation. In addition, alternative models are not mentioned or tested. Overall substantial additional work is needed for this study to reach the current field standard for work implying a role for phase-separation in genome regulation. Without such additional studies suggested below, the findings showing the impact of a previously identified sperm specific histone on global chromatin structure are not novel enough to be of broad interest.

We thank the reviewer for their overall positivity and for pinpointing the weaknesses of our manuscript. We agree and have accordingly performed substantial experiments and analyses to strengthen our claims and test against alternative models. We reply to the reviewer's specific comments below and hope these new results, which have substantially improved our paper, will satisfy the reviewer.

Below are my major concerns and suggestions for additional experiments.

1. A central premise of this work is that H2B.8 promotes phase-separation. In cells H2B.8 will be part of nucleosomes. The studies shown here with only DNA and H2B.8 are not physiologically relevant. Prior work has shown how chromatin can intrinsically phase-separate. Hence at a minimum, the authors need to test if chromatin reconstituted with canonical plant H2B vs. H2B.8 shows different phase-separation properties using the types of assays used in PMID: 31543265. For example, compare the concentration of salt needed for phase-separation and the FRAP properties when H2B.8 is present in nucleosome arrays vs. canonical H2B.

2. If the authors are able to carry out the studies in 1 and find that H2B.8 directly promotes phase-separation of chromatin, then they can mutate the tail of H2B.8 and quantify its effects on the phase-separation properties of chromatin.

We thank the reviewer for these constructive suggestions. We have accordingly assembled nucleosome arrays containing H2B.8, H2B.2 (a canonical H2B), H2B.8 Δ IDR and H2B.8 with the IDR sequence randomly scrambled (H2B.8-scrambledIDR), and examined their phase separation properties as described in PMID: 31543265 (revised Fig. 3b and Extended Data Fig. 3c). Please refer to our reply to Reviewer 1's Comment 2 for a detailed explanation of the result. In brief, we found that consistent with the finding of PMID: 31543265, nucleosomes containing H2B.2 or all forms of H2B.8 tested are able to undergo phase separation under physiological salt condition (Fig. 3b). However, without salt (Mg^{2+}), only nucleosomes containing the native H2B.8 and H2B.8-scrambledIDR can phase separate, while those containing H2B.2 or H2B.8 Δ IDR cannot (Fig. 3b). These new results indicate a novel mechanism of chromatin phase separation that is endowed by the IDR and independent of the assistance of cation.

We have replaced the original Fig. 3b and Extended Data Fig. 3c with these new results and described them in the results section as follows:

“To test this hypothesis, we expressed *A. thaliana* core histones (H2A, H2B, H3 and H4) in *E. coli*, assembled them into nucleosome arrays using a DNA template containing 12 repeats of the 601 nucleosome positioning sequence, and tested the phase separation properties of these nucleosome arrays (Fig. 3b, Extended Data Fig. 3c). We found that the addition of magnesium (Mg^{2+}) at physiologically relevant concentrations to chromatin reconstituted with either H2B.8 or a canonical H2B (H2B.2) that naturally lacks an IDR induces the formation of phase-separated condensates (Fig. 3b), consistent with the previous finding that chromatin undergoes phase separation *in vitro* at physiological salt conditions^{38,39}. However, without Mg^{2+} , H2B.8-containing nucleosome arrays can still form phase-separated condensates, whereas H2B.2-containing nucleosome arrays remain homogenous (Fig. 3b). H2B.8-containing chromatin condensates are insensitive to salt, as they do not grow when salt is introduced into the solution (Fig. 3b). To test if the cation-independent phase separation ability of H2B.8-containing chromatin relies on the IDR, we reconstituted nucleosome arrays using H2B.8 without the IDR (H2B.8 Δ IDR) (Extended Data Fig. 3c). H2B.8 Δ IDR-containing chromatin fails to undergo phase separation in the absence of salt (Fig. 3b), showing that without the assistance of cations the IDR is critical for condensate formation. Consistent with the idea that IDRs promote phase separation via the disordered state instead of specific sequence motifs^{29,40}, we found chromatin containing H2B.8 with the IDR sequence randomly scrambled (H2B.8-scrambledIDR) is able to phase separate in a salt-independent manner (Fig. 3b). Taken together, our results demonstrate that the IDR of H2B.8 mediates a novel form of chromatin phase separation that is independent of physiological cations.” (Lines 147-167).

3. The in vivo FRAP studies are confusing. If the authors are carrying out FRAP on H2B.8 incorporated into chromatin, then the off rates of the histone are likely slower than the timescale of the FRAP experiments. In this context interpreting the cellular H2B.8 FRAP data as indicating a gel-like state is misleading.

We thank the reviewer for pointing this out and completely agree. We have, therefore, removed the *in vivo* FRAP results from the manuscript.

4. Alternative models need to be considered and tested.

(i) In prior work referenced here-reference #27- studies were carried out on H2B.8 in Arabidopsis. Reference #27 mentions that H2B.8 lacks a mono-ubiquitylation site associated with transcription regulation and also proposes structure-based models for how H2B.8 may

affect chromatin. The authors should comment on how their studies integrate these prior findings and models.

We thank the reviewer for the excellent suggestion. As pointed out by the reviewer, Reference #27 speculates that two amino acids that are specific to H2B.8 histone body (152th arginine and 179th methionine) might affect the DNA-nucleosome interactions (based on modelled structure of H2B.8-containing nucleosome), and substitution of lysine (K) by asparagine (N) at the monoubiquitylation site (234th amino acid) may affect the ability of H2B.8 to be ubiquitylated. To test the importance of these residues, we examined the functions of various mutated H2B.8 in *h2b.8* mutant sperm. First, we found that H2B.8 Δ IDR that harbors all these substitutions and only lacks the IDR, is unable to rescue the sperm nuclear size phenotype of *h2b.8* mutant, unlike the full-length H2B.8 (revised Fig. 2a). This demonstrates the importance of the IDR presence for H2B.8 function. Furthermore, to specifically test for the function of N234 at the monoubiquitylation site, we expressed H2B.8 with N234 replaced by K234, as in the canonical H2B. H2B.8-N234K fully complements the *h2b.8* sperm phenotype (Fig. 2a), demonstrating that N234 is not important for H2B.8 function. These results have been incorporated into Fig. 2a and added to the corresponding results section:

“We next expressed H2B.8 Δ IDR in *h2b.8* mutant plants (*pH2B.8::H2B.8 Δ IDR-eGFP h2b.8*), and examined whether sperm nuclear condensation is also dependent on the IDR. Unlike the full-length H2B.8, H2B.8 Δ IDR failed to rescue the sperm nuclear size phenotype of *h2b.8* (Fig. 2a, c), demonstrating the importance of the IDR for sperm nuclear condensation.

Outside the IDR, H2B.8 has several amino acid differences from canonical H2Bs in the globular domain, including asparagine 234 (N234), which is canonically a lysine (K) that is subject to monoubiquitylation²⁷ (Extended Data Fig. 3a). The inability of H2B.8 Δ IDR to rescue the *h2b.8* sperm nuclear phenotype (Fig. 2a) shows that without the IDR, the H2B.8 globular domain cannot mediate nuclear compaction. Nonetheless, as H2B monoubiquitylation is an important modification⁴¹ that would be precluded by N234, we examined whether N234 is important for H2B.8 function by expressing a mutated H2B.8 with the 234th asparagine replaced by lysine (H2B.8-N234K). The expression of H2B.8-N234K fully complemented the *h2b.8* sperm nuclear size phenotype (Fig. 2a), showing that N234 is not essential for H2B.8 function.” (Lines 178-192).

(ii) The authors suggest that H2B.8 condensates exclude heterochromatin condensates. However, a simple alternative model is one where H2B.8 prevents recruitment of heterochromatin factors through either specific PTMs or by recruiting factors that inhibit heterochromatin formation. Thus, indirect effects of H2B.8 also need to be considered rather than just direct effects. In this context, have the authors carried out IP-MS on H2B.8 to assess what factors it interacts with and/or whether it has PTMs differences from canonical H2B that could inhibit heterochromatin formation?

We actually performed H2B.8 IP-MS previously but the data quality was not ideal (perhaps due to the low-input nature of the experiment) and we did not find any potential interactors that could explain H2B.8's effect on heterochromatin. However, we agree with the reviewer that we cannot exclude alternative hypotheses. We have, therefore, softened our claims and added a discussion on alternative models:

“Collectively, our results suggest that condensation of chromatin via H2B.8 phase separation affects heterochromatin condensation, likely because H2B.8-associated AT-rich euchromatic TEs are interspersed with heterochromatic TEs in pericentromeric regions (Fig. 4e, Extended

Data Figs. 4a, and 6e, f). Alternative hypotheses that are independent of H2B.8 phase separation are also plausible, for example, H2B.8 might directly recruit factors that interfere with heterochromatin condensation.” (Lines 312-317).

5. On line 9-10, the authors say; “However, these heterochromatin foci are enlarged in the sperm in comparison to leaf cells (Fig. 1a, b), despite sperm, being haploid, having only half as much DNA as leaf cells. Such enlargement indicates a reduced level of heterochromatin condensation in sperm.” It is quite possible that the puncta are larger because a larger proportion of the genome is in heterochromatin. The authors need to test or provide evidence against this alternative possibility before making conclusions that there is a reduced level of heterochromatin condensation.

We thank the reviewer for the constructive comment. We first quantified the heterochromatin enlargement more precisely by measuring the total volume of H3K9me2 foci in a nucleus instead of individual foci, and directly comparing sperm H3K9me2 foci volume to that of its haploid mother cell, the generative cell (GC). As a result, we observed that heterochromatin foci occupy half of the volume in generative nucleus (GN) compared to diploid leaf nucleus, as expected (Fig. 1b). However, in sperm, heterochromatin foci are 27.5% larger than those in GN (Fig. 1b). These results show that enlargement of heterochromatin foci occurs in sperm cells after the division of GC. We have added these new results in and please refer to our reply to Reviewer 1 Comment 7 for a full description.

Next, we measured the H3K9me2 foci volume in *h2b.8* mutant sperm cells to test if the enlargement of heterochromatin foci in the sperm is caused by H2B.8. Indeed, we observed that *h2b.8* mutation reduced the total volume of H3K9me2 foci in the sperm, to a volume resembling that in the GN (Fig. 5e). This demonstrates that H2B.8 is responsible for the heterochromatin foci enlargement in the sperm.

Regarding the link between heterochromatin enlargement and decondensation, the strongest evidence is provided by our Hi-C analysis. First, we observed decreased interchromosomal interactions between pericentromeres in H2B.8 ectopically expressed seedlings compared to WT (Fig. 6a), demonstrating heterochromatin decondensation. Further, we observed that “the effect of H2B.8 is local, as the interactions between pericentromeric regions and chromosomal arms increase at regions with abundant H2B.8 (Fig. 6c, Extended Data Fig. 7f)” (Lines 338-340). “These observations support the hypothesis that dispersal of heterochromatin foci is caused by H2B.8-mediated aggregation of euchromatic TEs that are abundant in and near pericentromeric regions (Fig. 4e, Extended Data Figs. 4a, 6f)” (Lines 340-342).

Finally, we tested the alternative hypothesis that the H3K9me2 foci enlargement is caused by the incorporation of an increased proportion of the genome into these foci. We first performed Western blots on the leaf and sperm, which showed a reduced amount of H3K9me2 (in comparison to H3) in the sperm compared to leaf (new Extended Data Fig. 1a), suggesting that there is not an increased heterochromatin incorporation in the sperm. We next took advantage of the H2B.8 ectopic expression lines, in which H2B.8 effectively causes heterochromatin enlargement in somatic cells (Fig. 5f). H3K9me2 ChIP-seq of these lines shows no significant alteration of H3K9me2 profile (new Extended Data Fig. 6e), consistent with the idea that H2B.8 does not affect the amount of chromatin incorporated into heterochromatin.

Taken together, with the new results we obtained, we believe that we have strong evidence that H2B.8 causes the decondensation of heterochromatin. However, although unlikely, we agree that alternative hypotheses are still formally possible. Therefore, we discussed the alternative

hypothesis in corresponding results sections and softened our claim that H2B.8 decondenses heterochromatin in the revised manuscript:

“However, these heterochromatin foci appear enlarged in the sperm in comparison to leaf cells (Fig. 1a). To examine this further, we quantified the total volumes of H3K9me2-enriched heterochromatin foci in sperm and diploid leaf nuclei, as well as the nucleus of the generative cell, the haploid mother cell that divides into sperm cells. We found that the volume of the heterochromatin foci is reduced by half in the haploid generative nucleus compared to the diploid leaf nucleus, as expected (Fig. 1b). However, sperm heterochromatin foci are 27.5% larger than those of the generative nucleus (Fig. 1b), suggesting a reduced level of heterochromatin condensation, or an increased proportion of the genome incorporated into heterochromatin. The latter is less likely, as less H3K9me2 (compared to total H3) was detected in sperm cells than leaf cells (Extended Data Fig. 1a).” (Lines 81-91).

“Collectively, our results suggest that condensation of chromatin via H2B.8 phase separation affects heterochromatin condensation, likely because H2B.8-associated AT-rich euchromatic TEs are interspersed with heterochromatic TEs in pericentromeric regions (Fig. 4e, Extended Data Figs. 4a, and 6e, f).” (Lines 312-315).

Response to Referee #3

The authors of this manuscript entitled “A histone variant condenses flowering plant sperm via chromatin phase separation” report that the histone H2B variant H2B.8 mediates chromatin compaction in sperm by phase separation. H2B.8 was previously shown to be an angiosperm-specific histone variant, but the functional role remained elusive. The authors convincingly show that the presence of the N-terminally-located intrinsically disordered region confers the ability to H2B.8 to undergo phase separation and to form condensates. This special property most likely allows angiosperms to compact their sperm genome in the absence of protamines while maintaining transcriptional competency. These are very interesting findings worth to be reported.

We thank the reviewer for the positive evaluation.

*One aspect that would strongly add to this manuscript are data demonstrating the functional relevance of H2B.8. If chromatin compaction mediated by H2B.8 is relevant, decrease of sperm fitness would be expected. This could be tested by e.g. sperm competition experiments of *h2b.8* mutant and wild-type sperm. But also without this data, this is a very nice, carefully done and important study, adding to our knowledge the mechanistic basis of reproductive traits.*

We highly appreciate the reviewer's positive and helpful comment. We measured the male and female transmission of the *h2b.8* mutant allele versus WT accordingly and found that, indeed, *h2b.8* mutation reduces male transmission by 27% (Supplementary Table 3). We also measured *in vitro* pollen germination of WT and *h2b.8* mutant. Consistent with our finding that H2B.8 does not affect transcription, we found *in vitro* pollen germination is not affected by the *h2b.8* mutation (Supplementary Table 4). We have added a new results section describing these results (please refer to this for a full description). We have also added this new result into the abstract, introduction and discussion.

Abstract (Lines 22-24): “Reciprocal crosses show that *h2b.8* mutation reduces male transmission, suggesting that H2B.8-mediated sperm compaction is important for fertility.”

Introduction (Lines 65-66): “H2B.8-induced nuclear compaction is important for fertility, as *h2b.8* mutation reduces male transmission.”

Results (Lines 271-294): “H2B.8 promotes male fertility

To investigate the biological significance of H2B.8-mediated sperm condensation, we performed reciprocal crosses between the *h2b.8* heterozygous mutant and wild type. When *h2b.8* heterozygous mutant was used as the male, F1 progeny are 27.3% less likely to carry the *h2b.8* allele than the wild-type allele ($P < 0.01$, Fisher's exact test, $N = 575$; Supplementary Table 3), whereas the transmission of *h2b.8* is not significantly different from the wild-type allele when passed through the female ($P = 0.64$, Fisher's exact test, $N = 574$; Supplementary Table 3), demonstrating that H2B.8 is important for male fertility. The fertility defect of *h2b.8* is not caused by disrupted pollen germination, as *h2b.8* pollen grains germinate at comparable rates to wild type *in vitro* (Supplementary Table 4). This result is consistent with the null effect of H2B.8 on transcription (Figs. 5a, b, Extended Data Figs. 6a-d), suggesting the fertility defect is most likely caused by enlarged sperm nuclei.

Manual crossing is a stressful process, with pistils (the female organs) slightly desiccated and pollinated at an earlier developmental stage than normal. To investigate if *h2b.8* affects male

fertility in a less stressful mating environment, we examined the segregation ratio of progeny generated from self-pollinated *h2b.8* heterozygous plants. We observed the expected Mendelian segregation ratio (N = 1462; Supplementary Table 5), showing that the *h2b.8* mutation does not affect fertility when plants are allowed to self-fertilize under laboratory conditions. As H2B.8 is transiently expressed in mature seeds (Extended Data Fig. 1d), this result also demonstrates that *h2b.8* does not affect seed development under laboratory conditions. Collectively, our observations suggest that H2B.8 is important for sperm fertility in challenging or stressful situations, such as those created by manual crossing, which might be relevant to reproduction under natural environmental conditions that are less favorable than standard laboratory conditions.”

Discussion (Lines 366-370): “This moderate level of condensation is still important for fertility, as *h2b.8* mutation can significantly reduce male transmission (Supplementary Table 3). As sperm transcription (Fig. 5a, Extended Data Fig. 6a) and *in vitro* pollen germination (Supplementary Table 4) are not affected by the *h2b.8* mutation, the fertility reduction is likely caused by the enlarged sperm nuclei.”

Other comments

1. The authors state: „H2B.8 condensates are largely devoid of H3K9me2 and distinct from heterochromatin foci (Fig. 5e).” This requires quantitative validation. Based on the provided pictures I am not entirely convinced that this statement is correct. In fact, the conclusion of the authors that H2B.8 causes heterochromatin foci to decondense, would be in agreement with a co-localization of H3K9me2 and H2B.8 condensates. The interaction between H2B.8-induced heterochromatin and H3K9me2 would be most convincingly demonstrated by H3K9me2 ChIP in H2B.8 overexpressing lines.

We thank the reviewer for raising this point. We have accordingly quantified the overlap between H2B.8 condensates and H3K9me2-enriched heterochromatin foci. The result validates our statement and shows that H2B.8 condensates only overlaps 19.0% of the volume of heterochromatin foci (described in the legend of Fig. 5g). This result is also consistent with our statement that “although H2B.8 and heterochromatic condensates are mostly distinct, some physical associations are observed” (Lines 310-311).

Besides the cytological evidence, our H2B.8 ChIP-seq data from sperm (Fig. 4b, c) and ectopically expressed seedlings (Fig. 4c, Extended Data Fig. 4c) also demonstrate that H2B.8 generally avoids H3K9me2-enriched heterochromatin. To further exclude the possibility that H2B.8 ectopic expression induces H3K9me2, we performed H3K9me2 ChIP-seq on H2B.8 ectopic expression lines and WT as suggested by the reviewer. We found that H3K9me2 localization is not affected by H2B.8 ectopic expression (new Extended Data Fig. 6e), further demonstrating that H2B.8 does not induce H3K9me2.

Our Hi-C experiment shows that the decondensation of heterochromatin caused by H2B.8 is likely caused by the presence of H2B.8-enriched euchromatic TEs near pericentromeric chromatin (Figs. 4e, 6c, Extended Data Figs. 4a). For a detailed explanation, please refer to our reply to Reviewer 2 Comment 5. However, we agree that this point should be strengthened to avoid confusion. Therefore, we have performed an additional analysis that demonstrates H2B.8 is enriched at the periphery of heterochromatic regions (new Extended Data Fig. 6f). The original sentence quoted by the reviewer is also deleted to avoid confusion.

These new results are added to the Fig. 5g legend as “H2B.8 condensates are largely distinct from heterochromatin foci, overlapping only $19.0\% \pm 10.7\%$ (standard deviation calculated from 53 nuclei) of the volume of heterochromatin foci” (Lines 458-460), and described in the revised text as “our results suggest that condensation of chromatin via H2B.8 phase separation affects heterochromatin condensation, likely because H2B.8-associated AT-rich euchromatic TEs are interspersed with heterochromatic TEs in pericentromeric regions (Fig. 4e, Extended Data Figs. 4a, and 6e, f)” (Lines 312-315).

2. The claim of increased short range and depleted long-range contacts is not obvious based on extended Data Fig. 6d. As shown by the authors, also long-range interactions correlate with abundance of H2B.8, similar to short range interactions (Fig. 6b,c), making the proposed scenario unlikely and inconsistent with the statement “The effect of H2B.8 is local, as the interactions between pericentromeric regions and chromosomal arms increase at regions with abundant H2B.8 (Fig. 6c).”

We understand the reviewer’s point and agree that the current presentation of Hi-C data is confusing. First, the spatial contact frequency, presented in the previous Extended Data Fig. 6d, generally adheres the exponential attenuation in relation to genomic distances, and as such, such plots generally do not display drastic changes (similarly ‘subtle’ changes are shown in, for example, PMID: 32312999, Fig. 3c; and PMID: 34625553, Fig. 7c). Second, the decreased long-range contacts are likely driven by the increased short-range contacts, given that the sum of the contacts is assumed constant in Hi-C analysis. Third, indeed the effect of H2B.8 is local, as well demonstrated in Fig. 6b, c (and the newly added Extended Data Fig. 7e, f). However, as the increased long-range interactions between pericentromeric regions and chromosomal arms typically occur over 10 Mb distance, it is beyond the scale of Extended Data Fig. 6d and not illustrated. Given the difficulty in interpreting Extended Data Fig. 6d, and that the major point that short-range interactions are generally increased in H2B.8 ectopic expression lines is well illustrated in Fig. 6a (and the newly added Extended Data Fig. 7d), we have removed Extended Data Fig. 6d and revised the following sentence to avoid confusion:

“Within chromosomes, ectopic H2B.8 caused increased short-range contacts (200 kb – 1.1 Mb) and depleted long range contacts ($> 1.1\text{mb}$) (Fig. 6a, Extended Data Fig. 7d).” (Lines 327-328).

3. I found this statement rather counterintuitive: “However, the localization of H2B.8 in the non-transcribing parts of the genome suggests that it may not adversely affect gene expression.” If H2B.8 localizes in the non-transcribed part of the genome, one would expect that it negatively affects transcription. It has been nicely shown by the authors that this is not the case, nevertheless, it cannot be predicted based on the localization of H2B.8.

We agree and thank the reviewer for pointing out this logical inconsistency. We have correspondingly revised the text. It now reads “the localization of H2B.8 in the non-transcribing parts of the genome could arise via two mechanisms. Either H2B.8 suppresses transcription, or it is excluded from transcribed regions. To test these hypotheses, ... These results demonstrate that H2B.8 does not inhibit gene transcription, but is instead evicted by transcription, which likely underlies the localization of H2B.8 in unexpressed euchromatin.” (Lines 252-270).

Reviewer Reports on the First Revision:

Referees' comments:

Referee #1 (Remarks to the Author):

The authors of the manuscript "A histone variant condenses flowering plant sperm via chromatin phase separation" performed an extensive revision of the paper and answered all comments. Overall, their model did not change with the revisions, and their conclusions were substantially strengthened by the new experiments. We find this a very interesting and well supported story that advances our understanding of mechanisms of chromatin condensation. We strongly recommend the publication of this paper upon addressing the following statistical comment.

As we suggested in comment n. 10, the authors generated a multivariate linear regression model to show that GC content and transcription are the best predictors of H2B.8 localization. There are some problems with the way this analysis was performed that should be addressed. We highlight below some issues and provide suggestions, but we encourage the authors to get a consult on how to best perform this analysis.

- First, the authors should write more detail in the methods sections about how they prepared the data for the model, why they decided to generate two different models instead of just one, and any additional information that can be helpful.

- For instance, how was the CG methylation level per bin calculated?

- Is there a reason for using 50-bp genomic windows? This seems like a very small window compared to the resolution of ChIP-seq data, therefore ~200-bp windows might be more appropriate. Moreover, we expect that several 50-bp windows don't even have any CG site and therefore the CG methylation level cannot be calculated there. How were these bins handled? If they were removed, this could strongly bias the result and doesn't seem appropriate.

- It seems like the authors used all 50-bp bins in both models? The individual data points need to be independent observations, therefore the analysis should be performed on bins that don't originate from the same ChIP DNA fragment. For instance, you could take one bin every 1 kb or 2 kb.

- The authors should include the p-values in the table.

- While this analysis is acceptable, a better way to understand which variables contribute to H2B.8 localization might be to compare the Adjusted R² of different models including either GC content alone, GC content + TPM, GC content + ... similarly to what was done in Fig. 4c of Morselli at al., *Elife*, 2015 (PMID: 25848745). This might show that including all variables does not significantly improve the predictive power of the model compared to GC content + expression, but expression improves the model compared to GC content alone. If the authors find this appropriate, the Akaike information criterion (AIC) can also be used to compare models and is easy to calculate in R. The authors can also consider using log(TPM) instead of splitting the genes in >5 TPM, 1 to 5 TPM, and <1

TPM.

Referee #2 (Remarks to the Author):

The authors have satisfactorily addressed my other concerns, but my concern remains about the link to phase-separation, which is the prominent conclusion as in the title. The authors have improved their original *in vitro* assay by now carrying it out in the context of chromatin. However, I am still not convinced they are observing bona-fide liquid or gel-like phase separation as opposed to a solid precipitation/aggregation for the reasons described below.

1. The authors say “However, without Mg²⁺, H2B.8-containing nucleosome arrays can still form phase-separated condensates, whereas H2B.2-containing nucleosome arrays remain homogenous (Fig. 3b). H2B.8-containing chromatin condensates are insensitive to salt, as they do not grow when salt is introduced into the solution (Fig. 3b).”

The H2B.8-containing arrays form condensates that are non-spherical, and the observation of a complete lack of salt dependence rather than having a different salt dependence is actually a bit concerning and suggestive of solid aggregates or precipitates.

In my original review I had suggested “at a minimum, the authors need to test if chromatin reconstituted with canonical plant H2B vs. H2B.8 shows different phase-separation properties using the types of assays used in PMID: 31543265.” These assays include FRAP to assess liquid vs gel-like behavior. The simpler option to FRAP is a droplet fusion, which is also carried out in PMID: 31543265. In their first submission the authors showed FRAP and droplet fusion data using fluorescently labelled histones alone (Ext. Data Fig. 3d,e). This time the authors don’t carry out either type of experiment. And further they seem to use DAPI, a DNA intercalating dye, to label the chromatin, which may alter its properties.

If fluorescent labeling of histones is no longer possible, the authors can still use brightfield microscopy to visualize the chromatin condensates without labeling with DAPI. They can also use brightfield microscopy to observe whether the condensates can fuse. They should do this for both the H2B.2-containing array condensates as a positive control (i.e. these should show fusion as seen in PMID: 31543265) and the H2B.8-containing array condensates.

2. The reaction conditions used for the array condensate assay are far from physiological, and this may be why the authors are observing non-spherical condensates and lack of any salt dependence. I had suggested PMID: 31543265 as a guide to the authors because this work uses some of the best practices in the phase-separation field. Below are my suggestions.

- To mimic physiological salt concentrations, monovalent salt conditions need to be tested like in PMID: 31543265. So, in addition to the effects of Mg²⁺, they should also test the effects of physiological NaCl, KCl, or KOAc concentrations (~150 mM) on the properties of both H2B.2 and H2B.8 arrays.

- The authors mention an overnight incubation at 4 °C before visualizing the condensates. This does not seem physiological. To assess the biological relevance the experiments should be carried out at either room temperature or a temperature that matches the cellular system. Additionally, the typical time (similar to that used in PMID: 31543265) before visualization is 30 min or less, not overnight. This is because with long times in the cold, the properties of condensates can change due to maturation and hardening of the phases.

- The authors should test under the physiological conditions mentioned above the role of the IDR of H2B.8 in condensate formation.

3. In the phase-separation conditions without salt, I am assuming there is no NaCl in the buffer. However, in the methods it says "All in vitro experiments were performed in phase separation reaction buffer (20 mM Tris pH 7.5, 100 mM NaCl)." Perhaps this is a typo. If so, the authors should correct it and clarify which exact final buffer conditions are used for which phase-separation experiment. Also do the authors use MgOAc as in PMID: 31543265 or do they use MgCl₂? Please clarify.

Referee #3 (Remarks to the Author):

The authors made substantial efforts to address my concerns and included many new data to validate their hypotheses. I am, however, skeptical about the newly generated data suggesting an effect of the H2B.8 mutant on male fertility. Essentially, transmission of the H2B.8 allele is reduced by 8%, which is a minor effect that is only observed after manual pollination and that could well be within the range of biological variation. I also find the argument that "the fertility reduction is likely caused by the enlarged sperm nuclei" highly speculative. Without providing data revealing the cause for the slight transmission defect, there is no solid support for this statement. I therefore recommend toning down the statements regarding the effect of H2B.8 on fertility.

Author Rebuttals to First Revision:

We thank the reviewers for the constructive comments. We have accordingly performed new analyses and experiments and revised the manuscript. Below we respond to the specific comments of each reviewer, with the comments reproduced in italics.

Response to Referee #1:

The authors of the manuscript “A histone variant condenses flowering plant sperm via chromatin phase separation” performed an extensive revision of the paper and answered all comments. Overall, their model did not change with the revisions, and their conclusions were substantially strengthened by the new experiments. We find this a very interesting and well supported story that advances our understanding of mechanisms of chromatin condensation. We strongly recommend the publication of this paper upon addressing the following statistical comment.

We thank the reviewer for the very positive comments.

As we suggested in comment n. 10, the authors generated a multivariate linear regression model to show that GC content and transcription are the best predictors of H2B.8 localization. There are some problems with the way this analysis was performed that should be addressed. We highlight below some issues and provide suggestions, but we encourage the authors to get a consult on how to best perform this analysis.

- First, the authors should write more detail in the methods sections about how they prepared the data for the model, why they decided to generate two different models instead of just one, and any additional information that can be helpful.

We thank the reviewer for the constructive suggestion and have revised the corresponding methods section accordingly (Lines 975-987).

The reason why we generated two separate models for various chromatin features (eg. histone variants and modifications, mCG and GC content) and genomic region categories (eg. intergenic, euTE, hetTE, genes) instead of a combined one is because we think it shows the association between H2B.8 and genomic features better and independently, given the strong and complex associations between chromatin features and genomic region categories. For example, gene regions are typically associated with transcription and H3K4me3, which are low in TEs; euTE and hetTE regions are associated with low and high CG content, respectively. In a separate model with only genomic region categories, it is clear that expressed genes have the strongest negative association, while euTE and intergenic regions have the strongest positive association. However, when combined with other chromatin feature variables, their regression weights are affected by the presence of other chromatin variables (figure 1).

	gene TPM>5	gene TPM 1-5	gene TPM 0-1	euTE	hetTE	intergenic
Regression weight (MLR2: categorial vars only)	-0.933	-0.791	-0.097	0.613	-0.190	0.666
Regression weight (all chromatin and categorial vars)	-0.341	-0.344	-0.096	0.307	0.050	0.175

figure 1. Regression weights of categorical variables in a multivariate linear regression model (MLR) of H2B.8 using all chromatin and categorical variables listed in Supplementary Table 2, compared to those in a model using only categorical variables (MLR2; Supplementary Table 2).

Further, because of the strong associations between categorical variables and chromatin variables, the addition of categorical variables to the model does not significantly improve the predictive power (figure 2).

figure 2. Adjusted R square values of different MLR models. MLR1 and MLR2 (Supplementary Table 2) were built using chromatin and genomic categorical variables, respectively. The third model was built using all variables in MLR1 and MLR2.

- For instance, how was the CG methylation level per bin calculated?

We used 50-bp bins to build the model, and the CG methylation in each bin was calculated using the number of sequenced C in the CG context within the bin divided by the number of (C+T) in the CG context. In a bin, if there are no CG sites or no sequencing reads, the bin will have no value.

- Is there a reason for using 50-bp genomic windows? This seems like a very small window compared to the resolution of ChIP-seq data, therefore ~200-bp windows might be more appropriate. Moreover, we expect that several 50-bp windows don't even have any CG site and therefore the CG methylation level cannot be calculated there. How were these bins handled? If they were removed, this could strongly bias the result and doesn't seem appropriate.

We used 50-bp genomic windows due to a) convenience as most of our sequencing data were processed using this bin size, and b) this gives us a large number of bins to assess the predictive power of the models. We agree that these are too small for the resolution of ChIP-seq data. There are indeed windows that have no CG sites (or no aligned sequencing reads, as mentioned in the previous point). These windows have no value, which were omitted during the modelling by default in the lm function of R. We agree with the reviewer that this would introduce bias. Therefore, we built the new models using 200-bp bins as suggested (shown in revised Supplementary Table 2 and Extended Data Fig. 4e). The results show that among the single variant models, transcription (H3K4me3, log₁₀(RPKM+1)) and GC content have the strongest predictive power. How the new results using 200-bp bins compare to previous results using 50-bp bins is shown in figure 3.

figure 3. Adjusted R square of single variable H2B.8 models using data parsed into 50-bp (left) or 200-bp bins (right).

- It seems like the authors used all 50-bp bins in both models? The individual data points need to be independent observations, therefore the analysis should be performed on bins that don't originate from the same ChIP DNA fragment. For instance, you could take one bin every 1 kb or 2 kb.

Yes, we originally used 50-bp bins in both models. As suggested by the reviewer, we have changed to 200-bp bins to run the new models. We do not think bins sharing the same ChIP DNA fragment would significantly affect the independence between bin values, given that each bin value is derived from a large number of independent fragments in good-quality ChIP-seq data and duplicated reads were removed during data processing. The increase of bin size suggested by the reviewer also helps with this matter. Indeed, we observed almost no difference in the models using all 200-bp bins (figure 3), or 200-bp bins that skip every 1 Kb or 2 Kb (figure 4).

figure 4. Adjusted R square of single variable H2B.8 models using data parsed into 200-bp bins that skip every 1 Kb (left) or 2 Kb (right). Models using data from all 200-bp bins are shown in figure 3.

- The authors should include the p-values in the table.

We apologize for this neglect. All p-values are <0.001 (F-statistic test), which is included in the table legend.

- While this analysis is acceptable, a better way to understand which variables contribute to H2B.8 localization might be to compare the Adjusted R² of different models including either GC content alone, GC content + TPM, GC content + ... similarly to what was done in Fig. 4c of Morselli et al., Elife, 2015 (PMID: 25848745). This might show that including all variables does not significantly improve the predictive power of the model compared to GC content + expression, but expression improves the model compared to GC content alone. If the authors find this appropriate, the Akaike information criterion (AIC) can also be used to compare models and is easy to calculate in R. The authors can also consider using log(TPM) instead of splitting the genes in >5 TPM, 1 to 5 TPM, and <1 TPM.

We thank the reviewer for the helpful suggestion. We have accordingly examined the predictive powers of our models and how they are affected by the addition of variables.

First, we added the Adjusted R² values into the revised Supplementary Table 2 (also shown in Extended Data Fig. 4e). It shows that indeed GC content and transcription (and transcription-related histone mark H3K4me3) are the best predictors of H2B.8. However, transcription has better predictive power than GC content. A careful look into this revealed that transcription predicts well in genic regions (occupying 50% of the genome), whereas GC content does a better job than transcription in TEs and intergenic regions, probably because these regions are generally not transcribed.

Second, we found that GC content and transcription ($\log_{10}(\text{RPKM}+1)$ or H3K4me3) improve the predictive power of the model compared to GC content, or transcription alone (figure 5). However, with the addition of more variables, the prediction power slowly increases (from ~ 0.4 using GC content and transcription, to 0.57 using all 14 variables; figure 5). First of all, this is somewhat expected, as unless H2B.8 abundance is solely dependent on one or two factors that are completely independent of other variables, adding more variables would improve prediction. Given the variations among genomic data, even simply adding independent measurements of the same chromatin feature would improve prediction. Second, we believe this result suggests that H2B.8 localization is under complex chromatin regulation rather than solely dependent on GC content and transcription. For example, it is very likely that H2B.8 is driven by chromatin features that associate with low GC content, instead of directly determined by GC content.

Based on these results, we revised the main text to 1) reflect the relative importance of transcription and GC content in predicting H2B.8, and 2) soften our claims on the determination of H2B.8 localization. We have also added an abridged version of figure 5 to the manuscript (new Extended Data Fig. 4e).

Lines 240-248: “Multivariate linear regression modelling of H2B.8 further showed that transcription and GC content are the best predictors of H2B.8 localization, with which strong anti-correlations exist (Fig. 4d, g, Extended Data Fig. 4d, e, Supplementary Table 2). This likely explains why H2B.8 is strongly depleted from transcribed genes (Fig. 4b, Extended Data Fig. 4c). In the rest of the genome, where there is little transcription, H2B.8 accumulates at GC-poor elements, mostly euchromatic TEs and intergenic regions (Extended Data Fig. 4d, f). In all, our results suggest that H2B.8 localization is mostly driven by transcription and GC content rather than sperm-specific factors.”

Line 249: “It is yet unclear why H2B.8 localization is associated with transcription and GC content ...”

figure 5. Adjusted R square (left y axis) and AIC (right y axis) scores of H2B.8 models built using indicated variables.

Response to Referee #2:

The authors have satisfactorily addressed my other concerns, but my concern remains about the link to phase-separation, which is the prominent conclusion as in the title. The authors have improved their original in vitro assay by now carrying it out in the context of chromatin. However, I am still not convinced they are observing bona-fide liquid or gel-like phase separation as opposed to a solid precipitation/aggregation for the reasons described below.

1. The authors say “However, without Mg²⁺, H2B.8-containing nucleosome arrays can still form phase-separated condensates, whereas H2B.2-containing nucleosome arrays remain homogenous (Fig. 3b). H2B.8-containing chromatin condensates are insensitive to salt, as they do not grow when salt is introduced into the solution (Fig. 3b).”

The H2B.8-containing arrays form condensates that are non-spherical, and the observation of a complete lack of salt dependence rather than having a different salt dependence is actually a bit concerning and suggestive of solid aggregates or precipitates.

In my original review I had suggested “at a minimum, the authors need to test if chromatin reconstituted with canonical plant H2B vs. H2B.8 shows different phase-separation properties using the types of assays used in PMID: 31543265.” These assays include FRAP to assess liquid vs gel-like behavior. The simpler option to FRAP is a droplet fusion, which is also carried out in PMID: 31543265. In their first submission the authors showed FRAP and droplet fusion data using fluorescently labelled histones alone (Ext. Data Fig. 3d,e). This time the authors don’t carry out either type of experiment. And further they seem to use DAPI, a DNA intercalating dye, to label the chromatin, which may alter its properties.

If fluorescent labeling of histones is no longer possible, the authors can still use brightfield microscopy to visualize the chromatin condensates without labeling with DAPI. They can also use brightfield microscopy to observe whether the condensates can fuse. They should do this for both the H2B.2-containing array condensates as a positive control (i.e. these should show fusion as seen in PMID: 31543265) and the H2B.8-containing array condensates.

We thank the reviewer for the constructive comments and suggestions, based on which we have performed a series of new experiments that significantly improved the manuscript.

Previously we performed the chromatin phase separation assays following the experimental conditions described by a paper that followed up on PMID: 31543265, PMID: 33326747. In PMID: 33326747, the authors found chromatin behaves as solid-/gel-like phase-separated condensates in solutions that do not contain DTT, BSA and acetate anions. This was exactly what we observed - the condensates were solid-/gel-like.

As suggested by the reviewer, we performed new *in vitro* phase separation assays using the conditions described in PMID: 31543265 (ie. using PEGylated and BSA passivated 384-well plates and a buffer that contains Tris·OAc, DTT and BSA). We observed the formation of spherical droplets for both H2B.2- and H2B.8-containing nucleosomal arrays (as well as for those reconstituted with H2B.8 H2B.8ΔIDR and H2B.8-scrambledIDR) (revised Fig. 3b), demonstrating that H2B.8 chromatin forms *bona fide* phase-separated condensates.

We understand the reviewer’s concern regarding the complete salt independence of H2B.8 chromatin phase separation, as chromatin has only been known to phase separate *in vitro* in the

presence of cations (PMID: 31543265; PMID: 33326747). Therefore, we have carefully examined the phase separation properties of H2B.8 (and H2B.2, H2B.8 Δ IDR and H2B.8-scrambledIDR) nucleosome arrays under different salt and array concentrations (displayed in Fig. 3b and Extended Data Fig. 3f, g). We found that H2B.8 arrays form phase-separated droplets at lower array concentrations than H2B.2 (and H2B.8 Δ IDR; Extended Data Fig. 3f) in physiological salt conditions, indicating stronger phase separation ability. Importantly, H2B.8 arrays can form phase-separated droplets at lower salt concentrations than H2B.2 (and H2B.8 Δ IDR) and even in the absence of salt (or at the presence of high chelating agent EDTA) (Fig. 3b and Extended Data Fig. 3g). These new results demonstrate that the IDR of H2B.8 endows a novel phase-separation property to the chromatin, which is independent of cations.

According to the reviewer's suggestions, we labelled histones with fluorophore and performed FRAP and droplet fusion experiments as described in PMID: 31543265, including the usage of YOYO-1 Iodide (491/509) instead of DAPI. In brief, FRAP recovery and droplet fusion were observed in H2B.8 (and H2B.2) reconstituted chromatin (revised Extended Data Fig. 3d, e), demonstrating the dynamic nature of H2B.8 chromatin condensates.

We have replaced the old Fig. 3b and added new Extended Data Fig. d-g to display these new results. The corresponding results section has been revised as follows:

Lines 147-180: "To test this hypothesis, we assembled nucleosomal arrays using recombinant fluorophore-labeled histone octamers and a DNA template containing 12 repeats of the 601 nucleosome positioning sequence, and tested the phase separation properties of these nucleosomal arrays as previously described³⁸ (Fig. 3b, Extended Data Fig. 3c). We found that the addition of cation (K^+ or/and Mg^{2+}) at physiologically relevant concentrations to chromatin reconstituted with either H2B.8 or a canonical H2B (H2B.2) that naturally lacks an IDR induces the formation of phase-separated droplets (Fig. 3b), consistent with the previous finding that chromatin undergoes phase separation *in vitro* at physiological salt conditions^{38,39}. Consistent with the reported liquid-like property of chromatin condensates³⁸, H2B.8- and H2B.2-containing chromatin droplets show fluorescence recovery after photobleaching (FRAP; Extended Data Fig. 3d) and droplet fusion upon contact (Extended Data Fig. 3e).

FRAP recovery of H2B.8 chromatin droplets is slower than that of H2B.2 (Extended Data Fig. 3d), reflecting reduced internal droplet dynamics and a more gel-like behavior. To further test if H2B.8 confers different phase separation properties to chromatin, we examined H2B.8- and H2B.2-containing nucleosomal arrays under different salt and array concentrations. We found that under the same physiological salt concentration, H2B.8-containing nucleosome arrays can form phase-separated condensates at lower chromatin concentrations, indicating stronger phase separation ability (Extended Data Fig. 3f). Further, unlike typical chromatin phase separation that requires the assistance of physiological cation^{38,39} (eg. H2B.2-containing chromatin; Figure 3b, Extended Data Fig. 3f,g), H2B.8-containing nucleosome arrays can form phase-separated droplets under low or no cation conditions (or in the presence of high concentration of chelating agent EDTA; Figure 3b, Extended Data Fig. 3g). To test if this phase separation property of H2B.8-containing chromatin relies on the IDR, we reconstituted nucleosomal arrays using H2B.8 without the IDR (H2B.8 Δ IDR) (Extended Data Fig. 3c). Resembling H2B.2-containing chromatin but distinct from H2B.8-containing chromatin, H2B.8 Δ IDR-containing nucleosome arrays cannot phase separate at lower array concentrations (<50 nM; Extended Data Fig. 3f) and also fail to undergo phase separation in the absence of salt (or under low salt or high EDTA conditions; Figure 3b, Extended Data Fig. 3f,g), showing that the IDR is critical for the strong and cation-independent chromatin phase separation. Consistent with

the idea that IDRs promote phase separation via the disordered state instead of specific sequence motifs^{29,40}, we found chromatin containing H2B.8 with the IDR sequence randomly scrambled (H2B.8-scrambledIDR) is able to phase separate in a salt-independent manner (Fig. 3b). Taken together, our results demonstrate that the IDR of H2B.8 mediates a novel form of chromatin phase separation.”

2. The reaction conditions used for the array condensate assay are far from physiological, and this may be why the authors are observing non-spherical condensates and lack of any salt dependence. I had suggested PMID: 31543265 as a guide to the authors because this work uses some of the best practices in the phase-separation field. Below are my suggestions.

- To mimic physiological salt concentrations, monovalent salt conditions need to be tested like in PMID: 31543265. So, in addition to the effects of Mg²⁺, they should also test the effects of physiological NaCl, KCl, or KOAc concentrations (~150 mM) on the properties of both H2B.2 and H2B.8 arrays.

According to the reviewer’s suggestion and the protocol described in PMID: 31543265, we performed phase separation assays under different salt concentrations (including the physiological concentrations the reviewer suggested), and tested the effects of both Mg²⁺ and K⁺. These are shown in Fig. 3b and Extended Data Figure 3f, g, and described in the results section as quoted above.

- The authors mention an overnight incubation at 4 °C before visualizing the condensates. This does not seem physiological. To assess the biological relevance the experiments should be carried out at either room temperature or a temperature that matches the cellular system. Additionally, the typical time (similar to that used in PMID: 31543265) before visualization is 30 min or less, not overnight. This is because with long times in the cold, the properties of condensates can change due to maturation and hardening of the phases.

We thank the reviewer for the suggestion. Accordingly in the new phase separation assays, we performed the incubation at room temperature for 30 minutes instead (described in the methods section at Line 870).

- The authors should test under the physiological conditions mentioned above the role of the IDR of H2B.8 in condensate formation.

Yes, nucleosomal arrays reconstituted with H2B.8ΔIDR were included in the experiments mentioned above (Fig. 3b and Extended Data Fig. 3f, g).

3. In the phase-separation conditions without salt, I am assuming there is no NaCl in the buffer. However, in the methods it says “All in vitro experiments were performed in phase separation reaction buffer (20 mM Tris pH 7.5, 100 mM NaCl).” Perhaps this is a typo. If so, the authors should correct it and clarify which exact final buffer conditions are used for which phase-separation experiment. Also do the authors use MgOAc as in PMID: 31543265 or do they use MgCl₂? Please clarify.

Yes, this was a mistake. This statement refers to the conditions under which we performed the initial phase separation assays using just H2B.8 and DNA (included in the initial submission). During our first revision, we used reconstituted nucleosomal arrays instead but forgot to delete

this sentence from the methods, hence causing the confusion. The buffer conditions we used in our revised experiments are described at Lines 865-869 in the methods section.

We used $\text{Mg}(\text{OAc})_2$, which is now labelled in the corresponding figures (Fig. 3b and Extended Data Fig. 3f, g).

Response to Referee #3:

The authors made substantial efforts to address my concerns and included many new data to validate their hypotheses. I am, however, skeptical about the newly generated data suggesting an effect of the H2B.8 mutant on male fertility. Essentially, transmission of the H2B.8 allele is reduced by 8%, which is a minor effect that is only observed after manual pollination and that could well be within the range of biological variation. I also find the argument that "the fertility reduction is likely caused by the enlarged sperm nuclei" highly speculative. Without providing data revealing the cause for the slight transmission defect, there is no solid support for this statement. I therefore recommend toning down the statements regarding the effect of H2B.8 on fertility.

We appreciate the positive evaluation of the reviewer.

Regarding the fertility defect, we agree that at a glance, a male transmission at 42% does seem insignificant. However, first, 42% male transmission (Supplementary Table 3) does not mean 8% reduction, as the 42% (*h2b.8/+* genotype) should be compared with 58% (+/+ genotype) instead of 50%. Let's say out of 100 F1s, 42 and 58 individuals are carrying the *h2b.8* mutant and WT alleles, respectively. This means that pollen carrying the *h2b.8* mutant allele is 16/58 (27.6%) less likely to fertilize a WT egg than pollen carrying a WT *H2B.8* allele. Another way of saying this is that the mutant allele is $42/58 = 0.724$ as frequent as the WT allele.

Second, we were also initially worried about the small fertility defect observed after manual pollination, as the reviewer rightly pointed out. Therefore, we have performed two independent experiments at different times by different lab members (T.B. the first time, and S.Z. the second time). These experiments gave consistent results (the *h2b.8* mutant allele is 0.73 and 0.70 as frequent as the WT allele, with $P = 0.0078$ and 0.0009, respectively), and therefore, we trust and included the fertility data in the manuscript. We have now included data from both experiments in the revised Supplementary Table 3 and described the nature of the two independent experiments in the table legend and methods (Lines 1035-1043). We thank the reviewer for raising this important point, which improved our manuscript.

We agree that it is highly speculative to say that the reduced fertility was caused by enlarged sperm nuclei. This is why we used two suggestive qualifiers in the sentence "suggesting the fertility defect is most likely caused by enlarged sperm nuclei" (Lines 294-295). We also mentioned that this defect seems to be caused by imperfect conditions "our observations suggest that H2B.8 is important for sperm fertility in challenging or stressful situations, such as those created by manual crossing, which might be relevant to reproduction under natural environmental conditions that are less favorable than standard laboratory conditions" (Lines 304-307). We re-examined all mentioning of the fertility effect in the manuscript, and believe they all have suggestive qualifiers and are justified given the robustness of our data.

Reviewer Reports on the Second Revision:

Referees' comments:

Referee #1 (Remarks to the Author):

The authors have addressed all of my comments. This is a great paper. Steve Jacobsen

Referee #2 (Remarks to the Author):

The authors have done an excellent job of addressing my remaining questions. The new data with varying salt and Mg²⁺ conditions, droplet fusion and FRAP clearly show a physiologically relevant difference between the H2B.2 and H2B.8 containing chromatin phases. The authors need to include the specific buffer conditions used for the FRAP and droplet fusion studies (Extended Data Fig. 3d&e) in the methods. Other than this required small addition, I have no further concerns. This study should be of much interest to both the chromatin and phase-separation communities.

Referee #3 (Remarks to the Author):

The authors supported the observed transmission defect by adding additional experimental data. While the defect remains small and is only detectable under artificial settings, it nevertheless is significant and possibly biological relevant. I have no further comments to be addressed.

We thank all referees for their positive evaluation. Below we reproduce all referees' comments in italics and respond to the specific comment of Referee #2.

Author Rebuttals to Second Revision:

Referee #1:

The authors have addressed all of my comments. This is a great paper. Steve Jacobsen

Referee #2:

The authors have done an excellent job of addressing my remaining questions. The new data with varying salt and Mg²⁺ conditions, droplet fusion and FRAP clearly show a physiologically relevant difference between the H2B.2 and H2B.8 containing chromatin phases. The authors need to include the specific buffer conditions used for the FRAP and droplet fusion studies (Extended Data Fig. 3d&e) in the methods. Other than this required small addition, I have no further concerns. This study should be of much interest to both the chromatin and phase-separation communities.

We thank the referee for the constructive comment. We have revised the methods accordingly to make it clear that the buffer conditions used for FRAP and droplet fusion studies were described in the previous methods section.

At Lines 801-803, it now reads:

“In vitro FRAP (Fluorescence Recovery After Photobleaching) and droplet fusion”

“In vitro FRAP and droplet fusion experiments were carried out after chromatin condensates were formed using conditions detailed above.”

Referee #3:

The authors supported the observed transmission defect by adding additional experimental data. While the defect remains small and is only detectable under artificial settings, it nevertheless is significant and possibly biological relevant. I have no further comments to be addressed.